# The Early Bird Catches the Worm: A Positional Decay Reweighting Approach to Membership Inference in Large Language Models

## Abstract

Membership inference attacks (MIAs) against large language models (LLMs) aim to detect whether a specific data point was included in the training dataset. While existing likelihood-based MIA methods have shown promise, they typically aggregate token-level scores using uniform weights (e.g., via simple averaging). We argue that this uniform aggregation is suboptimal because it fails to explicitly account for the decaying nature of memorization signals. Inspired by the information-theoretic principle that conditioning reduces uncertainty, we hypothesize that the memorization signal is strongest at the beginning of a sequence—where model uncertainty is highest—and generally decays with token position. To leverage this insight, we introduce Positional Decay Reweighting (PDR), a simple and lightweight plug-and-play method. PDR applies decay functions to explicitly re-weight token-level scores from existing likelihood-based MIA methods, systematically amplifying the strong signals from early tokens while attenuating noise from later ones. Extensive experiments show that PDR consistently enhances a wide range of advanced methods across multiple benchmarks.

## 1 Introduction

As Large Language Models (LLMs) are trained on vast and diverse corpora from the internet (Achiam et al., 2023; Touvron et al., 2023b), there exists a non-negligible risk that sensitive or personally identifiable information may be memorized and unintentionally exposed through model outputs (Grynbaum & Mac, 2023; Mozes et al., 2023). Membership Inference Attack (MIA) aims to determine whether a sample was part of a model's training set (Hu et al., 2022b; Wu & Cao, 2025). MIA has become increasingly critical in scenarios such as training data auditing, copyright infringement detection, and test set contamination analysis (Bertran et al., 2023; Zhang et al., 2025b), where identifying memorized content is essential for ensuring data integrity and compliance.

For LLMs, performing MIA methods introduces several critical challenges. First, the high-dimensionality and semantic richness of natural language make it difficult to define simple decision boundaries between training and non-training samples (Wu & Cao, 2025). Second, the internal representations and prediction behaviors of LLMs are shaped by deeply stacked transformer architectures, whose complexity often obfuscates direct interpretability (Achiam et al., 2023; Touvron et al., 2023b). Third, many real-world deployments of LLMs, such as commercial APIs, only provide black-box access, further limiting the attacker's ability to probe model internals or gradients (Achiam et al., 2023). These factors collectively make membership inference in the context of LLMs a significantly harder problem compared to that in traditional MIA methods.

Existing MIA methods for LLMs can be broadly categorized into likelihood-based and non-likelihood-based approaches. Among dominant likelihood-based methods, Loss (Yeom et al., 2018) averages log-likelihoods across all tokens in the test sequence to serve as the detection score, while Min-k% (Shi et al., 2024) and Min-k%++ (Zhang et al., 2025b) select some the tokens with the lowest-probability from a sequence to compute its detection score. Methods like ReCaLL (Xie et al., 2024), and Ref (Carlini et al., 2021) introduce a reference point to calibrate likelihood-based scores, either prefixing target data points with non-member context or using a smaller auxiliary LLM. FSD (Zhang et al., 2025a) fine-tunes the target LLM on some non-member samples before

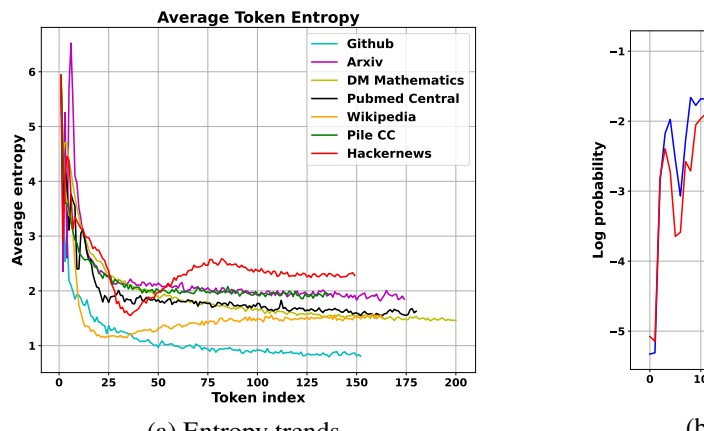 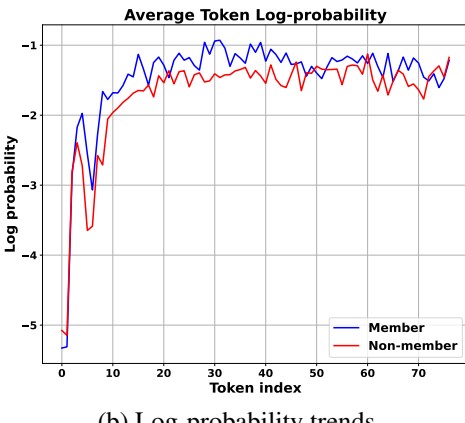

(a) Entropy trends             (b) Log-probability trends

Figure 1: Visualization of (a) token-level entropy on subsets of the challenging Mimir dataset and (b) the average token-level log-probability for members and non-members on WikiMIA dataset for LLaMA-13B model.

computing the likelihood-based score (Zhang et al., 2025a). While varied in their specific strategies, these methods share a fundamental, unaddressed limitation: they typically aggregate token-level scores using uniform weights. Whether aggregating scores from all tokens or a selected subset in the sequence, they assign equal weight to each included token's contribution to the final detection score, failing to explicitly account for the positional decay of memorization signals.

Our work is motivated by a key insight from information theory: conditioning on more information cannot increase entropy (Shannon, 1948). In autoregressive models, this implies that as more context accumulates, the model's predictive uncertainty for the same token should not increase. This motivated us to hypothesize an empirical trend: in typical language generation, token-level entropy usually tends to decrease as a sequence progresses. We empirically investigate this hypothesis in Fig. 1 (a). The results reveal a dominant, albeit sometimes noisy, downward trend across diverse datasets. While corpora with heterogeneous structures like ArXiv and HackerNews show volatility, all datasets share a crucial characteristic: a high-entropy initial region that drops sharply. Consequently, an unusually confident prediction (high probability) for an early, high-entropy token is far more surprising—and thus more indicative of memorization—than comparable confidence later in the sequence. This is because in later positions, the abundance of context makes predictions easier for both member and non-member samples, thus shrinking the discriminative gap between them.

This leads to our core hypothesis: *the memorization signal is not uniformly distributed, but is concentrated in the initial stages of a sequence, with its discriminative power generally decaying with token position.* However, existing likelihood-based methods, by utilizing uniform score aggregation, dilute this skewed and powerful signal with less informative signals from later positions. Capitalizing on this key observation, we introduce Positional Decay Reweighting (PDR), a simple, effective, and "plug-and-play" method designed to align the scoring process with this positional signal decay. By applying monotonic decay functions (e.g., linear, exponential, polynomial), PDR systematically amplifies the high-value signals from early tokens while attenuating potential noise from later ones, thereby focusing the inference on the most informative parts of the sequence. Its key advantage is versatility: PDR can be seamlessly integrated into existing likelihood-based scoring functions. Extensive experiments validate that this straightforward modification yields substantial and consistent performance gains, improving upon advanced Min-$k$%++ by up to 4.7 AUROC points on the WikiMIA benchmark of 128 length. Our main contributions can be summarized as follows: *(1)* We are the first to systematically demonstrate and analyze the positional decay of memorization signals from the view of token-level entropy, exposing the "uniform score aggregation" limitation inherent in prior methods. *(2)* We propose Positional Decay Reweighting (PDR), a lightweight, plug-and-play framework that reweights token scores to amplify early signals while attenuating later noise. *(3)* Our results across diverse LLMs and benchmarks establish PDR as an effective plug-and-play method, yielding notable performance gains especially for Min-$k$%++.

## 2 RELATED WORK

**Membership Inference Attacks.** Membership Inference Attacks (MIA) have long been a core topic in security and privacy (Shokri et al., 2017; Yeom et al., 2018). These attacks aim to determine whether a specific data point was included in the training dataset of a learning model. Extensive investigations across both vision (Dubiński et al., 2024) and language (Watson et al., 2022; Mattern et al., 2023) domains have led to advances in attack methodologies. Notable examples include **LiRA** (Carlini et al., 2022), which leverages shadow models to estimate logit distributions for likelihood-ratio tests, and **RMIA** (Zarifzadeh et al., 2023), which constructs robust pairwise likelihood-ratio tests using a population of reference models. Beyond exact matching, **RaMIA** (Tao & Shokri, 2025) extends the scope by testing if the model was trained on any data within a specified semantic range, capturing privacy risks from similar or partially overlapping data. These developments have provided deeper insights into privacy risks (Mireshghallah et al., 2022), test-set contamination (Oren et al., 2023), and copyright protection (Meeus et al., 2023; Duarte et al., 2024).

**Membership Inference Attacks for LLM.** While MIA is a long-standing problem, its application to the pre-training stage of LLMs poses unique challenges, such as the impracticality of training shadow models and data characteristics that make inference difficult (Shi et al., 2024; Zhang et al., 2025b). To this end, a category of existing methods focuses on the attack framework itself, for instance, the distribution-free **DF-MIA** (Huang et al., 2025) for fine-tuned models, and **MIA-Tuner** (Fu et al., 2025), which cleverly uses soft prompt tuning. Another is likelihood-based methods. The foundational **Loss** method (Yeom et al., 2018) uses the average negative log-likelihood to compute the anomaly score, and the **Ref** (Carlini et al., 2021) method calibrates this score using a smaller reference model. **Neighbor** (Mattern et al., 2023) eliminates the need for access to the training data distribution by comparing the model score of a sample to those of its synthetically generated neighbors. More advanced techniques focus on outlier tokens; **Min-$k$%** (Shi et al., 2024) averages the probabilities of the tokens with the lowest scores, while **Min-$k$%++** (Zhang et al., 2025b) extends this by normalizing token-level scores before selection. Other recent works further refine likelihood-based scoring, such as **ReCaLL** (Xie et al., 2024), which scores samples by measuring the change in likelihood when conditioned on a non-member prefix, or by fine-tuning the model to amplify score differences (Zhang et al., 2025a). **CAMIA** (Chang et al., 2025) learns to distinguish between member and non-member samples by aggregating multiple dynamic signals, including the rate of change in token loss. Different from these methods, our work analyzes the positional decay of memorization signals through the lens of token entropy. Based on this insight, we introduce a plug-and-play framework to enhance existing likelihood-based methods, rather than proposing an entirely new scoring function.

**Token Position in LLMs.** The importance of token position has been recognized in various domains of large language model research. For instance, to optimize inference, methods like Token-Butler (Akhauri et al., 2025) predict critical tokens to prune the KV-Cache, while OrthoRank (Shin et al., 2025) identifies important tokens by measuring their hidden state orthogonality to "sink token". The significance of token-level analysis extends to the sub-token level, where understanding internal character positions can improve performance on fine-grained tasks (Xu et al., 2024). Different from them, our work investigates how token positions impact membership inference, enhancing existing likelihood-based MIA methods through position-based token reweighting.

## 3 BACKGROUND

In this section, we first formalize the problem of pre-training data detection as defined in prior studies (Shokri et al., 2017; Shi et al., 2024; Duan et al., 2024), and then the likelihood-based scoring functions for MIA methods in LLMs.

### 3.1 PROBLEM STATEMENT

Pre-training data detection is cast as a membership inference attack (MIA) (Shokri et al., 2017). Denote a pre-trained auto-regressive LLM as $M$ and its unknown training corpus as $\mathcal{D}$. For an arbitrary text sample $x$, MIA aims to infer whether $x \in \mathcal{D}$ (member sample) or $x \notin \mathcal{D}$ (non-member sample). Let $s(x; M)$ represent the scoring function that assigns a real-valued "membership" score to $x$ based on $M$'s outputs. We make a binary decision via

$$\hat{y} = \mathbb{I}\big(s(x; M) \geq \epsilon\big), \tag{1}$$

where $\epsilon$ is a case-specific threshold and $\mathbb{I}(\cdot)$ is the indicator function. Consistent with the grey-box setting (Shi et al., 2024; Duan et al., 2024; Zhang et al., 2025b), we assume that only $M$'s output statistics (logits, token probabilities, loss values) are accessible; internal weights and gradients remain hidden. Designing an effective $s(x; M)$ to maximize the separation between member and non-member distributions is at the core of the detection task.

## 3.2 LIKELIHOOD-BASED SCORE FUNCTIONS

Modern LLMs are trained by maximizing the likelihood of training token sequences (Radford et al., 2019; Brown et al., 2020). Concretely, given a sequence $\boldsymbol{x} = (x_1, \ldots, x_T)$, an auto-regressive LLM factorizes its joint probability using the chain rule: $p(\boldsymbol{x}) = \prod_{t=1}^{T} p(x_t \mid x_{<t})$, where $x_{<t} = (x_1, \ldots, x_{t-1})$ is the prefix context. At inference time, the model generates text token by sampling from the conditional distribution $p(\cdot \mid x_{<t})$. In light of this, researchers usually design likelihood-based scoring functions to detect pretraining data in LLMs (Yeom et al., 2018). For example, based on the observation that members tend to have higher log-likelihood than non-members, the loss-based score (Yeom et al., 2018) is defined as the (negative) log-likelihood of the input sequence,

$$s_{\text{loss}}(x) = \frac{1}{T} \sum_{t=1}^{T} \log p(x_t \mid x_{<t}), \tag{2}$$

where we flip the sign of the conventional loss-based score so that, consistent with other methods, higher scores indicate stronger membership. Instead of using the likelihood of all tokens, Min-$k$% (Shi et al., 2024) selects the $k$% tokens with the smallest log-probabilities and averages them:

$$s_{\text{Min-}k\%}(x) = \frac{1}{|\mathcal{S}_k|} \sum_{x_t \in \text{Min-}k\%(\boldsymbol{x})} \log p(x_t \mid x_{<t}), \tag{3}$$

where $\mathcal{S}_k$ represents the set of token positions corresponding to the smallest $k$% log-probabilities in the sequence. The intuition is that a non-member example is more likely to include a few outlier words with low likelihoods than members. Other methods are deferred to Appendix A.

## 4 METHODOLOGY

In this section, we first present our core motivation based on an empirical observation about token entropy. We then introduce our general, plug-and-play weighting method, Positional Decay Reweighting (PDR), and demonstrate how apply it to enhance existing likelihood-based scores.

### 4.1 MOTIVATION

Our methodology is built on a key insight into how autoregressive language models process information. From an information-theoretic perspective, a fundamental principle is that conditioning on more information cannot increase entropy, i.e., $H(z|x, y) \leq H(z|y)$. In the context of autoregressive models, the uncertainty at each step can be quantified by the conditional entropy of the next token over the vocabulary $V$, given the prefix context $x_{<t}$:

$$H(p(\cdot|x_{<t})) = -\sum_{v \in V} p(v|x_{<t}) \log p(v|x_{<t}). \tag{4}$$

Although the classic principle compares the entropy of the same random variable, whereas here we are comparing the entropy for different variables ($x_t$ and $x_{t+1}$), it is a widely observed empirical phenomenon that the entropy at position $t$ is frequently greater than at position $t + 1$.

To empirically investigate this phenomenon, we visualized the average token-level entropy across multiple diverse datasets, as shown in Fig. 1 (a). The visualization reveals two crucial findings. First, despite the varied nature of the corpora, they all exhibit a **dominant trend**: a high-entropy initial region followed by a general downward trend as the sequence progresses. Second, it highlights key differences in these trends; while datasets like Github and Wikipedia show a relatively smooth decay, corpora with more heterogeneous structures—such as ArXiv (with section headings and equations) and HackerNews (mixing prose, code, and quotes)—display significantly more volatility.

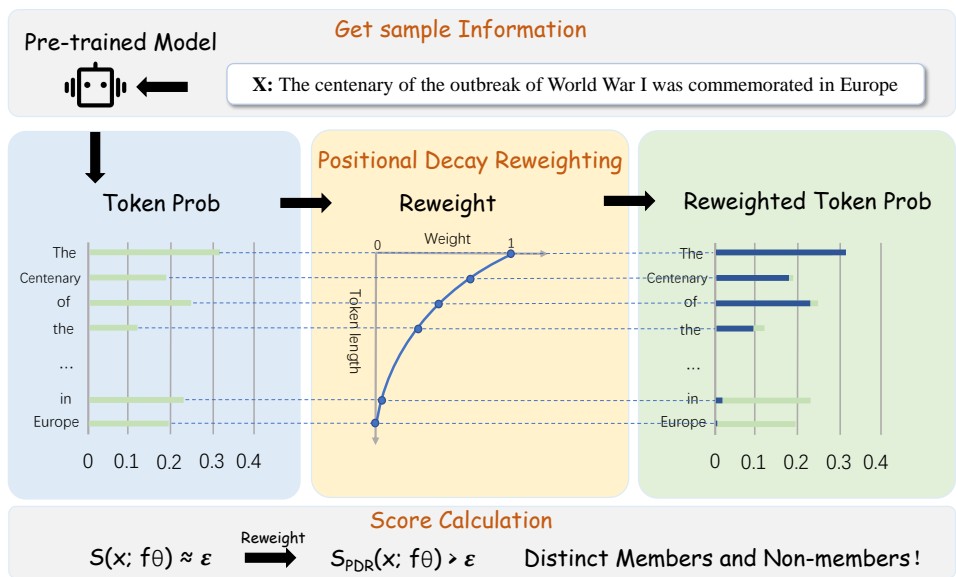

Figure 2: **Overview of Positional Decay Reweighting (PDR).** Our method reweights the predictive probabilities of input samples based on token positions, emphasizing early tokens with higher weights. This reweighting enhances the distinction between member and non-member samples by amplifying critical signals in the score $\mathcal{S}$, making it more effective for MIA. The framework is lightweight, plug-and-play, and can be applied to various likelihood-based scoring methods.

These empirical findings have direct implications for membership inference, as the initial high-entropy region provides a unique setting to distinguish memorization from generalization. An unusually confident (i.e., high-probability) prediction for a token in such a position strongly suggests that this confidence does not stem from contextual generalization, but rather from rote memorization of specific sequences in its training set. Conversely, in later positions (low-entropy regions), the abundance of context makes predictions easier for **both** member and non-member samples, thus shrinking the discriminative gap between them. This is directly confirmed by Fig. 1 (b), which shows that this gap is largest at the beginning of the sequence and diminishes over time.

This analysis leads to our refined core hypothesis: *the memorization signal is not uniformly distributed but is heavily skewed towards the beginning of a sequence, with its strength generally decaying with token position.* This crucial insight reveals a limitation in existing likelihood-based MIA methods. Whether they use scores from all tokens (like Loss) or from a subset of low-likelihood tokens (e.g., Min-$k$%), **they overwhelmingly rely on uniform weighting schemes** . By treating the scores from different positions as equally important, they dilute the potent, high-fidelity signals concentrated in the early positions with noisy, less informative signals from the end. This oversight prevents them from fully exploiting the powerful evidence of memorization. Therefore, a principled, position-aware approach is not merely an incremental improvement, but a necessary step to enhance MIA performance.

### 4.2 PLUG-AND-PLAY POSITIONAL DECAY REWEIGHTING (PDR)

Based on our core hypothesis established above—that memorization signals are heavily skewed towards the beginning of a sequence—we argue that the performance of likelihood-based MIA methods is fundamentally limited by their uniform scoring mechanism. To rectify this, we propose Positional Decay Reweighting (PDR): a simple, effective, and "plug-and-play" framework designed to inject this crucial positional prior into existing methods. Our overview is illustrated in Fig. 2.

PDR operates by re-weighting token-level scores using monotonically decreasing functions based on a token's position $t$ in a sequence of length $T$. This systematically assigns higher importance to earlier tokens, where the signal is strongest, and lower importance to later ones. We explore three simple, standardized, and effective families of decay functions:

1. **Linear Decay:** This function linearly decreases the weight from 1. The rate of decay is controlled by a single hyperparameter $\alpha \in [0, 1]$:

$$w_{\text{linear}}(t) = 1 - \alpha \left( \frac{t-1}{T-1} \right), \tag{5}$$

where $T$ is the total sequence length. When $\alpha = 0$, all tokens are weighted equally, reducing to the original unweighted score.

2. **Exponential Decay:** This function applies a sharper, non-linear decay, placing a much stronger emphasis on the initial tokens:

$$w_{\text{exp}}(t) = \exp(-\alpha \cdot (t-1)). \tag{6}$$

The hyperparameter $\alpha \geq 0$ controls the steepness of the decay.

3. **Polynomial Decay:** This function provides a flexible decay curve whose shape is controlled by the exponent $\alpha$. The hyperparameter $\alpha > 0$ determines the curvature of the decay. Values of $\alpha > 1$ result in a slower initial decay, while values $0 < \alpha < 1$ lead to a faster initial decay:

$$w_{\text{poly}}(t) = \left( 1 - \frac{t-1}{T-1} \right)^{\alpha}. \tag{7}$$

We defer the visualization of three weight decay functions into Fig.6 of Appendix B.

### 4.3 APPLYING PDR TO MIA SCORING FUNCTIONS

A key advantage of PDR is its "plug-and-play" nature. It operates as a lightweight wrapper designed to correct existing likelihood-based methods, requiring no modification to the target model's architecture or training process. This makes it a broadly applicable technique. We now demonstrate how PDR integrates with two representative scoring functions.

For methods that aggregate scores across the entire sequence, such as the standard Loss score in equation 2, PDR injects the positional prior by applying weights to each token's log-probability before aggregation. The resulting PDR-Loss score is defined as:

$$s_{\text{PDR-Loss}}(x) = \frac{1}{T} \sum_{t=1}^{T} w(t) \cdot \log P(x_t | x_{<t}). \tag{8}$$

The integration is more nuanced for outlier-based methods like Min-$k$% in equation 3. Here, a crucial detail is the order of operations. To preserve the integrity of the outlier selection process, PDR is applied *after* the tokens have been selected based on their original, unweighted scores. The re-weighting then uses the *original position* of these selected tokens, ensuring that we are amplifying the most informative signals as identified by the baseline method. The PDR-Min-$k$% score is thus:

$$s_{\text{PDR-Min-}k\%}(x) = \frac{1}{|\mathcal{S}_k|} \sum_{t \in \mathcal{S}_k} w(t) \cdot \log P(x_t | x_{<t}), \tag{9}$$

where $\mathcal{S}_k$ is the set of token positions with the smallest $k$% log-probabilities.

PDR can also be combined with other scoring functions, such as the reference-based method (Ref), the normalized outlier method (Min-$k$%++), and finetuned-based FSD . The full set of PDR-enhanced scoring functions and algorithm are detailed in Appendix C. By systematically amplifying the signal from critical early tokens, PDR aims to widen the score distribution gap between member and non-member samples, thereby enhancing overall detection performance.

## 5 EXPERIMENTS

### 5.1 SETUP

**Benchmarks.** We evaluate our method on two commonly-used benchmarks for pre-training data detection. (1) **WikiMIA** (Shi et al., 2024) uses Wikipedia texts, distinguishing members by

Table 1: AUROC results on WikiMIA benchmark (Shi et al., 2024). **w/ LPDR** utilizes our linear weights for reweighting. *Ori.* and *Para.* denote the original and paraphrased settings. [†]Neighbor results are from Zhang et al. (2025b). For each method pair, the higher score is in **bold**. The performance gains of our method on the average results are highlighted in purple.

| Len. | Method | Mamba-1.4B | | Pythia-6.9B | | LLaMA-13B | | GPT-NeoX-20B | | OPT-66B | | Average | |
|------|--------|------|------|------|------|------|------|------|------|------|------|------|------|
| | | *Ori.* | *Para.* | *Ori.* | *Para.* | *Ori.* | *Para.* | *Ori.* | *Para.* | *Ori.* | *Para.* | *Ori.* | *Para.* |
| 32 | Lowercase | 60.9 | 60.6 | 62.2 | 61.7 | 64.0 | 63.2 | 68.3 | 66.9 | 62.8 | 62.3 | 63.7 | 63.0 |
| | Zlib | 61.9 | 62.3 | 64.4 | 64.2 | 67.8 | 68.3 | 69.3 | 68.5 | 65.8 | 65.3 | 65.8 | 65.7 |
| | [†]Neighbor | 64.1 | 63.6 | 65.8 | 65.5 | 65.8 | 65 | 70.2 | 68.3 | 68.2 | 66.7 | 66.8 | 65.8 |
| | Loss | 61.0 | 61.3 | 63.8 | 64.1 | 67.5 | 68.0 | 69.1 | 68.6 | 65.6 | 65.3 | 65.4 | 65.5 |
| | **w/ LPDR** (Ours) | **61.5** | **61.8** | **64.0** | **64.2** | **67.7** | **68.2** | 68.9 | 68.3 | 65.6 | 65.0 | **65.5**[+0.1] | 65.5 |
| | Ref | 62.2 | 62.3 | 63.6 | 63.5 | 57.9 | 56.2 | 67.6 | 66.7 | 68.6 | 67.9 | 64.0 | 63.3 |
| | **w/ LPDR** | 62.2 | 62.3 | 63.5 | 63.5 | 57.8 | 56.1 | 67.4 | 66.6 | 68.6 | **68.0** | 63.9 | 63.3 |
| | Min-$k$% | 63.3 | 62.9 | 66.3 | 65.1 | 66.8 | 66.2 | 72.2 | 69.6 | 67.5 | 65.8 | 67.2 | 65.9 |
| | **w/ LPDR** | 63.5 | 63.1 | 66.3 | 65.1 | 66.8 | 66.2 | 72.0 | 69.4 | **67.7** | 65.8 | **67.3**[+0.1] | 65.9 |
| | Min-$k$%++ | 66.4 | 65.7 | 70.3 | 67.6 | 84.4 | 82.7 | 75.1 | 69.7 | 69.7 | 67.0 | 73.2 | 70.5 |
| | **w/ LPDR** | **67.4** | **66.3** | **70.8** | **67.7** | **85.9** | **84.1** | **75.2** | 69.5 | **70.2** | **67.1** | **73.9**[+0.7] | **70.9**[+0.4] |
| 64 | Lowercase | 57.0 | 57.0 | 58.2 | 57.7 | 62.0 | 61.0 | 66.3 | 65.6 | 61.1 | 60.0 | 60.9 | 60.3 |
| | Zlib | 60.4 | 59.1 | 62.6 | 61.6 | 65.3 | 65.3 | 68.1 | 66.5 | 63.9 | 62.2 | 64.1 | 62.9 |
| | [†]Neighbor | 60.6 | 60.6 | 63.2 | 63.1 | 64.1 | 64.7 | 67.1 | 67.4 | 64.1 | 64.6 | 63.8 | 64.1 |
| | Loss | 58.2 | 56.4 | 60.7 | 59.3 | 63.6 | 63.1 | 66.6 | 64.4 | 62.3 | 60.3 | 62.3 | 60.7 |
| | **w/ LPDR** | **59.7** | **59.5** | **62.6** | **62.4** | **65.0** | **66.2** | **67.6** | **67.2** | **64.2** | **63.1** | **63.8**[+1.5] | **63.7**[+3.0] |
| | Ref | 60.6 | 59.6 | 62.4 | 62.9 | 63.4 | 60.9 | 66.0 | 66.0 | 66.9 | 67.8 | 63.9 | 63.5 |
| | **w/ LPDR** | **61.1** | **60.8** | **63.3** | **64.0** | 59.8 | 57.9 | **66.8** | **67.2** | **68.2** | **69.1** | 63.9 | **63.8**[+0.3] |
| | Min-$k$% | 61.7 | 58.0 | 65.0 | 61.1 | 66.0 | 63.5 | 72.2 | 66.1 | 66.5 | 62.5 | 66.3 | 62.2 |
| | **w/ LPDR** | **62.9** | **61.8** | **66.7** | **65.1** | **67.4** | **67.2** | 70.8 | **68.7** | **68.1** | **66.0** | **67.2**[+0.9] | **65.8**[+3.6] |
| | Min-$k$%++ | 67.2 | 62.2 | 71.6 | 64.2 | 84.3 | 78.8 | 76.5 | 66.2 | 69.8 | 63.3 | 73.9 | 66.9 |
| | **w/ LPDR** | **68.2** | **65.5** | **72.1** | **68.3** | **87.2** | **84.3** | 76.4 | **68.2** | **70.1** | **66.6** | **74.8**[+0.9] | **70.6**[+3.7] |
| 128 | Lowercase | 58.5 | 57.7 | 60.5 | 59.9 | 60.6 | 56.3 | 68.0 | 67.6 | 58.9 | 57.6 | 61.3 | 59.8 |
| | Zlib | 65.6 | 65.3 | 67.6 | 67.4 | 69.7 | 69.6 | 72.3 | 72.0 | 67.3 | 66.9 | 68.5 | 68.2 |
| | [†]Neighbor | 64.8 | 62.6 | 67.5 | 64.3 | 68.3 | 64 | 71.6 | 69.6 | 67.7 | 63.4 | 68.0 | 64.8 |
| | Loss | 63.3 | 62.7 | 65.1 | 64.7 | 67.8 | 67.2 | 70.7 | 69.7 | 65.5 | 64.5 | 66.5 | 65.7 |
| | **w/ LPDR** | 63.6 | **64.1** | 65.6 | **66.6** | 68.7 | **69.1** | 70.7 | **71.4** | **66.7** | **66.9** | **67.1**[+0.6] | **67.6**[+1.9] |
| | Ref | 62.0 | 61.1 | 63.3 | 62.9 | 62.6 | 59.7 | 68.3 | 68.4 | 66.9 | 67.0 | 64.6 | 63.8 |
| | **w/ LPDR** | **64.1** | **64.6** | **65.1** | **65.9** | **64.9** | **61.7** | **69.5** | **70.2** | **68.6** | **69.5** | **66.4**[+1.8] | **66.4**[+2.6] |
| | Min-$k$% | 66.8 | 64.4 | 69.5 | 67.0 | 71.5 | 68.6 | 75.6 | 73.0 | 70.6 | 67.2 | 70.8 | 68.0 |
| | **w/ LPDR** | 65.5 | **65.8** | 67.8 | **68.9** | 71.2 | **71.0** | 74.5 | **75.2** | 70.6 | **69.9** | 69.9 | **70.2**[+2.2] |
| | Min-$k$%++ | 67.7 | 63.3 | 69.8 | 65.9 | 83.8 | 76.2 | 75.4 | 70.6 | 71.1 | 67.0 | 73.6 | 68.6 |
| | **w/ LPDR** | **70.2** | **68.2** | **72.4** | **72.2** | **88.4** | **84.3** | **75.7** | **72.6** | **72.9** | **69.5** | **75.9**[+2.3] | **73.3**[+4.7] |

timestamps, and includes different length text for both *original* and *paraphrased* settings. (2) **MIMIR** (Duan et al., 2024), built on the Pile dataset (Gao et al., 2020), is more challenging as it minimizes distributional and temporal shifts between member and non-member data.

**Baselines.** We consider several representative and advanced methods as our baselines. A fundamental approach is **Loss** (Yeom et al., 2018), which directly uses all tokens' likelihood as a detection score. Reference-based methods include **Ref** (Carlini et al., 2021), employing a smaller language model for likelihood calibration, as well as **Zlib** and **Lowercase** (Carlini et al., 2021), use zlib compression entropy or lowercase text likelihood for the same purpose. Besides, **Neighbor** (Mattern et al., 2023) evaluates membership by comparing the sample's score against those of its synthetically generated neighbors. Focusing on the most indicative tokens, **Min-$k$%** (Shi et al., 2024) averages the lowest $k$% of token scores. An enhancement to this is **Min-$k$%++** (Zhang et al., 2025b), which incorporates score normalization for each token before selection. What's more, we include **FSD** (Zhang et al., 2025a), leverages score differences obtained after fine-tuning the model on non-member data.

**Models.** For WikiMIA, we use Pythia (Biderman et al., 2023) (2.8B, 6.9B, 12B), LLaMA (Touvron et al., 2023a) (13B, 30B), GPT-NeoX (Black et al., 2022)(20B), OPT (Zhang et al., 2022) (66B), and Mamba (Gu & Dao, 2023) (1.4B, 2.8B). For MIMIR, we follow Duan et al. (2024) and use the Pythia model series (160M, 1.4B, 2.8B, 6.9B, 12B). For FSD, we follow Zhang et al. (2025a) and use GPT-J-6B, OPT-6.7B, Pythia-6.9B, LLaMA-7B, and GPT-NeoX-20B.

**Metrics and Settings.** Following standard practice (Carlini et al., 2021; Shi et al., 2024), we use AUROC as the primary metric and also report True Positive Rate (TPR) at low False Positive Rates. For brevity, we use LPDR, EPDR, and PPDR to denote our PDR with Linear, Exponential, and Polynomial decay. We use a commonly-used $k = 20$ for Min-$k$% and Min-$k$%++. In the main body, we primarily report results for LPDR. To demonstrate the general effectiveness of a simple and strong positional prior, we use a fixed $\alpha = 1$ for all our experiments with LPDR except for very short sequences (WikiMIA, $T = 32$), where such a sharp is suboptimal. More details about datasets, baselines and settings are deferred to Appendix D.

## 5.2 MAIN RESULTS.

**Results on WikiMIA.** As shown in Tab. 1, we report the AUROC results of different methods on WikiMIA with varying sequence lengths of {32, 64, 128} on different backbones; please see Appendix E.1 for overall results on more methods (including our EPDR, PPDR), backbones and TPR numbers. Besides, we also provide the best results on WikiMIA dataset cross five model in Appendix E.2 and plot the ROC curves in Appendix E.3 to demonstrate the consistent superiority of our method across various False Positive Rate (FPR) thresholds. We observe that introducing our proposed linear positional decay reweighting strategy generally enhances the performance of existing likelihood-based MIA methods. This improvement is especially evident on the advanced Min-$k$%++. For instance, when combined with the Min-$k$%++ method, the performance gains from our LPDR become more pronounced as sequence length increases. Our LPDR improves the average AUROC by 0.7 (*Ori.*) and 0.4 (*Para.*) for length 32, by 0.9 (*Ori.*) and 3.7 (*Para.*) for length 64, and achieves the most significant gains of 2.3 (*Ori.*) and 4.7 (*Para.*) for length 128. These results prove the effectiveness of the our designed weight decay method in re-weighting token-level scores.

**Combination with FSD on WikiMIA.** Since ours is a plug-and-play reweighting method, it can also be used to enhance the finetune-based FSD. We perform the experiments on WikiMIA dataset with different LLMs, where we first finetune LLMs with non-member samples following its official code, then use Min-$k$% and Min-$k$%++ as its score functions, respectively. To combine ours with FSD, we apply our LPDR to reweight the score functions. We show the results in Fig. 3 and defer details into Appendix F. We can find that our linear PDR provides consistent improvements on FSD no matter with Min-$k$%++ or Min-$k$%++ as the score functions. It demonstrates that ours is also beneficial for finetune-based methods by reweighting its score functions.

Table 2: AUROC scores of various MIA methods over five Pythia models on the Mimir dataset. Pub, Wiki, and Hack denote Pubmed Central, Wikipedia (en) and HackerNews, respectively. Average* scores are computed by excluding Arxiv and HackerNews. [†]Neighbor results are from Zhang et al. (2025b), induces significant extra computational cost than others ($25\times$ in this case), for which reason we don't run on the 12B model.

| Method | Wiki | Pile-CC | Pub | DM Math | GitHub | ArXiv | Hack | Average* |
|---|---|---|---|---|---|---|---|---|
| Lowercase | 52.2 | 49.3 | 51.1 | 48.9 | 71.1 | 51.5 | 50.9 | 54.5 |
| Zlib | 52.7 | 50.4 | 50.4 | 48.1 | 71.9 | 51.4 | 50.8 | 54.7 |
| [†]Neighbor | 51.9 | 50.1 | 49.2 | 47.4 | 69.3 | 51.5 | 51.5 | 53.6 |
| Loss | 51.9 | 50.3 | 50.3 | 48.5 | 70.8 | 52.1 | 51.2 | 54.4 |
| **w/ LPDR** | **52.8** | **50.7** | 50.3 | **48.6** | **70.9** | 51.8 | 51.3 | **54.6** |
| Min-$k$% | 51.8 | 50.7 | 50.9 | 49.2 | 70.9 | 52.3 | 52.4 | 54.7 |
| **w/ LPDR** | **54.2** | **51.2** | **51.0** | **49.5** | **71.0** | 51.4 | 51.5 | **55.4** |
| Min-$k$%++ | 54.0 | 50.5 | 51.9 | 50.3 | 70.4 | 52.6 | 52.9 | 55.4 |
| **w/ LPDR** | **55.5** | **50.8** | **52.4** | 50.3 | 70.1 | 52.7 | 52.1 | **55.8** |

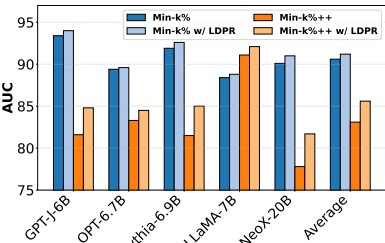

Figure 3: AUROC comparison of our LPDR method when integrated with Min-$k$% and Min-$k$%++ across various LLMs on WikiMIA dataset within the FSD framework.

**MIMIR Results.** As noted by prior work, MIMIR is particularly difficult because its training and non-training texts are sourced from the same datasets, minimizing distributional shifts. Furthermore, we identify that the sub-datasets within MIMIR exhibit notable differences in structural composition. As visually confirmed by their volatile entropy profiles in Fig. 1 (a), corpora like *ArXiv* and *HackerNews* are structurally heterogeneous. This distinguishes them from more homogeneous corpora like Wikipedia or GitHub. We treat the heterogeneous datasets as stress tests and compute an `Average*` score on the five sub-datasets that align with our method's positional prior. As listed in Tab. 2, on this benchmark, baselines themselves perform close to random guess, underscoring its difficulty. We can find that introducing ours can improve the Loss, Min-$k$%, and Min-$k$%++. It validates the efficacy of PDR's positional prior in structurally homogeneous text datasets. More detailed results are deferred to Appendix G.

## 5.3 FURTHER ANALYSIS

**Ablation Study about Weight Design.** We conduct an ablation study with several alternative weighting schemes. We consider applying our PDR-generated weights in different Order, including *Random*, a shuffled sequence, and *Reverse*, a monotonically increasing sequence. Furthermore, we use token-level entropy as a direct weight, either from a single *Sample* or the entire *Dataset*. Additionally, for Min-$k$% and Min-$k$%++ methods, we compare our standard approach (reweighting the subset of scores after selection) with an alternative where reweighting is applied to the full sequence *Before* the lowest $k$% of scores are selected.

As shown in Tab. 3, both *Random* and *Reverse* orders significantly degrade performance, confirming that a monotonically decreasing weight is crucial. For Min-$k$% methods, the results show that applying reweighting after selecting the most informative tokens is superior to reweighting the entire sequence *Before* selection. This suggests that the initial selection effectively isolates the most relevant signals, which are then more effectively amplified by our PDR. Another insightful comparison, is with entropy-based weighting. Furthermore, comparing our three PDR variants (LPDR, EPDR, and PPDR), we observe that their performance is generally comparable, and all can effectively enhance the performance of most baseline methods. While using *Dataset*-level entropy yields strong performance, this approach is impractical as it requires full test dataset statistics. Conversely, *Sample*-level entropy is practical but ineffective due to high variance. PDR strikes a critical balance: its simple, data-agnostic prior is a powerful and practical plug-and-play strategy, achieving results competitive with the impractical dataset-level entropy approach.

**Ablation Study about Weight Design.** We conduct an ablation study with several alternative weighting schemes. We consider applying our PDR-generated weights in different Order, including *Random*, a shuffled sequence, and *Reverse*, a monotonically increasing sequence. Furthermore, we use token-level entropy as a direct weight, either from a single *Sample* or the entire *Dataset*. Additionally, following CAMIA's (Chang et al., 2025) concept of loss decreasing rate, we explore a dynamic strategy where we fit a linear slope to each sample's loss sequence and use this fitted slope as the decay parameter $\alpha$. For Min-$k$% and Min-$k$%++ methods, we compare our standard approach (reweighting the subset of scores after selection) with an alternative where reweighting is applied to the full sequence *Before* the lowest $k$% of scores are selected.

As shown in Tab. 3, both *Random* and *Reverse* orders significantly degrade performance, confirming that a monotonically decreasing weight is crucial. For Min-$k$% methods, the results show that applying reweighting after selecting the most informative tokens is superior to reweighting the entire sequence *Before* selection. This suggests that the initial selection effectively isolates the most relevant signals, which are then more effectively amplified by our PDR. Another insightful comparison is with entropy-based weighting. While using *Dataset*-level entropy yields strong performance, this approach is impractical as it requires full test dataset statistics. Conversely, *Sample*-level entropy is practical but ineffective due to high variance. For the fitted slope method (detailed in Appendix H), we find its performance is hampered by the high volatility of token-level losses. This volatility leads to small fitted slopes, as no strong trend can be reliably captured. The resulting weights are therefore too smooth to amplify the signal, yielding only marginal gains on WikiMIA and even more limited improvements on the challenging MIMIR dataset. PDR strikes a critical balance: its simple, data-agnostic prior is a powerful and practical plug-and-play strategy.

Table 3: Ablation study about weighting schemes. We evaluate slope, different weight orderings (Random, Reverse), entropy-based weights (Sample, Dataset), and the reweighting for K-select in Min-$k$% and Min-$k$%++. Results are reported on the WikiMIA dataset using the Pythia-6.9B model with a sequence length of $T = 128$.

| Method | Base | Slope | Weights Order | | Entropy | | K select | w/ PDR(Ours) | | |
| | | | Random | Reverse | Sample | Dataset | Before | LPDR | EPDR | PPDR |
|---|---|---|---|---|---|---|---|---|---|---|
| Loss | 65.1 | 65.1 | 64.5 | 63.4 | 63.2 | **66.4** | - | 65.6 | 64.8 | 66.1 |
| Ref | 63.3 | 63.5 | 61.8 | 58.0 | 62.1 | 63.8 | - | **65.1** | **67.5** | **65.7** |
| Min-$k$% | 69.5 | 69.5 | 64.3 | 59.8 | 64.1 | **70.6** | 68.1 | 67.8 | 69.2 | 66.7 |
| Min-$k$%++ | 69.8 | 69.8 | 66.5 | 61.0 | 68.7 | 70.5 | 70.3 | **72.4** | 71.2 | **72.7** |

**The Effect of Sharpness in Weight Decay.** We analyze the impact of the decay sharpness $\alpha$ in Fig. 4. The results demonstrate LPDR's general effectiveness, as it consistently improves upon the baseline across all tested $\alpha$ values. We observe two distinct trends: for full-sequence methods (Loss and Ref), performance peaks with a sharp decay at $\alpha = 1$, while for outlier-based methods (Min-$k\%$ and Min-$k\%$++), a smoother decay is better. However, even for the latter, the performance gain at $\alpha = 1$ remains substantial. This analysis validates our choice to use a fixed $\alpha = 1$ as a simple and robust default in our main experiments ($T \geq 64$), highlighting PDR's ability to deliver significant gains without requiring method-specific tuning. This choice highlights PDR's ability to deliver significant gains without requiring extensive, method-specific hyperparameter tuning. We plot different weight decay functions with varying varying rates in Fig. 6 of Appendix B. We extend our analysis by exploring the impact of varying truncation ratios in Appendix J. These supplementary experiments consistently demonstrate the robustness and superiority of our proposed method across diverse settings.

**Score Changes when Using Our Method.** To further illustrate the mechanism of PDR, we selected a pair of member and non-member samples that are indistinguishable within Min-$k\%$ for sharing the same score. Fig. 5 visualizes how our LPDR method resolves this ambiguity. By applying a monotonically decreasing weight, PDR enhances the importance of tokens at earlier positions and decreases the importance of tokens at later positions. For the member sample (a), whose memorization signals (high-probability tokens) are concentrated at the beginning, this reweighting process significantly amplifies its final score. In contrast, the non-member sample (b) is less affected. As a result, PDR effectively breaks the tie, creating a clear distinction between the two samples and enhancing the overall detection accuracy.

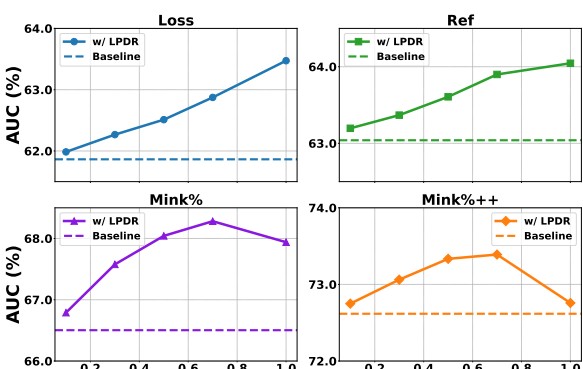

Figure 4: AUC comparison of different $\alpha$ for LPDR on log-likelihood methods (Loss, Ref, Min-$k\%$, Min-$k\%$++), results from Pythia-12B model at WikiMIA original dataset of 64 length.

For additional examples, detailed sample analysis, and score distribution visualizations, please refer to Appendices K, L, and M, respectively.

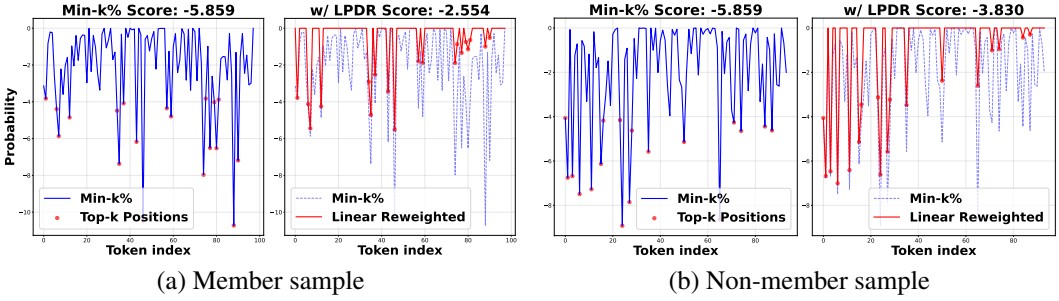

(a) Member sample        (b) Non-member sample

Figure 5: Token-level score changes for (a) member sample and (b) non-member sample after applying LPDR to Min-$k\%$. The red dot means the selected token by Min-$k\%$, blue and red line denote the original and reweighted token-level score, respectively.

## 6 CONCLUSION

In this paper, we address a critical oversight in likelihood-based MIA methods: the failure to account for the positional nature of memorization signals. Based on the observation that a model's predictive uncertainty decreases as a sequence progresses, we argue that membership signals are strongest in early tokens. Existing methods dilute these signals by treating all positions equally. We introduce Positional Decay Reweighting (PDR), a simple, plug-and-play framework that applies a monotoni-

cally decreasing weight to token scores, enhancing existing MIA methods without requiring model changes. Extensive experiments validate that PDR significantly improves the performance of strong baselines like Min-$k$% and Min-$k$%++, both standalone and within advanced frameworks like FSD. Our work provides a more robust approach to membership inference by systematically prioritizing more reliable positional signals, contributing to a deeper understanding of privacy risks in LLMs.

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

# A    LIKELIHOOD-BASED SCORE FUNCIONS

This section provides the formal definitions for the baseline likelihood-based MIA score functions discussed in the main paper. For a given input sequence $\boldsymbol{x} = \{x_1, \ldots, x_T\}$ of length $T$, these methods compute a score based on the token-level log-probabilities produced by the target model $P$.

**Loss.** (Yeom et al., 2018) The standard Loss-based method, also known as Negative Log-Likelihood (NLL), uses the average log-probability of a sequence as its score. A higher score (lower loss) is indicative of membership. The score is defined as:

$$S_{\text{Loss}}(\boldsymbol{x}) = \frac{1}{T} \sum_{t=1}^{T} \log P(x_t|x_{<t}) \tag{10}$$

**Ref.** (Carlini et al., 2021) The Reference-based method (Ref) calibrates the target model's likelihood by subtracting the log-likelihood from a smaller reference model ($P_{\text{ref}}$). This helps to normalize for tokens that are inherently common or easy to predict. The score is:

$$S_{\text{Ref}}(\boldsymbol{x}) = \frac{1}{T} \sum_{t=1}^{T} \left( \log P(x_t|x_{<t}) - \log P_{\text{ref}}(x_t|x_{<t}) \right) \tag{11}$$

**Min-$k$%.** (Shi et al., 2024) The Min-$k$% method operates on the assumption that member samples have fewer "outlier" tokens with very low probabilities. It computes the score by averaging only the lowest $k$% log-probabilities in the sequence. Let $\mathcal{S}_k$ be the set of token positions corresponding to the lowest $k$% log-probabilities. The score is:

$$S_{\text{Min-}k\%}(\boldsymbol{x}) = \frac{1}{|\mathcal{S}_k|} \sum_{t \in \mathcal{S}_k} \log P(x_t|x_{<t}) \tag{12}$$

**Min-$k$%++.** (Zhang et al., 2025b) The Min-$k$%++ method enhances Min-$k$% by first normalizing the log-probability at each position $t$ using pre-computed mean ($\mu_t$) and standard deviation ($\sigma_t$) statistics for that position. This accounts for positional biases in the model's predictions. Let the normalized score be $z_t = (\log P(x_t|x_{<t}) - \mu_t)/\sigma_t$. Let $\mathcal{S}_k$ be the set of positions corresponding to the lowest $k$% normalized scores $z_t$. The final score is:

$$S_{\text{Min-}k\%++}(\boldsymbol{x}) = \frac{1}{|\mathcal{S}_k|} \sum_{t \in \mathcal{S}_k} z_t \tag{13}$$

**FSD.** (Zhang et al., 2025a) Finetuning-based Score Difference (FSD) is a framework that enhances any base scoring function $S(\cdot)$. It computes the difference between the score from the original model ($M$) and the score from a model fine-tuned on non-member data ($M'$). A larger difference suggests membership.

$$S_{\text{FSD}}(\boldsymbol{x}) = S(\boldsymbol{x}; M) - S(\boldsymbol{x}; M') \tag{14}$$

# B    VISUALIZATION OF WEIGHT DECAY FUNCTIONS

In this section, we visualize weight decay functions (Linear, Exponential, or Polynomial), with varying $\alpha$. We use the following ranges:

- For **Linear** decay, the range for the coefficient $\alpha$ is: $\{0.1, 0.3, 0.5, 0.7, 1.0\}$
- For **Exponential** decay, the range for the coefficient $\alpha$ is: $\{0.002, 0.004, 0.006, 0.008, 0.01, 0.02, 0.04, 0.06, 0.08, 0.1\}$.
- For **Polynomial** decay, the range for the coefficient $\alpha$ is: $\{0.1, 0.3, 0.5, 0.7, 1.0, 1.2, 1.5, 1.8, 2.0\}$.

Fig. 6 visualizes how the decay function becomes steeper as the hyperparameter ($\alpha$) increases.

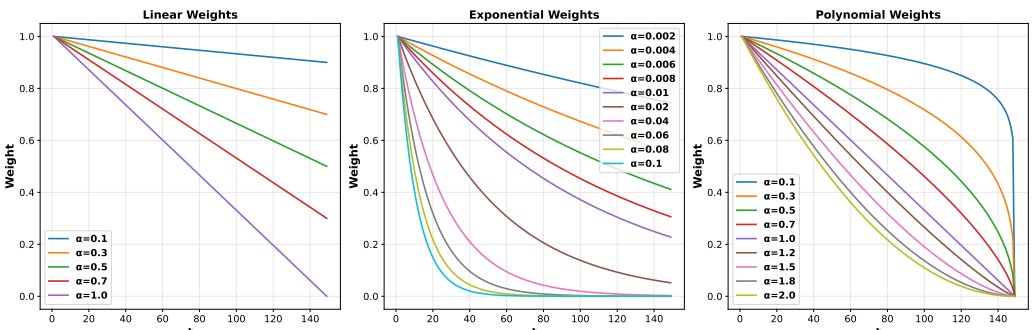

Figure 6: Visualization of different positional weight functions (Linear, Exponential, Polynomial) with various hyperparameters. The x-axis represents the token position in a sequence of length 150, while the y-axis shows the corresponding weight applied to that position.

## C  LIKELIHOOD-BASED SCORE FUNCTIONS WITH PDR

This section details how our Position Difference Reweighting (PDR) method is integrated with various likelihood-based MIA score functions. For a given input sequence $\boldsymbol{x} = \{x_1, \ldots, x_T\}$ of length $T$, PDR introduces a positional weight $w(t)$ for each token $x_t$ at position $t$. The final score is then computed based on the weighted combination of token-level scores.

**PDR-Loss.** The standard Loss-based method, often conceptualized as Negative Log-Likelihood (NLL), uses the average log-probability of a sequence as its score. With PDR, we apply positional weights to the log-probabilities of each token before averaging. A lower weighted loss (which corresponds to a higher score) suggests the sequence is a member. The PDR-enhanced score is:

$$S_{\text{PDR-Loss}}(\boldsymbol{x}) = \frac{1}{T} \sum_{t=1}^{T} w(t) \cdot \log P(x_t | x_{<t}) \tag{15}$$

**PDR-Ref.** The Reference-based method (Ref) calibrates the target model's likelihood by subtracting the log-likelihood from a smaller reference model ($P_{\text{ref}}$). PDR is applied to the resulting difference at each position. The score is defined as:

$$S_{\text{PDR-Ref}}(\boldsymbol{x}) = \frac{1}{T} \sum_{t=1}^{T} w(t) \cdot (\log P(x_t | x_{<t}) - \log P_{\text{ref}}(x_t | x_{<t})) \tag{16}$$

**PDR-Min-$k$%.** Following the standard Min-$k$% procedure, we first identifies the token positions corresponding to the lowest $k$% log-probabilities. Then computes the final score by taking a weighted average of the log-probabilities at only these selected positions, where each score is multiplied by its corresponding positional weight $w(t)$. Let $\mathcal{S}_k$ be the set of token positions corresponding to the lowest $k$% values of $\{\log P(x_t | x_{<t})\}_{t=1}^{T}$. The score is:

$$S_{\text{PDR-Min-}k\%}(\boldsymbol{x}) = \frac{1}{|\mathcal{S}_k|} \sum_{t \in \mathcal{S}_k} w(t) \cdot \log P(x_t | x_{<t}) \tag{17}$$

**PDR-Min-$k$%++.** Similarly, for Min-$k$%++, we first identify the positions of the lowest $k$% normalized z-scores. The PDR-enhanced score is then the weighted average of these selected z-scores, with positional weights applied before averaging. Let $z_t = (\log P(x_t | x_{<t}) - \mu_t)/\sigma_t$, and let $\mathcal{S}_k$ be the set of positions for the lowest $k$% values of $\{z_t\}_{t=1}^{T}$. The score is:

$$S_{\text{PDR-Min-}k\%++}(\boldsymbol{x}) = \frac{1}{|\mathcal{S}_k|} \sum_{t \in \mathcal{S}_k} w(t) \cdot z_t \tag{18}$$

**PDR-FSD.** Finetuning-based Score Difference (FSD) calculates the difference between a score function $S(\cdot)$ computed before and after fine-tuning the model on non-member data. Our PDR

method can be applied to the base score function $S(\cdot)$ used within the FSD framework. If we denote the fine-tuned model as $M'$, the FSD score using a PDR-enhanced base method $S_{+\text{PDR}}$ is:

$$S_{\text{PDR-FSD}}(\boldsymbol{x}) = S_{+\text{PDR}}(\boldsymbol{x}; M) - S_{+\text{PDR}}(\boldsymbol{x}; M') \tag{19}$$

where $S_{+\text{PDR}}(\boldsymbol{x}; M)$ and $S_{+\text{PDR}}(\boldsymbol{x}; M')$ are the PDR-enhanced scores computed using the original model $M$ and the fine-tuned model $M'$, respectively.

The detailed process of applying our PDR method to various likelihood-based MIA methods is outlined in Algorithm 1. This algorithm specifically illustrates the computation for combining PDR with Loss, Ref, Min-$k$%, and Min-$k$%++.

---

**Algorithm 1** Overall algorithm for applying PDR to different logit-based MIA methods

---

1: **Input:**
    Test dataset $\mathcal{D} = \{\boldsymbol{x}^1, \ldots, \boldsymbol{x}^N\}$;
    Target model's predictive distribution $P(x_t|x_{<t})$;
    A chosen decay function $f_{\text{decay}} \in \{\text{Linear}, \text{Exponential}, \text{Polynomial}\}$;
    Decay hyperparameter $\alpha$ or $p$;
    A base MIA scoring method $\mathcal{M} \in \{\text{Loss}, \text{Ref}, \text{Min-}K\%, \text{Min-}K\%\text{++}\}$;
    *(Optional)* Reference model $P_{\text{ref}}(x_t|x_{<t})$;
    *(Optional)* Positional normalization stats $\{\mu_t, \sigma_t\}_{t=1}^T$.
2: **Output:** A list of PDR-enhanced scores $\mathcal{S}_{\text{PDR}} = \{s_1, \ldots, s_N\}$.
3: Initialize an empty list $\mathcal{S}_{\text{PDR}}$.
4: **for** $i = 1$ to $N$ **do**               ▷ Iterate over all sequences in the dataset
5:     $\boldsymbol{x}^i = \{\boldsymbol{x}_1^i, \ldots, \boldsymbol{x}_T^i\}$
    *// Step 1: Compute Positional Weights for the current sequence*
6:     **for** $t = 1$ to $T$ **do**           ▷ Iterate over all positions in the sequence
7:         **if** $f_{\text{decay}}$ is Linear **then**
8:             $w(t) \leftarrow 1 - \alpha \cdot \frac{t-1}{T-1}$
9:         **else if** $f_{\text{decay}}$ is Exponential **then**
10:            $w(t) \leftarrow \exp(-\alpha \cdot (t-1))$
11:         **else if** $f_{\text{decay}}$ is Polynomial **then**
12:            $w(t) \leftarrow \left(1 - \frac{t-1}{T-1}\right)^p$
13:         **end if**
14:     **end for**
    *// Step 2: Apply PDR to the chosen base MIA method*
15:     Initialize current score $s^i \leftarrow 0$.
16:     **if** $\mathcal{M}$ is Loss **then**
17:         $s^i \leftarrow -\frac{1}{T} \sum_{t=1}^{T} w(t) \cdot \log P(x_t^i|x_{<t}^i)$
18:     **else if** $\mathcal{M}$ is Ref **then**
19:         $s^i \leftarrow \frac{1}{T} \sum_{t=1}^{T} w(t) \cdot \left(\log P(x_t^i|x_{<t}^i) - \log P_{\text{ref}}(x_t^i|x_{<t}^i)\right)$
20:     **else if** $\mathcal{M}$ is Min-$k$% **then**
21:         Let $\mathcal{S}_k$ be the set of token positions with the smallest $k$% log-probabilities.
22:         $s^i \leftarrow \frac{1}{|\mathcal{S}_k|} \sum_{t \in \mathcal{S}_k} w(t) \cdot \log P(x_t^i|x_{<t}^i)$
23:     **else if** $\mathcal{M}$ is Min-$k$%++ **then**
24:         For each $t$, compute normalized score $z_t = \frac{\log P(x_t^i|x_{<t}^i) - \mu_t}{\sigma_t}$.
25:         Let $\mathcal{S}_k$ be the set of token positions with the smallest $k$% normalized scores $z_t$.
26:         $s^i \leftarrow \frac{1}{|\mathcal{S}_k|} \sum_{t \in \mathcal{S}_k} w(t) \cdot z_t$
27:     **end if**
28:     Append $s^i$ to $\mathcal{S}_{\text{PDR}}$.
29: **end for**
30: **return** $\mathcal{S}_{\text{PDR}}$

---

# D   EXPERIMENTS SETTING DETAILS

## D.1   BENCHMARKS

We focus on two commonly-used benchmarks for pre-training data detection. (1) **WikiMIA** (Shi et al., 2024) is the first benchmark for pre-training data detection, comprising texts from Wikipedia events. The distinction between training and non-training data is established based on temporal timestamps. To enable fine-grained evaluation, WikiMIA organizes data into splits according to sentence length. It also includes two evaluation settings: the *original* setting evaluates the detection of verbatim training texts, while the *paraphrased* setting uses ChatGPT to paraphrase training texts and evaluates on paraphrased inputs. (2) **MIMIR** (Duan et al., 2024) is built upon the Pile dataset (Gao et al., 2020). This benchmark poses greater challenges compared to WikiMIA, as the shared dataset origin between training and non-training texts eliminates substantial distribution shifts and temporal discrepancies (Duan et al., 2024).

## D.2   BASELINES

We consider several representative methods as our baselines:

- **Loss** (Yeom et al., 2018) is a general technique that directly uses the loss of the model as the detection score.

- **Ref** (Carlini et al., 2021) employs an additional, typically smaller, language model as a reference to calibrate the likelihood of the input text.

- **zlib** and **lowercase** (Carlini et al., 2021) use the compression entropy of zlib and the likelihood of the lowercase text as references to calibrate the likelihood.

- **Min-$k$%** (Shi et al., 2024) examines the exact probabilities of the token and averages a subset of the lowest token scores from the input sequence.

- **Min-$k$%++** (Zhang et al., 2025b) extends Min-$k$% by standardizing the log-probability of each token using the mean and standard deviation of log-probabilities at that specific position, making scores more comparable across different positions before applying the Min-$k$% selection.

- **FSD** (Zhang et al., 2025a) involves fine-tuning the model on non-member samples and using the difference in logit-based scores before and after fine-tuning for detection.

## D.3   ENVIRONMENT

All experiments were conducted on the Ubuntu 20.04.4 LTS operating system, Intel(R) Xeon(R) Gold 5220 CPU @ 2.20GHz with a single NVIDIA A40 48GB GPU and 512GB of RAM. The framework is implemented with Python 3.9.0 and PyTorch 2.6.0. Other key packages include transformer 4.40.1, numpy 1.24 and accelerate 0.26.0.

## D.4   MODELS

This section details the specific models used in our experiments. For the Ref method, the choice of reference model depends on the dataset following (Carlini et al., 2021; Shi et al., 2024; Zhang et al., 2025b).On the **WikiMIA** dataset, we used the following reference models for different model families:

- For the **Pythia** family, we used `Pythia-70M`.

- For the **Llama** family, we used `Llama-7B`.

- For the **GPT** family, we used `GPT-Neo-125M`.

- For the **Mamba** family, we used `Mamba-130M`.

- For the **OPT** family, we used `OPT-350M`.

### D.5 HYPERPARAMETER

For Min-$k$% and Min-$k$%++, we consistently use $k = 20\%$ following common practice. In our experiments, settings are as follows:

- **LPDR**: on the WikiMIA dataset, we generally set $\alpha = 1.0$ for sequence lengths of 64 and 128. For the shorter length of 32, a smaller weight was preferred like 0.1 or 0.5. On the Mimir dataset, we set $\alpha = 1.0$.

- **EPDR**: on the WikiMIA dataset, Ref and Min-$k$%++ set $\alpha = 0.02$, where as Loss and Min-$k$% set $\alpha = 0.002$. On Mimir, $\alpha = 0.002$ was used.

- **PPDR**: on the WikiMIA dataset, we used $p = 0.1$ for length 32, but a much steeper decay of $p = 2.0$ for lengths 64 and 128. On Mimir, we set a gentler $p = 0.1$.

A clear trend emerges from these results. On the WikiMIA dataset, longer sequences tend to benefit from more aggressive, steeper weight functions, while shorter sequences and the more challenging Mimir dataset favor gentler, more gradual decay.

### D.6 FSD SETTINGS

For the Finetuning-based Score Difference (FSD) experiments, we follow the implementation details from the original paper. To construct the non-member dataset for fine-tuning, we first randomly sample 30% of the entire dataset. All non-member samples within this subset are then used as the fine-tuning dataset. The remaining 70% of the data is reserved for testing. We use LoRA (Hu et al., 2022a) to fine-tune the base model for 3 epochs with a batch size of 8. The initial learning rate is set to 0.001 and is adjusted using a cosine scheduling strategy.

## E WIKIMIA RESULTS

### E.1 WIKIMIA RESULTS ON FIXED $\alpha$

This section presents the comprehensive results on the WikiMIA benchmark. We report detailed AU-ROC and TPR scores across all models and sequence lengths under both the original and paraphrased settings. The AUROC results are shown in Tab. 4 and Tab. 5, respectively, while the corresponding TPR results are reported in Tab. 6 and Tab. 7.

Table 4: AUC-ROC on WikiMIA benchmark under original setting. w/ LPDR utilizes linear weights for reweighting, w/ EPDR utilizes exponential weights for reweighting, w/ PPDR utilizes polynomial weights for reweighting.[†]Neighbor results are from Zhang et al. (2025b).

| Length | Models | Lowercase | Zlib | †Neighbor | Loss | w/ LPDR | w/ EPDR | w/ PPDR | Ref | w/ LPDR | w/ EPDR | w/ PPDR | Min-$k$% | w/ LPDR | w/ EPDR | w/ PPDR | Min-$k$% ++ | w/ LPDR | w/ EPDR | w/ PPDR |
|---|---|---|---|---|---|---|---|---|---|---|---|---|---|---|---|---|---|---|---|---|
| | Mamba-1.4B | 60.9 | 61.9 | 64.1 | 61.0 | 61.5 | 60.9 | 61.2 | 62.2 | 62.2 | 62.7 | 62.2 | 63.3 | 63.5 | 63.2 | 63.8 | 66.4 | 67.4 | 66.6 | 67.1 |
| | Mamba-2.8B | 63.6 | 64.7 | 67.0 | 64.1 | 64.5 | 64.0 | 64.2 | 67.0 | 66.9 | 67.1 | 66.7 | 66.1 | 66.2 | 66.0 | 66.1 | 69.0 | 69.4 | 68.9 | 69.1 |
| | Pythia-2.8B | 60.9 | 62.1 | 64.2 | 61.4 | 61.7 | 61.2 | 61.4 | 61.3 | 61.3 | 62.4 | 61.0 | 61.7 | 61.9 | 61.6 | 61.9 | 64.0 | 64.7 | 64.2 | 64.4 |
| | Pythia-6.9B | 62.2 | 64.4 | 65.8 | 63.8 | 64.0 | 63.7 | 63.8 | 63.6 | 63.5 | 64.6 | 63.2 | 66.3 | 66.3 | 66.1 | 65.4 | 70.3 | 70.8 | 70.1 | 70.5 |
| 32 | Pythia-12B | 64.8 | 65.8 | 66.6 | 65.4 | 65.4 | 65.2 | 65.4 | 65.1 | 65.1 | 66.1 | 64.7 | 68.1 | 68.0 | 67.8 | 67.4 | 72.2 | 72.3 | 71.3 | 72.0 |
| | Llama-13B | 64.0 | 67.8 | 65.8 | 67.5 | 67.7 | 67.5 | 67.5 | 57.9 | 57.8 | 57.2 | 57.6 | 66.8 | 66.8 | 66.8 | 66.6 | 84.4 | 85.9 | 86.2 | 85.0 |
| | Llama-30B | 64.1 | 69.8 | 67.6 | 69.4 | 69.6 | 69.4 | 69.5 | 63.5 | 63.5 | 62.8 | 63.2 | 69.3 | 69.4 | 69.3 | 69.2 | 84.4 | 85.4 | 85.5 | 84.6 |
| | OPT-66B | 62.8 | 65.8 | 68.2 | 65.6 | 65.6 | 65.5 | 65.7 | 68.6 | 68.6 | 69.3 | 68.4 | 67.5 | 67.7 | 67.4 | 67.5 | 69.7 | 70.2 | 69.4 | 70.0 |
| | GPT-NeoX-20B | 68.3 | 69.3 | 70.2 | 68.9 | 68.9 | 68.9 | 69.0 | 67.6 | 67.4 | 66.7 | 67.0 | 72.2 | 72.0 | 71.8 | 71.1 | 75.1 | 75.2 | 74.4 | 75.0 |
| | Average | 63.5 | 65.7 | 66.6 | 65.3 | **65.4** | 65.1 | 65.3 | 64.1 | 64.0 | 64.3 | 63.8 | 66.8 | **66.9** | 66.7 | 66.5 | 72.8 | **73.5** | 72.9 | 73.1 |
| | Mamba-1.4B | 57.0 | 60.4 | 60.6 | 58.2 | 59.7 | 58.2 | 60.7 | 60.6 | 61.1 | 61.9 | 60.3 | 61.7 | 62.9 | 62.0 | 62.8 | 67.2 | 68.2 | 67.9 | 68.2 |
| | Mamba-2.8B | 61.7 | 63.0 | 63.6 | 61.2 | 63.0 | 61.2 | 63.6 | 64.3 | 66.1 | 66.6 | 65.5 | 65.1 | 66.2 | 65.4 | 65.3 | 70.6 | 70.4 | 70.0 | 69.6 |
| | Pythia-2.8B | 57.8 | 60.6 | 61.3 | 58.4 | 60.1 | 58.4 | 60.8 | 59.6 | 60.5 | 62.2 | 60.0 | 61.2 | 63.3 | 61.4 | 62.8 | 64.8 | 65.9 | 65.1 | 65.6 |
| | Pythia-6.9B | 58.2 | 62.6 | 63.2 | 60.7 | 62.6 | 60.6 | 63.0 | 62.4 | 63.3 | 65.0 | 62.7 | 65.0 | 66.7 | 65.2 | 65.3 | 71.6 | 72.1 | 71.2 | 71.2 |
| 64 | Pythia-12B | 59.6 | 63.5 | 62.6 | 61.9 | 63.5 | 61.8 | 63.9 | 63.0 | 64.0 | 65.9 | 63.4 | 66.5 | 67.9 | 66.7 | 67.2 | 72.6 | 72.8 | 71.8 | 71.5 |
| | Llama-13B | 62.0 | 65.3 | 64.1 | 63.6 | 65.0 | 63.7 | 65.7 | 63.4 | 59.8 | 60.1 | 57.0 | 66.0 | 67.4 | 66.1 | 66.7 | 84.3 | 87.2 | 87.0 | 87.4 |
| | Llama-30B | 61.9 | 67.5 | 67.1 | 66.1 | 67.5 | 66.2 | 67.9 | 68.9 | 65.5 | 65.4 | 63.2 | 68.4 | 69.6 | 68.6 | 68.9 | 84.3 | 87.7 | 87.0 | 87.4 |
| | OPT-66B | 61.1 | 63.9 | 64.1 | 62.3 | 64.2 | 62.2 | 64.4 | 66.9 | 68.2 | 69.0 | 68.0 | 66.5 | 68.1 | 66.7 | 66.8 | 69.8 | 70.1 | 69.5 | 69.0 |
| | GPT-NeoX-20B | 66.3 | 68.1 | 67.1 | 66.6 | 67.6 | 66.5 | 67.6 | 66.0 | 66.8 | 67.1 | 65.4 | 72.2 | 70.8 | 72.2 | 68.6 | 76.5 | 76.4 | 76.0 | 75.0 |
| | Average | 60.6 | 63.9 | 63.7 | 62.1 | **63.7** | 62.1 | **64.2** | 63.9 | 63.9 | 64.8 | 62.8 | 65.8 | **67.0** | 66.0 | 66.0 | 73.5 | **74.5** | 73.9 | 73.9 |
| | Mamba-1.4B | 58.5 | 65.6 | 64.8 | 63.3 | 63.6 | 62.9 | 64.2 | 62.0 | 64.1 | 66.7 | 65.4 | 66.8 | 65.5 | 66.9 | 64.8 | 67.7 | 70.2 | 69.8 | 70.5 |
| | Mamba-2.8B | 62.4 | 68.5 | 67.7 | 66.3 | 66.7 | 65.9 | 66.9 | 66.9 | 69.4 | 71.7 | 70.2 | 70.3 | 69.3 | 70.4 | 68.0 | 71.9 | 73.4 | 71.1 | 72.3 |
| | Pythia-2.8B | 59.5 | 65.0 | 65.2 | 62.8 | 63.1 | 62.6 | 63.3 | 59.6 | 61.4 | 63.5 | 62.2 | 66.9 | 64.3 | 66.6 | 63.2 | 66.3 | 66.8 | 66.4 | 66.9 |
| | Pythia-6.9B | 60.5 | 67.6 | 67.5 | 65.1 | 65.6 | 64.8 | 66.1 | 63.3 | 65.1 | 67.5 | 65.7 | 69.5 | 67.8 | 69.2 | 66.7 | 72.4 | 71.2 | 72.7 | |
| 128 | Pythia-12B | 61.4 | 67.8 | 67.1 | 65.8 | 66.2 | 65.6 | 66.7 | 63.9 | 65.1 | 67.2 | 66.0 | 70.7 | 70.0 | 70.5 | 68.9 | 71.8 | 73.5 | 71.9 | 73.5 |
| | Llama-13B | 60.6 | 69.7 | 68.3 | 67.8 | 68.7 | 67.8 | 69.1 | 62.6 | 64.9 | 62.2 | 64.1 | 71.5 | 71.2 | 72.1 | 70.9 | 83.8 | 88.4 | 89.1 | 89.1 |
| | Llama-30B | 59.0 | 71.8 | 72.2 | 70.3 | 71.0 | 70.3 | 71.2 | 71.9 | 70.4 | 66.0 | 68.0 | 73.7 | 72.5 | 73.8 | 71.6 | 82.7 | 85.9 | 87.4 | 87.0 |
| | OPT-66B | 58.9 | 67.3 | 67.7 | 65.5 | 66.7 | 65.6 | 67.6 | 66.9 | 68.6 | 70.3 | 69.0 | 70.6 | 70.6 | 71.3 | 70.3 | 71.1 | 72.9 | 71.9 | 72.4 |
| | GPT-NeoX-20B | 68.0 | 72.3 | 71.6 | 70.7 | 70.7 | 70.4 | 71.0 | 68.3 | 69.5 | 69.3 | 69.6 | 75.6 | 74.5 | 75.7 | 72.6 | 75.4 | 75.7 | 73.9 | 75.1 |
| | Average | 61.0 | 68.4 | 68.0 | 66.4 | **66.9** | 66.2 | **67.4** | 65.0 | 66.5 | 67.2 | 66.7 | 70.6 | 69.5 | **70.7** | 68.5 | 73.4 | **75.5** | 74.7 | **75.5** |

Table 5: AUC-ROC on WikiMIA benchmark under paraphrased setting. w/ LPDR utilizes linear weights for reweighting, w/ EPDR utilizes exponential weights for reweighting, w/ PPDR utilizes polynomial weights for reweighting. †Neighbor results are from Zhang et al. (2025b).

| Length | Models | Lowercase | Zlib | †Neighbor | Loss | w/ LPDR | w/ EPDR | w/ PPDR | Ref | w/ LPDR | w/ EPDR | w/ PPDR | Min-k% | w/ LPDR | w/ EPDR | w/ PPDR | Min-k% ++ | w/ LPDR | w/ EPDR | w/ PPDR |
|---|---|---|---|---|---|---|---|---|---|---|---|---|---|---|---|---|---|---|---|---|
| 32 | Mamba-1.4B | 60.6 | 62.3 | 63.6 | 61.3 | 61.8 | 61.3 | 61.5 | 62.3 | 62.3 | 62.7 | 62.4 | 62.9 | 63.1 | 62.9 | 63.4 | 65.7 | 66.3 | 65.3 | 66.2 |
| | Mamba-2.8B | 63.5 | 64.8 | 66.3 | 64.5 | 64.8 | 64.3 | 64.6 | 66.6 | 66.5 | 66.7 | 66.2 | 65.3 | 65.4 | 65.2 | 64.9 | 67.3 | 67.5 | 66.6 | 67.2 |
| | Pythia-2.8B | 60.3 | 62.3 | 64.5 | 61.6 | 61.8 | 61.4 | 61.5 | 61.2 | 61.2 | 62.3 | 60.9 | 60.9 | 61.1 | 60.8 | 60.6 | 61.3 | 61.7 | 61.0 | 61.4 |
| | Pythia-6.9B | 61.7 | 64.2 | 65.5 | 64.1 | 64.2 | 63.9 | 64.1 | 63.5 | 63.5 | 64.4 | 63.1 | 65.1 | 65.1 | 64.9 | 64.4 | 67.6 | 67.7 | 66.6 | 67.7 |
| | Pythia-12B | 64.4 | 65.9 | 66.8 | 65.6 | 65.7 | 65.4 | 65.7 | 64.9 | 64.9 | 66.0 | 64.5 | 67.2 | 67.2 | 66.9 | 66.2 | 69.4 | 69.4 | 68.1 | 69.1 |
| | Llama-13B | 63.2 | 68.3 | 65.0 | 68.0 | 68.2 | 68.0 | 68.2 | 56.2 | 56.1 | 54.9 | 55.9 | 66.2 | 66.2 | 66.3 | 65.8 | 82.7 | 84.1 | 84.3 | 83.3 |
| | Llama-30B | 61.3 | 70.4 | 66.3 | 70.2 | 70.3 | 70.1 | 70.2 | 62.4 | 62.4 | 61.3 | 62.2 | 68.5 | 68.4 | 68.3 | 67.7 | 81.2 | 82.4 | 82.6 | 81.6 |
| | OPT-66B | 62.3 | 65.3 | 66.7 | 65.3 | 65.0 | 65.1 | 65.3 | 68.0 | 68.0 | 68.9 | 68.1 | 65.8 | 65.8 | 65.5 | 64.9 | 67.0 | 67.1 | 66.0 | 66.9 |
| | GPT-NeoX-20B | 66.9 | 68.5 | 68.3 | 68.3 | 68.4 | 68.5 | 66.7 | 66.6 | 66.0 | 66.1 | 69.6 | 69.4 | 69.3 | 68.0 | 69.7 | 69.5 | 68.4 | 69.2 | |
| | Average | 62.7 | 65.8 | 65.9 | 65.5 | 65.6 | 65.3 | 65.5 | 63.5 | 63.5 | 63.7 | 63.3 | 65.7 | 65.8 | 65.6 | 65.1 | 70.2 | 70.6 | 69.9 | 70.3 |
| 64 | Mamba-1.4B | 57.0 | 59.1 | 60.6 | 56.4 | 59.5 | 56.4 | 60.5 | 59.6 | 60.8 | 61.1 | 60.2 | 58.0 | 61.8 | 58.4 | 61.6 | 62.2 | 65.5 | 63.5 | 65.8 |
| | Mamba-2.8B | 62.0 | 61.9 | 63.7 | 59.8 | 62.9 | 59.7 | 63.6 | 64.5 | 66.3 | 66.6 | 65.2 | 62.4 | 65.1 | 62.7 | 64.0 | 64.9 | 67.9 | 66.0 | 67.5 |
| | Pythia-2.8B | 56.1 | 59.0 | 59.6 | 56.5 | 59.3 | 56.5 | 60.3 | 59.2 | 60.8 | 62.0 | 60.1 | 56.7 | 61.6 | 57.0 | 61.3 | 57.7 | 62.0 | 59.8 | 62.1 |
| | Pythia-6.9B | 57.7 | 61.6 | 63.1 | 59.3 | 62.4 | 59.3 | 62.9 | 62.9 | 64.0 | 65.6 | 63.1 | 61.1 | 65.1 | 61.5 | 63.7 | 64.2 | 65.8 | 65.6 | 68.0 |
| | Pythia-12B | 59.2 | 62.1 | 62.8 | 60.0 | 63.0 | 60.0 | 63.6 | 63.2 | 64.5 | 66.1 | 63.7 | 62.5 | 66.1 | 62.7 | 64.8 | 65.1 | 68.2 | 65.5 | 67.5 |
| | Llama-13B | 61.0 | 65.3 | 64.7 | 63.1 | 66.2 | 63.4 | 66.9 | 60.9 | 57.9 | 57.7 | 55.2 | 63.5 | 67.2 | 64.1 | 66.2 | 78.8 | 84.3 | 83.5 | 85.0 |
| | Llama-30B | 59.8 | 67.4 | 66.7 | 65.5 | 68.4 | 65.7 | 69.0 | 65.3 | 62.8 | 62.9 | 60.8 | 64.9 | 67.9 | 65.3 | 66.7 | 74.7 | 81.7 | 80.5 | 82.4 |
| | OPT-66B | 60.0 | 62.2 | 64.6 | 60.3 | 63.1 | 60.3 | 63.5 | 67.8 | 69.1 | 69.8 | 68.9 | 62.5 | 66.0 | 62.8 | 64.9 | 63.3 | 66.6 | 64.3 | 66.4 |
| | GPT-NeoX-20B | 65.6 | 66.5 | 67.4 | 64.4 | 67.2 | 64.4 | 67.2 | 66.0 | 67.2 | 66.9 | 65.4 | 66.1 | 68.7 | 66.4 | 66.5 | 66.2 | 68.2 | 66.2 | 67.5 |
| | Average | 59.8 | 62.8 | 63.7 | 60.6 | 63.5 | 60.6 | 64.2 | 63.3 | 63.7 | 64.3 | 62.5 | 61.9 | 65.5 | 62.3 | 64.4 | 66.3 | 70.3 | 68.3 | 70.3 |
| 128 | Mamba-1.4B | 57.7 | 65.3 | 62.6 | 62.7 | 64.1 | 62.7 | 64.6 | 61.1 | 64.6 | 67.9 | 66.3 | 64.4 | 65.8 | 65.1 | 66.1 | 63.3 | 68.2 | 69.5 | 69.8 |
| | Mamba-2.8B | 61.2 | 68.4 | 64.6 | 65.7 | 67.3 | 65.7 | 67.9 | 66.6 | 69.6 | 72.1 | 70.5 | 68.0 | 69.9 | 69.1 | 68.7 | 68.9 | 71.5 | 70.7 | 71.6 |
| | Pythia-2.8B | 59.6 | 65.0 | 61.9 | 62.3 | 64.0 | 62.5 | 64.1 | 59.5 | 62.6 | 64.4 | 63.8 | 64.7 | 63.5 | 64.9 | 62.7 | 62.7 | 65.8 | 66.0 | 66.8 |
| | Pythia-6.9B | 59.9 | 67.4 | 64.3 | 64.7 | 66.6 | 64.7 | 67.3 | 62.9 | 65.9 | 68.5 | 67.2 | 67.0 | 68.9 | 67.8 | 67.8 | 65.9 | 72.2 | 72.0 | 73.3 |
| | Pythia-12B | 60.4 | 67.9 | 64.3 | 65.4 | 67.0 | 65.4 | 67.6 | 63.9 | 66.2 | 68.5 | 66.9 | 68.5 | 69.1 | 69.3 | 68.0 | 67.7 | 72.2 | 71.9 | 73.0 |
| | Llama-13B | 56.3 | 69.6 | 64.0 | 67.2 | 69.1 | 67.6 | 69.9 | 59.7 | 61.7 | 58.8 | 60.6 | 68.6 | 71.0 | 70.3 | 70.7 | 76.2 | 84.3 | 87.2 | 86.1 |
| | Llama-30B | 55.3 | 71.5 | 67.2 | 69.3 | 71.2 | 69.7 | 72.0 | 69.8 | 68.6 | 64.4 | 66.0 | 70.3 | 72.5 | 71.6 | 71.9 | 73.4 | 80.5 | 83.9 | 82.9 |
| | OPT-66B | 57.6 | 66.9 | 63.4 | 64.5 | 66.9 | 64.8 | 67.9 | 67.0 | 69.5 | 70.9 | 70.0 | 67.2 | 69.9 | 68.4 | 69.5 | 67.0 | 69.5 | 70.0 | 70.4 |
| | GPT-NeoX-20B | 67.6 | 72.0 | 69.6 | 69.7 | 71.4 | 69.6 | 71.9 | 68.4 | 70.2 | 70.0 | 70.6 | 73.0 | 75.2 | 73.8 | 73.5 | 70.6 | 72.6 | 72.3 | 72.9 |
| | Average | 59.5 | 68.2 | 64.7 | 65.7 | 67.5 | 65.9 | 68.1 | 64.3 | 66.5 | 67.3 | 66.9 | 68.0 | 69.5 | 68.9 | 68.8 | 68.4 | 73.0 | 73.7 | 74.1 |

Table 6: TPR on WikiMIA benchmark under original setting. w/ LPDR utilizes linear weights for reweighting, w/ EPDR utilizes exponential weights for reweighting, w/ PPDR utilizes polynomial weights for reweighting. †Neighbor results are from Zhang et al. (2025b).

| Length | Models | Lowercase | Zlib | †Neighbor | Loss | w/ LPDR | w/ EPDR | w/ PPDR | Ref | w/ LPDR | w/ EPDR | w/ PPDR | Min-k% | w/ LPDR | w/ EPDR | w/ PPDR | Min-k% ++ | w/ LPDR | w/ EPDR | w/ PPDR |
|---|---|---|---|---|---|---|---|---|---|---|---|---|---|---|---|---|---|---|---|---|
| 32 | Mamba-1.4B | 11.1 | 15.5 | 11.9 | 14.2 | 15.2 | 14.0 | 14.0 | 7.8 | 7.8 | 8.8 | 7.2 | 14.2 | 14.7 | 15.0 | 13.4 | 11.4 | 11.6 | 14.0 | 11.6 |
| | Mamba-2.8B | 16.8 | 16.3 | 16.0 | 14.7 | 17.6 | 15.8 | 15.2 | 9.8 | 10.1 | 10.9 | 10.1 | 17.3 | 16.3 | 15.0 | 16.5 | 11.4 | 11.4 | 14.0 | 11.4 |
| | Pythia-2.8B | 11.1 | 15.8 | 15.0 | 14.7 | 17.6 | 15.0 | 15.5 | 6.2 | 6.2 | 11.4 | 5.4 | 16.5 | 16.8 | 16.5 | 17.6 | 10.6 | 10.9 | 14.2 | 10.9 |
| | Pythia-6.9B | 10.6 | 16.3 | 16.5 | 14.2 | 15.2 | 14.5 | 13.4 | 6.7 | 6.5 | 12.1 | 5.7 | 17.8 | 18.1 | 18.1 | 18.1 | 14.5 | 15.2 | 17.3 | 15.2 |
| | Pythia-12B | 16.3 | 17.1 | 19.4 | 17.1 | 17.8 | 17.6 | 15.5 | 9.0 | 8.8 | 11.1 | 9.8 | 23.0 | 22.7 | 23.3 | 23.8 | 16.5 | 17.3 | 19.9 | 15.8 |
| | Llama-13B | 9.6 | 11.6 | 11.6 | 14.0 | 14.2 | 14.0 | 14.7 | 4.7 | 4.9 | 5.2 | 4.9 | 18.9 | 19.9 | 20.2 | 21.2 | 33.1 | 40.1 | 43.4 | 38.2 |
| | Llama-30B | 11.4 | 14.5 | 9.3 | 18.3 | 17.8 | 18.3 | 18.9 | 10.1 | 10.9 | 7.2 | 9.3 | 22.0 | 22.7 | 23.0 | 21.2 | 31.8 | 35.7 | 37.0 | 29.5 |
| | OPT-66B | 11.4 | 16.5 | 21.7 | 14.2 | 15.2 | 15.2 | 16.0 | 11.1 | 11.1 | 10.6 | 10.1 | 21.7 | 21.7 | 20.9 | 20.2 | 11.9 | 14.0 | 15.2 | 14.7 |
| | GPT-NeoX-20B | 16.8 | 20.4 | 22.2 | 20.4 | 22.7 | 20.2 | 21.2 | 15.5 | 16.5 | 17.6 | 15.2 | 28.9 | 30.0 | 28.2 | 26.6 | 19.1 | 19.9 | 20.4 | 21.2 |
| | Average | 12.8 | 16.0 | 16.0 | 15.8 | 17.1 | 16.0 | 16.0 | 9.0 | 9.2 | 10.5 | 8.6 | 20.0 | 20.4 | 20.0 | 19.8 | 17.1 | 19.6 | 21.7 | 18.7 |
| 64 | Mamba-1.4B | 8.8 | 14.1 | 8.8 | 9.5 | 13.0 | 11.3 | 17.3 | 4.6 | 6.7 | 7.0 | 4.2 | 15.8 | 15.5 | 17.3 | 17.3 | 13.7 | 12.7 | 12.3 | 9.9 |
| | Mamba-2.8B | 16.5 | 14.8 | 10.6 | 10.2 | 15.8 | 12.7 | 16.2 | 9.2 | 9.9 | 10.9 | 9.9 | 19.0 | 19.7 | 19.0 | 21.5 | 18.7 | 14.4 | 16.5 | 12.7 |
| | Pythia-2.8B | 10.2 | 14.4 | 10.2 | 10.2 | 14.1 | 10.9 | 16.2 | 10.6 | 5.6 | 10.9 | 3.5 | 18.3 | 22.2 | 16.9 | 21.8 | 13.4 | 14.1 | 14.4 | 12.3 |
| | Pythia-6.9B | 11.6 | 16.2 | 10.9 | 13.4 | 13.0 | 12.3 | 15.5 | 12.0 | 6.3 | 12.0 | 4.9 | 19.0 | 18.7 | 17.3 | 16.9 | 16.2 | 20.1 | 19.7 | 18.7 |
| | Pythia-12B | 12.3 | 11.3 | 11.3 | 9.2 | 13.7 | 8.8 | 16.2 | 13.0 | 7.4 | 10.9 | 8.1 | 21.5 | 19.0 | 21.1 | 23.2 | 16.9 | 22.2 | 23.2 | 19.4 |
| | Llama-13B | 11.6 | 12.7 | 10.2 | 11.3 | 16.5 | 10.6 | 14.4 | 4.2 | 4.9 | 6.0 | 6.7 | 17.3 | 21.1 | 15.8 | 21.5 | 31.3 | 47.9 | 52.5 | 44.7 |
| | Llama-30B | 9.9 | 15.5 | 9.9 | 13.7 | 16.9 | 14.1 | 18.7 | 10.6 | 8.8 | 7.0 | 8.1 | 16.5 | 20.8 | 18.7 | 18.7 | 33.8 | 43.7 | 46.1 | 45.1 |
| | OPT-66B | 10.9 | 13.7 | 12.0 | 13.4 | 13.7 | 13.4 | 16.9 | 13.4 | 9.5 | 12.3 | 7.4 | 21.8 | 22.2 | 22.9 | 18.3 | 19.4 | 16.5 | 18.7 | 18.3 |
| | GPT-NeoX-20B | 16.2 | 19.4 | 13.0 | 12.3 | 15.1 | 13.0 | 21.5 | 14.8 | 13.7 | 16.2 | 12.7 | 19.0 | 26.4 | 19.7 | 18.7 | 22.5 | 20.8 | 21.5 | 21.1 |
| | Average | 12.0 | 14.7 | 10.8 | 11.5 | 14.7 | 11.9 | 17.0 | 10.3 | 8.1 | 10.4 | 7.3 | 18.7 | 20.6 | 18.7 | 19.8 | 20.7 | 25.0 | 25.0 | 22.5 |
| 128 | Mamba-1.4B | 13.0 | 19.4 | 15.8 | 11.5 | 18.0 | 13.0 | 11.5 | 10.1 | 9.4 | 15.8 | 12.2 | 9.4 | 19.4 | 11.5 | 21.6 | 10.1 | 13.7 | 12.2 | 11.5 |
| | Mamba-2.8B | 13.7 | 23.7 | 15.1 | 19.4 | 18.0 | 21.6 | 16.5 | 10.1 | 15.8 | 14.4 | 16.5 | 20.1 | 33.8 | 20.1 | 23.0 | 19.4 | 26.6 | 18.7 | 18.0 |
| | Pythia-2.8B | 10.8 | 18.7 | 8.6 | 9.4 | 13.7 | 11.5 | 14.4 | 10.1 | 10.1 | 15.8 | 7.9 | 13.7 | 18.0 | 18.0 | 15.8 | 14.4 | 17.3 | 13.7 | 16.5 |
| | Pythia-6.9B | 13.0 | 20.9 | 10.8 | 14.4 | 15.1 | 15.8 | 15.8 | 13.7 | 15.1 | 17.3 | 13.7 | 18.0 | 28.8 | 21.6 | 21.6 | 20.1 | 23.7 | 18.0 | 20.1 |
| | Pythia-12B | 13.0 | 23.7 | 10.1 | 18.0 | 15.1 | 18.0 | 13.0 | 12.2 | 11.5 | 12.2 | 11.5 | 20.1 | 25.9 | 22.3 | 23.0 | 18.0 | 26.6 | 25.9 | 25.9 |
| | Llama-13B | 15.8 | 18.7 | 12.9 | 21.6 | 19.4 | 20.9 | 15.8 | 10.8 | 5.4 | 5.0 | 5.8 | 20.1 | 28.8 | 22.3 | 28.1 | 38.1 | 56.8 | 57.6 | 54.7 |
| | Llama-30B | 10.1 | 18.0 | 15.1 | 23.7 | 18.0 | 21.6 | 18.7 | 10.8 | 12.2 | 7.9 | 13.7 | 22.3 | 26.6 | 23.0 | 28.1 | 22.3 | 36.0 | 57.6 | 46.8 |
| | OPT-66B | 15.1 | 21.6 | 12.9 | 20.9 | 16.5 | 22.3 | 20.1 | 15.8 | 20.1 | 15.8 | 13.0 | 21.6 | 28.1 | 23.7 | 29.5 | 16.5 | 18.0 | 19.4 | 18.0 |
| | GPT-NeoX-20B | 13.0 | 22.3 | 15.8 | 18.0 | 15.1 | 16.5 | 15.8 | 16.5 | 17.3 | 15.8 | 17.3 | 21.6 | 30.2 | 22.3 | 28.8 | 24.5 | 21.6 | 26.6 | 18.7 |
| | Average | 13.0 | 20.8 | 13.0 | 17.4 | 16.5 | 17.9 | 15.7 | 12.2 | 13.0 | 13.3 | 12.4 | 18.5 | 26.6 | 20.5 | 24.4 | 20.4 | 26.7 | 27.7 | 25.6 |

Table 7: TPR on WikiMIA benchmark under paraphrased setting. w/ LPDR utilizes linear weights for reweighting, w/ EPDR utilizes exponential weights for reweighting, w/ PPDR utilizes polynomial weights for reweighting. †Neighbor results are from Zhang et al. (2025b).

| Length | Models | Lowercase | Zlib | †Neighbor | Loss | w/ LPDR | w/ EPDR | w/ PPDR | Ref | w/ LPDR | w/ EPDR | w/ PPDR | Min-k% | w/ LPDR | w/ EPDR | w/ PPDR | Min-k% ++ | w/ LPDR | w/ EPDR | w/ PPDR |
|---|---|---|---|---|---|---|---|---|---|---|---|---|---|---|---|---|---|---|---|---|
| 32 | Mamba-1.4B | 13.2 | 13.2 | 7.2 | 14.2 | 16.0 | 12.9 | 15.5 | 5.9 | 5.9 | 9.6 | 6.2 | 11.9 | 14.0 | 12.4 | 15.0 | 7.8 | 11.4 | 11.6 | 9.8 |
| | Mamba-2.8B | 15.0 | 12.7 | 9.3 | 16.5 | 17.8 | 16.3 | 17.3 | 10.1 | 9.3 | 11.9 | 10.6 | 19.9 | 20.2 | 20.4 | 15.2 | 11.1 | 12.9 | 11.1 | 9.8 |
| | Pythia-2.8B | 11.6 | 14.5 | 8.5 | 14.2 | 15.5 | 15.0 | 14.2 | 7.2 | 7.2 | 12.9 | 7.5 | 16.3 | 16.8 | 15.2 | 15.8 | 10.9 | 10.6 | 10.9 | 11.4 |
| | Pythia-6.9B | 11.9 | 12.7 | 9.6 | 15.0 | 14.0 | 14.2 | 14.5 | 6.2 | 5.7 | 12.9 | 5.7 | 21.7 | 22.0 | 22.7 | 18.1 | 14.5 | 16.5 | 15.5 | 15.0 |
| | Pythia-12B | 16.5 | 15.5 | 9.8 | 17.3 | 16.8 | 17.8 | 18.9 | 8.0 | 8.3 | 10.3 | 8.3 | 19.9 | 20.4 | 20.7 | 18.3 | 15.5 | 13.4 | 14.0 | 13.4 |
| | Llama-13B | 9.6 | 15.0 | 8.5 | 16.3 | 16.5 | 16.0 | 16.8 | 5.4 | 5.9 | 4.4 | 6.2 | 14.2 | 15.2 | 14.5 | 15.5 | 33.9 | 37.7 | 33.9 | 35.9 |
| | Llama-30B | 13.2 | 15.2 | 9.3 | 14.7 | 17.6 | 15.5 | 16.8 | 8.3 | 7.5 | 5.9 | 7.8 | 18.1 | 17.8 | 17.8 | 19.4 | 25.8 | 29.2 | 31.8 | 24.8 |
| | OPT-66B | 12.9 | 16.8 | 12.1 | 15.2 | 17.6 | 15.5 | 15.5 | 9.3 | 9.6 | 10.9 | 8.5 | 16.8 | 18.3 | 18.1 | 14.7 | 16.3 | 16.3 | 17.1 | 13.4 |
| | GPT-NeoX-20B | 14.5 | 19.6 | 15.2 | 17.6 | 19.4 | 18.9 | 19.9 | 15.2 | 15.0 | 14.2 | 14.0 | 19.4 | 19.4 | 19.4 | 19.6 | 13.4 | 12.3 | 11.9 | 11.1 |
| | Average | 13.1 | 15.0 | 9.9 | 15.7 | 16.8 | 15.8 | 16.6 | 8.4 | 8.3 | 10.3 | 8.3 | 17.5 | 18.2 | 17.9 | 16.9 | 16.6 | 17.9 | 17.5 | 16.1 |
| 64 | Mamba-1.4B | 9.5 | 15.1 | 9.5 | 8.1 | 12.0 | 9.9 | 12.7 | 8.1 | 9.2 | 10.9 | 6.3 | 7.7 | 14.8 | 9.2 | 16.2 | 7.0 | 8.5 | 6.3 | 10.2 |
| | Mamba-2.8B | 14.8 | 14.8 | 18.3 | 12.3 | 13.0 | 13.0 | 15.8 | 11.3 | 12.3 | 14.8 | 9.5 | 11.6 | 19.7 | 12.3 | 18.3 | 9.5 | 14.1 | 12.7 | 16.5 |
| | Pythia-2.8B | 11.3 | 16.5 | 11.3 | 9.5 | 13.4 | 12.3 | 15.1 | 13.0 | 9.9 | 12.7 | 7.4 | 9.9 | 18.3 | 11.3 | 18.7 | 8.1 | 9.9 | 7.4 | 13.4 |
| | Pythia-6.9B | 11.3 | 15.8 | 12.7 | 10.6 | 13.0 | 12.7 | 14.4 | 16.2 | 7.7 | 15.5 | 5.6 | 12.7 | 18.3 | 13.0 | 17.3 | 10.2 | 12.7 | 10.9 | 16.5 |
| | Pythia-12B | 13.4 | 16.2 | 10.6 | 11.6 | 14.1 | 11.3 | 12.0 | 14.4 | 10.2 | 14.4 | 9.2 | 12.7 | 20.8 | 16.9 | 17.6 | 16.9 | 15.8 | 15.8 | 16.9 |
| | Llama-13B | 13.7 | 13.4 | 14.4 | 12.0 | 14.1 | 12.3 | 14.8 | 4.6 | 6.0 | 6.0 | 5.6 | 13.4 | 21.1 | 12.3 | 14.4 | 23.2 | 31.3 | 31.3 | 29.2 |
| | Llama-30B | 7.7 | 16.9 | 11.6 | 13.4 | 13.0 | 13.7 | 15.8 | 8.5 | 7.4 | 6.7 | 6.7 | 14.8 | 17.3 | 10.6 | 16.5 | 25.8 | 30.6 | 30.6 | 35.6 |
| | OPT-66B | 12.3 | 14.8 | 13.7 | 13.4 | 14.8 | 15.1 | 14.1 | 13.7 | 8.5 | 12.7 | 6.7 | 13.0 | 22.2 | 14.8 | 19.7 | 11.6 | 15.5 | 11.3 | 13.4 |
| | GPT-NeoX-20B | 13.4 | 19.0 | 18.3 | 14.8 | 19.4 | 15.5 | 18.7 | 15.5 | 14.4 | 15.5 | 12.0 | 15.1 | 20.8 | 14.4 | 17.6 | 10.6 | 13.4 | 15.1 | 13.0 |
| | Average | 11.9 | 15.8 | 13.4 | 11.7 | 14.1 | 12.9 | 14.8 | 11.7 | 9.5 | 12.1 | 7.7 | 12.3 | 19.2 | 12.8 | 17.4 | 12.4 | 17.7 | 15.7 | 18.3 |
| 128 | Mamba-1.4B | 11.5 | 17.3 | 13.7 | 13.7 | 18.7 | 10.1 | 14.4 | 11.5 | 12.2 | 16.5 | 12.2 | 5.0 | 20.9 | 10.8 | 23.7 | 6.5 | 9.4 | 13.7 | 8.6 |
| | Mamba-2.8B | 15.1 | 20.1 | 17.3 | 16.5 | 22.3 | 16.5 | 15.8 | 10.8 | 15.1 | 18.0 | 20.9 | 12.2 | 30.2 | 18.7 | 20.1 | 13.0 | 16.5 | 15.1 | 15.1 |
| | Pythia-2.8B | 8.6 | 16.5 | 12.2 | 14.4 | 11.5 | 7.2 | 13.0 | 7.2 | 10.1 | 11.5 | 5.8 | 12.2 | 21.6 | 15.8 | 18.0 | 7.9 | 10.1 | 13.0 | 8.6 |
| | Pythia-6.9B | 11.5 | 20.9 | 17.3 | 16.5 | 15.1 | 14.4 | 16.5 | 12.2 | 8.6 | 10.8 | 8.6 | 16.5 | 23.0 | 16.5 | 21.6 | 18.0 | 18.7 | 14.4 | 17.3 |
| | Pythia-12B | 12.2 | 19.4 | 10.1 | 19.4 | 14.4 | 16.5 | 12.2 | 8.6 | 10.8 | 10.1 | 13.0 | 18.7 | 33.1 | 10.8 | 29.5 | 9.4 | 15.1 | 20.1 | 18.7 |
| | Llama-13B | 15.8 | 21.6 | 13.7 | 18.0 | 23.7 | 21.6 | 25.9 | 4.3 | 7.9 | 6.5 | 7.9 | 15.1 | 30.9 | 20.9 | 33.1 | 35.3 | 46.8 | 45.3 | 49.6 |
| | Llama-30B | 11.5 | 19.4 | 14.4 | 18.0 | 25.9 | 15.1 | 23.0 | 17.3 | 13.0 | 11.5 | 10.1 | 17.3 | 33.8 | 16.5 | 37.4 | 21.6 | 34.5 | 49.6 | 46.8 |
| | OPT-66B | 11.5 | 18.7 | 12.9 | 18.7 | 15.8 | 18.0 | 15.8 | 16.5 | 17.3 | 13.0 | 12.2 | 15.1 | 31.7 | 21.6 | 23.7 | 13.7 | 15.8 | 22.3 | 18.0 |
| | GPT-NeoX-20B | 15.1 | 22.3 | 18.7 | 20.1 | 25.9 | 20.1 | 24.5 | 18.0 | 12.2 | 11.5 | 13.0 | 23.0 | 32.4 | 23.7 | 31.7 | 20.9 | 9.4 | 20.9 | 12.2 |
| | Average | 12.6 | 19.6 | 14.5 | 17.3 | 19.3 | 15.9 | 17.5 | 11.4 | 12.4 | 12.9 | 11.5 | 15.0 | 28.6 | 17.3 | 26.5 | 16.2 | 19.6 | 23.8 | 21.7 |

Table 8: Best AUROC results cross different $\alpha$ on WikiMIA benchmark (Shi et al., 2024). **w/ LPDR**(Ours) utilizes linear weights for reweighting. *Ori.* and *Para.* denote the original and paraphrased settings. For each method pair, the higher score is in **bold**. The performance gains of our method on the average results are highlighted in purple.

| Len. | Method | Mamba-1.4B | | Pythia-6.9B | | LLaMA-13B | | GPT-NeoX-20B | | OPT-66B | | Average | |
|---|---|---|---|---|---|---|---|---|---|---|---|---|---|
| | | *Ori.* | *Para.* | *Ori.* | *Para.* | *Ori.* | *Para.* | *Ori.* | *Para.* | *Ori.* | *Para.* | *Ori.* | *Para.* |
| 32 | Loss | 61.0 | 61.3 | 63.8 | 64.1 | 67.5 | 68.0 | 69.1 | 68.6 | 65.6 | 65.3 | 65.4 | 65.5 |
| | w/ LPDR (Ours) | **61.9** | **62.0** | **64.0** | **64.2** | **67.7** | **68.2** | 69.1 | 68.6 | 65.6 | 65.2 | **65.7**$^{+0.3}$ | **65.6**$^{+0.1}$ |
| | Ref | 62.2 | 62.3 | **63.6** | **63.5** | **57.9** | **56.2** | **67.6** | **66.7** | 68.6 | 68.0 | **64.0** | **63.3** |
| | w/ LPDR (Ours) | 62.2 | **62.4** | 63.5 | 63.5 | 57.8 | 56.1 | 67.4 | 66.6 | 68.6 | **68.1** | 63.9 | 63.3 |
| | Min-$k$% | 63.3 | 62.9 | 66.3 | 65.1 | 66.8 | 66.2 | **72.2** | **69.6** | 67.5 | 65.8 | 67.2 | 65.9 |
| | w/ LPDR (Ours) | **64.0** | **63.5** | 66.3 | 65.1 | **67.0** | **66.3** | 72.0 | 69.4 | **67.7** | 65.8 | **67.4**$^{+0.2}$ | **66.0**$^{+0.1}$ |
| | Min-$k$%++ | 66.4 | 65.7 | 70.3 | 67.6 | 84.4 | 82.7 | 75.1 | 69.7 | 69.7 | 67.0 | 73.2 | 70.5 |
| | w/ LPDR (Ours) | **67.6** | **66.3** | **70.8** | **67.8** | **86.3** | **84.5** | **75.2** | 69.7 | **70.2** | **67.1** | **74.0**$^{+0.8}$ | **71.1**$^{+0.6}$ |
| 64 | Loss | 58.2 | 56.4 | 60.7 | 59.3 | 63.6 | 63.1 | 66.6 | 64.4 | 62.3 | 60.3 | 62.3 | 60.7 |
| | w/ LPDR (Ours) | **59.7** | **59.5** | **62.6** | **62.4** | **65.0** | **66.2** | **67.6** | **67.2** | **64.2** | **63.1** | **63.8**$^{+1.5}$ | **63.7**$^{+3.0}$ |
| | Ref | 60.6 | 59.6 | 62.4 | 62.9 | **63.4** | 60.9 | 66.0 | 66.0 | 66.9 | 67.8 | 63.9 | 63.4 |
| | w/ LPDR (Ours) | **61.1** | **60.8** | **63.3** | **64.0** | 63.3 | 60.9 | **66.8** | **67.2** | **68.2** | **69.1** | **64.5**$^{+0.6}$ | **64.4**$^{+1.0}$ |
| | Min-$k$% | 61.7 | 58.0 | 65.0 | 61.1 | 66.0 | 63.5 | 72.2 | 66.1 | 66.5 | 62.5 | 66.3 | 62.2 |
| | w/ LPDR (Ours) | **63.5** | **61.8** | **67.2** | **65.1** | **67.4** | **67.2** | **72.9** | **69.1** | **68.5** | **66.0** | **67.9**$^{+1.6}$ | **65.8**$^{+3.6}$ |
| | Min-$k$%++ | 67.2 | 62.2 | 71.6 | 64.2 | 84.3 | 78.8 | 76.5 | 66.2 | 69.8 | 63.3 | 73.9 | 66.9 |
| | w/ LPDR (Ours) | **68.3** | **65.5** | **72.7** | **68.3** | **87.2** | **84.3** | **77.2** | **68.2** | **70.6** | **66.6** | **75.2**$^{+1.3}$ | **70.6**$^{+3.7}$ |
| 128 | Loss | 63.3 | 62.7 | 65.1 | 64.7 | 67.8 | 67.2 | 70.7 | 69.7 | 65.5 | 64.5 | 66.5 | 65.8 |
| | w/ LPDR (Ours) | **63.7** | **64.1** | **65.6** | **66.6** | **68.7** | **69.1** | **70.9** | **71.4** | **66.7** | **66.9** | **67.1**$^{+0.6}$ | **67.6**$^{+1.8}$ |
| | Ref | 62.0 | 61.1 | 63.3 | 62.9 | 62.6 | 59.7 | 68.3 | 68.4 | 66.9 | 67.0 | 64.6 | 63.8 |
| | w/ LPDR (Ours) | **64.1** | **64.6** | **65.1** | **65.9** | **64.9** | **61.7** | **69.5** | **70.2** | **68.6** | **69.5** | **66.4**$^{+1.8}$ | **66.4**$^{+2.6}$ |
| | Min-$k$% | 66.8 | 64.4 | 69.5 | 67.0 | 71.5 | 68.6 | 75.6 | 73.0 | 70.6 | 67.2 | 70.8 | 68.0 |
| | w/ LPDR (Ours) | **67.4** | **66.5** | **69.8** | **69.3** | **72.2** | **71.4** | **76.4** | **75.8** | **72.1** | **70.6** | **71.6**$^{+0.8}$ | **70.7**$^{+2.7}$ |
| | Min-$k$%++ | 67.7 | 63.3 | 69.8 | 65.9 | 83.8 | 76.2 | 75.4 | 70.6 | 71.1 | 67.0 | 73.6 | 68.6 |
| | w/ LPDR (Ours) | **70.2** | **68.2** | **72.4** | **72.2** | **88.4** | **84.3** | **75.9** | **72.6** | **73.3** | **69.5** | **76.0**$^{+2.4}$ | **73.4**$^{+4.8}$ |

### E.2  WikiMIA Results on best $\alpha$

In this subsection, we present the best-performing results for our LPDR method on the WikiMIA benchmark. These results were obtained by selecting the optimal hyperparameter $\alpha$ for the linear decay function from the search space detailed in Section B. Table 8 showcases these results.

### E.3  ROC curve Visualization

This section provides ROC curve visualizations to offer a more detailed view of our method's performance. Figure 7 plots the ROC curves for several baseline methods and their LPDR-enhanced counterparts on the WikiMIA benchmark (length 128). Specifically, we show results for (a) Llama-13B on the paraphrased setting and (b) Pythia-6.9B on the original setting. As illustrated, the PDR-enhanced methods consistently offer a more favorable trade-off, achieving a higher True Positive Rate (TPR) for any given False Positive Rate (FPR). This enhanced discriminative capability helps to explain the AUROC gains reported throughout the paper.

### E.4  Static Analysis

To rigorously assess the statistical significance of our method's improvements, we performed a non-parametric bootstrap analysis based on the model prediction scores and ground-truth labels. Our procedure is as follows: To rigorously assess the statistical significance of our method's improvements, we performed a non-parametric bootstrap analysis based on the model prediction scores and ground-truth labels. Our procedure is as follows:

- **Bootstrap Resampling:** For each method, we generated $N = 1000$ bootstrap replicates. Each replicate was created by sampling indices from the original test set with replacement, forming a new dataset of the same size. A random seed was fixed to ensure reproducibility. If a replicate happened to contain samples from only one class, it was discarded for that specific calculation.

- **Metrics and Confidence Intervals:** For each of the $N$ bootstrap replicates, we calculated the AUROC and TPR@0.5%FPR. After generating all replicates, we computed the mean

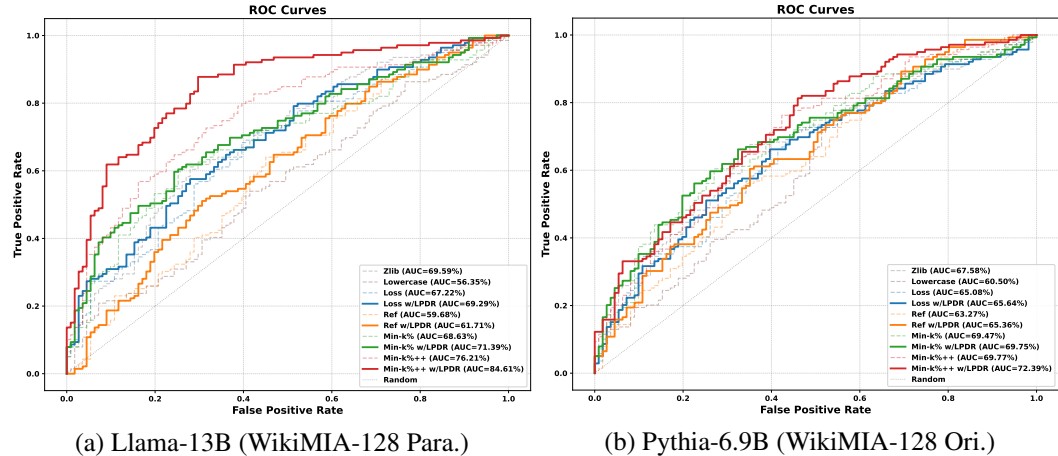

(a) Llama-13B (WikiMIA-128 Para.)      (b) Pythia-6.9B (WikiMIA-128 Ori.)

Figure 7: ROC curve comparison for various baseline methods and their PDR-enhanced versions on the WikiMIA-128 benchmark.

and standard deviation of these metrics. The 95% confidence intervals were then derived empirically from the 2.5th and 97.5th percentiles of the resulting distribution of 1000 metric values.

- **Paired Significance Testing:** To evaluate if the improvements of our PDR-enhanced methods over their respective baselines are statistically significant, we conducted a paired bootstrap test. This approach is crucial for reducing variance caused by the sampling process. For each of the $N = 1000$ replicates, we used the *exact same set of resampled indices* to evaluate both the baseline method and the PDR-enhanced method. We then calculated the performance difference for that replicate: $\delta = \text{AUROC}_{\text{PDR}} - \text{AUROC}_{\text{Baseline}}$. A one-sided P-value was subsequently derived by calculating the proportion of replicates where this difference was not positive ($\delta \leq 0$). This P-value directly tests the null hypothesis $H_0 : \text{AUROC}_{\text{PDR}} \leq \text{AUROC}_{\text{Baseline}}$. A P-value less than 0.05 is considered to indicate a statistically significant improvement.

The analysis reveals a clear trend: the effectiveness of PDR is strongly correlated with sequence length. For short sequences (e.g., 32 tokens), the performance gains are marginal and not always statistically significant. However, as the sequence length increases to 64 and 128, PDR's improvements become both substantial and statistically significant (p-value $< 0.05$) for most baselines. For instance, on Pythia-6.9B with length 128, PDR boosts the AUROC of Min-k%++ from 65.9 to 72.2 (p-value $< 0.001$). This demonstrates that the positional prior becomes a more robust and discriminative signal as more context becomes available in longer sequences, confirming that the observed gains are not due to random noise.

Table 9: Performance comparison on Pythia-6.9B across different sequence lengths. We report AUROC and TPR@0.5%FPR (mean ± std). **w/ LPDR** denotes our method using linear weights. The p-value indicates the statistical significance of the improvement of LPDR over the corresponding baseline.

| Length | Method | AUROC | AUROC Std | TPR@0.5%FPR | TPR Std | p-value |
|---|---|---|---|---|---|---|
| 32 | Zlib | 64.2 | 2.0 | 12.7 | 2.6 | - |
| | Lowercase | 61.7 | 2.1 | 11.9 | 3.2 | - |
| | Loss | 64.1 | 2.0 | 15.0 | 2.8 | - |
| | **w/ LPDR** | 64.2 | 2.0 | 15.0 | 2.8 | 0.275 |
| | Ref | 63.5 | 2.0 | 6.2 | 3.0 | - |
| | **w/ LPDR** | 63.5 | 2.0 | 5.7 | 3.2 | 0.785 |
| | Min-$k$% | 65.1 | 2.0 | 21.7 | 3.5 | - |
| | **w/ LPDR** | 65.1 | 2.0 | **22.0** | 4.0 | 0.342 |
| | Min-$k$%++ | 67.6 | 2.0 | 14.5 | 3.2 | - |
| | **w/ LPDR** | **67.8** | 2.0 | 14.5 | 2.6 | 0.218 |
| 64 | Zlib | 61.6 | 2.5 | 15.8 | 3.5 | - |
| | Lowercase | 57.7 | 2.5 | 11.3 | 2.2 | - |
| | Loss | 59.3 | 2.5 | 10.6 | 3.7 | - |
| | **w/ LPDR** | **62.4** | 2.5 | **13.0** | 5.4 | **0.001** |
| | Ref | 62.9 | 2.4 | 16.2 | 2.8 | - |
| | **w/ LPDR** | **64.0** | 2.3 | 7.7 | 3.6 | 0.152 |
| | Min-$k$% | 61.1 | 2.5 | 12.7 | 3.0 | - |
| | **w/ LPDR** | **65.1** | 2.4 | **18.3** | 5.2 | **0.019** |
| | Min-$k$%++ | 64.2 | 2.4 | 10.2 | 3.2 | - |
| | **w/ LPDR** | **68.3** | 2.4 | **12.7** | 4.6 | **0.006** |
| 128 | Zlib | 67.4 | 3.4 | 20.9 | 5.8 | - |
| | Lowercase | 59.9 | 3.6 | 11.5 | 4.2 | - |
| | Loss | 64.7 | 3.5 | 16.5 | 6.8 | - |
| | **w/ LPDR** | **66.6** | 3.4 | 15.1 | 4.9 | 0.061 |
| | Ref | 62.9 | 3.5 | 8.6 | 5.7 | - |
| | **w/ LPDR** | **65.9** | 3.4 | **10.8** | 5.0 | **0.016** |
| | Min-$k$% | 67.0 | 3.4 | 16.5 | 6.1 | - |
| | **w/ LPDR** | **68.9** | 3.3 | **23.0** | 9.4 | 0.270 |
| | Min-$k$%++ | 65.9 | 3.5 | 18.0 | 7.1 | - |
| | **w/ LPDR** | **72.2** | 3.3 | **18.7** | 6.9 | **0.001** |

## F FSD RESULTS

This section presents the detailed results of combining our method with the Finetuning-based Score Difference (FSD) framework on the WikiMIA dataset. Please refer to Tab. 10 and Tab. 11 for the full experimental results.

Table 10: AUROC results on WikiMIA benchmark, compare with FSD

| Dataset | Method | GPT-J-6B | | OPT-6.7B | | Pythia-6.9B | | LLaMA-7B | | NeoX-20B | | Average | |
|---|---|---|---|---|---|---|---|---|---|---|---|---|---|
| | | Base | w/ FSD | Base | w/ FSD | Base | w/ FSD | Base | w/ FSD | Base | w/ FSD | Base | w/ FSD |
| WikiMIA | Min-$k$% | 67.9 | 93.4 | 62.5 | 89.4 | 66.7 | 91.9 | 65.4 | 88.4 | 73.4 | 90.1 | 67.2 | 90.6 |
| | w/ LPDR | 68.1 | 94.0 | 62.9 | 89.6 | 67.0 | 92.6 | 66.7 | 88.8 | 73.5 | 91.0 | **67.6** | **91.2** |
| | w/ EPDR | 67.1 | 94.1 | 62.1 | 90.0 | 66.0 | 92.8 | 65.8 | 89.1 | 72.1 | 91.3 | 66.6 | **91.4** |
| | w/ PPDR | 67.9 | 93.8 | 62.9 | 89.5 | 66.9 | 92.2 | 66.6 | 88.9 | 73.3 | 90.7 | **67.5** | 91.0 |
| | Min-$k$%++ | 67.6 | 81.6 | 63.0 | 83.3 | 68.1 | 81.5 | 79.9 | 91.1 | 74.4 | 77.8 | 70.6 | 83.1 |
| | w/ LPDR | 68.3 | 84.8 | 63.6 | 84.5 | 69.4 | 85.0 | 80.8 | 92.1 | 74.8 | 81.7 | **71.4** | **85.6** |
| | w/ EPDR | 68.2 | 83.7 | 63.3 | 84.5 | 69.2 | 83.6 | 80.8 | 91.8 | 74.8 | 81.2 | **71.2** | 85.0 |
| | w/ PPDR | 68.3 | 85.5 | 63.7 | 84.6 | 69.5 | 85.8 | 80.7 | 92.1 | 74.6 | 82.4 | **71.4** | **86.1** |

Table 11: TPR results on WikiMIA benchmark, compare with FSD

| Dataset | Method | GPT-J-6B | | OPT-6.7B | | Pythia-6.9B | | LLaMA-7B | | NeoX-20B | | Average | |
|---|---|---|---|---|---|---|---|---|---|---|---|---|---|
| | | Base | w/ FSD | Base | w/ FSD | Base | w/ FSD | Base | w/ FSD | Base | w/ FSD | Base | w/ FSD |
| WikiMIA | Min-$k$% | 17.2 | 55.6 | 13.9 | 40.6 | 17.2 | 57.6 | 14.7 | 32.6 | 24.7 | 36.2 | 17.5 | 44.5 |
| | w/ LPDR | 18.0 | 61.9 | 16.4 | 41.9 | 18.2 | 61.1 | 16.4 | 33.7 | 27.7 | 52.9 | **19.3** | **50.3** |
| | w/ EPDR | 16.5 | 60.1 | 13.5 | 45.4 | 16.5 | 59.8 | 17.0 | 33.6 | 21.5 | 50.8 | 17.0 | 49.9 |
| | w/ PPDR | 17.9 | 59.9 | 15.4 | 39.9 | 19.9 | 61.9 | 18.5 | 31.6 | 27.0 | 50.3 | **19.7** | **48.7** |
| | Min-$k$%++ | 15.9 | 24.2 | 11.7 | 29.0 | 19.0 | 25.0 | 20.4 | 39.1 | 17.5 | 10.5 | 16.9 | 25.6 |
| | w/ LPDR | 18.9 | 32.2 | 12.9 | 35.7 | 19.0 | 37.4 | 22.2 | 48.4 | 19.7 | 15.5 | **18.5** | **33.9** |
| | w/ EPDR | 19.0 | 39.6 | 12.2 | 32.4 | 19.4 | 34.9 | 20.2 | 46.4 | 19.0 | 18.7 | **18.0** | **34.4** |
| | w/ PPDR | 18.9 | 34.9 | 12.9 | 36.7 | 17.0 | 40.4 | 24.0 | 51.6 | 19.7 | 17.4 | **18.5** | **36.2** |

# G   MIMIR RESULTS

This section presents the complete AUROC and TPR results on the challenging MIMIR benchmark, which is known for its minimal distribution shift and increased difficulty compared to WikiMIA. The results in Tab. 12 and Tab. 13 comprehensively demonstrate the performance of our proposed PDR methods (LPDR, EPDR, PPDR) and all baselines on MIMIR, across different models and sub datasets.

Table 12: AUROC results on the challenging MIMIR benchmark.[†]Neighbor results are from Zhang et al. (2025b), induces significant extra computational cost than others (25× in this case), for which reason we don't run on the 12B model.

| Method | Wikipedia | | | | | Github | | | | | Pile CC | | | | | PubMed Central | | | | |
|---|---|---|---|---|---|---|---|---|---|---|---|---|---|---|---|---|---|---|---|---|
| | 160M | 1.4B | 2.8B | 6.9B | 12B | 160M | 1.4B | 2.8B | 6.9B | 12B | 160M | 1.4B | 2.8B | 6.9B | 12B | 160M | 1.4B | 2.8B | 6.9B | 12B |
| Lowercase | 50.1 | 51.3 | 51.7 | 53.5 | 54.3 | 67.2 | 70.3 | 71.3 | 72.9 | 73.7 | 47.8 | 48.6 | 49.5 | 50.1 | 50.6 | 49.5 | 50.4 | 51.5 | 51.5 | 52.7 |
| Zlib | 51.1 | 52.0 | 52.4 | 53.5 | 54.3 | 67.5 | 71.0 | 72.3 | 73.9 | 74.9 | 49.6 | 50.1 | 50.3 | 50.8 | 51.1 | 49.9 | 50.0 | 50.1 | 50.6 | 51.2 |
| [†]Neighbor | 50.7 | 51.7 | 52.2 | 53.2 | / | 65.3 | 69.4 | 70.5 | 72.1 | / | 49.6 | 50.0 | 50.1 | 50.8 | / | 47.9 | 49.1 | 49.7 | 50.1 | / |
| Loss | 50.2 | 51.3 | 51.8 | 52.8 | 53.5 | 65.7 | 69.8 | 71.3 | 73.0 | 74.0 | 49.6 | 50.0 | 50.1 | 50.7 | 51.1 | 49.9 | 49.8 | 49.9 | 50.6 | 51.3 |
| w/ LPDR | 51.2 | 52.0 | 52.6 | 53.7 | 54.4 | 66.0 | 70.0 | 71.5 | 73.0 | 73.9 | 49.9 | 50.4 | 50.5 | 51.0 | 51.4 | 50.1 | 49.9 | 50.0 | 50.6 | 51.1 |
| w/ EPDR | 50.5 | 51.5 | 51.9 | 53.0 | 53.7 | 65.7 | 69.9 | 71.3 | 73.0 | 74.0 | 49.6 | 50.0 | 50.1 | 50.7 | 51.0 | 49.7 | 49.6 | 49.7 | 50.3 | 50.9 |
| w/ PPDR | 50.4 | 51.4 | 51.9 | 52.9 | 53.6 | 66.0 | 70.0 | 71.5 | 73.2 | 74.2 | 49.6 | 50.1 | 50.2 | 50.8 | 51.1 | 49.9 | 49.8 | 49.9 | 50.6 | 51.3 |
| Min-$k$% | 48.8 | 51.0 | 51.7 | 53.1 | 54.2 | 65.7 | 70.0 | 71.4 | 73.3 | 74.2 | 50.1 | 50.5 | 50.5 | 51.2 | 51.5 | 50.3 | 50.3 | 50.5 | 51.2 | 52.3 |
| w/ LPDR | 52.7 | 53.3 | 54.1 | 55.0 | 56.0 | 65.9 | 70.2 | 71.7 | 73.2 | 74.1 | 50.2 | 51.0 | 51.1 | 51.6 | 51.9 | 50.7 | 50.7 | 50.6 | 51.1 | 52.0 |
| w/ EPDR | 50.4 | 51.7 | 52.5 | 53.9 | 54.9 | 65.7 | 70.0 | 71.5 | 73.3 | 74.3 | 50.4 | 50.7 | 50.7 | 51.4 | 51.6 | 50.1 | 50.1 | 50.2 | 50.8 | 51.7 |
| w/ PPDR | 49.2 | 51.3 | 52.1 | 53.5 | 54.5 | 66.1 | 70.3 | 71.8 | 73.5 | 74.5 | 50.2 | 50.6 | 50.6 | 51.4 | 51.6 | 50.4 | 50.4 | 50.5 | 51.2 | 52.4 |
| Min-$k$%++ | 49.2 | 53.1 | 53.8 | 56.1 | 57.9 | 64.7 | 69.6 | 70.9 | 72.8 | 74.2 | 49.7 | 50.0 | 49.8 | 51.2 | 51.8 | 50.2 | 50.8 | 51.5 | 52.8 | 54.0 |
| w/ LPDR | 51.0 | 54.8 | 55.5 | 57.5 | 59.0 | 64.6 | 69.5 | 70.6 | 72.3 | 73.7 | 49.7 | 50.6 | 49.9 | 51.8 | 51.9 | 50.4 | 51.8 | 52.2 | 53.6 | 54.4 |
| w/ EPDR | 49.9 | 53.7 | 54.4 | 56.7 | 58.5 | 64.7 | 69.7 | 70.8 | 72.7 | 74.1 | 49.9 | 50.3 | 49.9 | 51.5 | 51.8 | 50.0 | 50.9 | 51.5 | 52.8 | 53.7 |
| w/ PPDR | 49.3 | 53.5 | 54.2 | 56.5 | 58.3 | 64.8 | 69.8 | 71.0 | 72.9 | 74.3 | 49.7 | 50.2 | 49.8 | 51.4 | 51.9 | 50.2 | 51.1 | 51.7 | 53.1 | 54.3 |

| Method | ArXiv | | | | | DM Mathematics | | | | | HackerNews | | | | | Average | | | | |
|---|---|---|---|---|---|---|---|---|---|---|---|---|---|---|---|---|---|---|---|---|
| | 160M | 1.4B | 2.8B | 6.9B | 12B | 160M | 1.4B | 2.8B | 6.9B | 12B | 160M | 1.4B | 2.8B | 6.9B | 12B | 160M | 1.4B | 2.8B | 6.9B | 12B |
| Lowercase | 50.8 | 50.7 | 51.3 | 51.9 | 52.8 | 48.9 | 49.0 | 49.0 | 49.1 | 48.2 | 49.0 | 50.4 | 51.1 | 51.6 | 52.3 | 52.4 | 53.4 | 54.1 | 54.8 | 55.4 |
| Zlib | 50.1 | 50.9 | 51.3 | 52.2 | 52.7 | 48.1 | 48.2 | 48.0 | 48.1 | 48.1 | 49.7 | 50.3 | 50.8 | 51.2 | 51.7 | 52.7 | 53.7 | 54.1 | 54.9 | 55.4 |
| [†]Neighbor | 50.7 | 51.4 | 51.8 | 52.2 | / | 49.0 | 47.0 | 46.8 | 46.6 | / | 50.9 | 51.7 | 51.5 | 51.9 | / | 52.0 | 52.9 | 53.2 | 53.8 | / |
| Loss | 51.0 | 51.5 | 51.9 | 52.9 | 53.4 | 48.8 | 48.5 | 48.4 | 48.5 | 48.5 | 49.4 | 50.5 | 51.3 | 52.1 | 52.8 | 52.5 | 53.5 | 53.9 | 54.7 | 55.3 |
| w/ LPDR | 50.4 | 51.1 | 51.5 | 52.7 | 53.2 | 48.7 | 48.6 | 48.4 | 48.5 | 48.6 | 49.7 | 50.6 | 51.3 | 52.2 | 52.6 | **52.7** | **53.7** | **54.1** | **54.9** | **55.4** |
| w/ EPDR | 50.7 | 51.2 | 51.6 | 52.5 | 53.0 | 48.5 | 48.3 | 48.2 | 48.3 | 48.3 | 49.5 | 50.3 | 51.0 | 51.6 | 52.1 | 52.4 | 53.4 | 53.8 | 54.6 | 55.1 |
| w/ PPDR | 50.9 | 51.4 | 51.9 | 52.9 | 53.4 | 48.8 | 48.6 | 48.4 | 48.5 | 48.5 | 49.4 | 50.5 | 51.3 | 52.1 | 52.8 | **52.6** | **53.6** | **54.0** | **54.8** | **55.4** |
| Min-$k$% | 50.4 | 51.4 | 52.1 | 53.4 | 54.3 | 49.3 | 49.3 | 49.1 | 49.2 | 49.2 | 50.6 | 51.2 | 52.4 | 53.5 | 54.5 | 52.4 | 53.7 | 54.2 | 55.2 | 56.0 |
| w/ LPDR | 49.2 | 50.8 | 51.2 | 52.5 | 53.2 | 49.5 | 49.7 | 49.3 | 49.5 | 49.5 | 50.7 | 51.3 | 52.4 | 52.8 | 54.3 | **53.0** | **54.3** | **54.6** | **55.5** | **56.1** |
| w/ EPDR | 49.9 | 51.0 | 51.5 | 52.8 | 53.5 | 49.3 | 49.4 | 49.1 | 49.2 | 49.2 | 50.5 | 51.0 | 51.9 | 52.9 | 53.3 | 52.6 | 53.8 | 54.3 | 55.2 | 55.9 |
| w/ PPDR | 50.2 | 51.4 | 52.1 | 53.4 | 54.3 | 49.3 | 49.4 | 49.2 | 49.3 | 49.3 | 50.5 | 51.1 | 52.3 | 53.5 | 54.4 | **52.6** | **53.9** | **54.4** | **55.4** | **56.1** |
| Min-$k$%++ | 49.3 | 50.9 | 53.0 | 53.6 | 56.2 | 50.1 | 50.2 | 50.2 | 50.5 | 50.4 | 50.7 | 51.3 | 52.6 | 54.1 | 55.8 | 52.2 | 54.1 | 54.9 | 56.2 | 57.4 |
| w/ LPDR | 50.0 | 51.1 | 52.4 | 53.9 | 56.1 | 50.1 | 50.3 | 50.2 | 50.4 | 50.5 | 50.7 | 50.9 | 51.7 | 53.0 | 54.3 | **52.6** | **54.7** | **55.1** | **56.6** | **57.6** |
| w/ EPDR | 49.6 | 50.8 | 52.6 | 53.4 | 55.7 | 49.8 | 50.0 | 49.9 | 50.2 | 50.2 | 50.6 | 50.9 | 52.0 | 53.3 | 54.7 | 52.3 | 54.2 | 54.9 | 56.2 | 57.3 |
| w/ PPDR | 49.5 | 51.0 | 53.1 | 53.8 | 56.3 | 50.2 | 50.2 | 50.2 | 50.6 | 50.5 | 50.5 | 51.2 | 52.5 | 54.0 | 55.6 | **52.3** | **54.3** | **55.0** | **56.4** | **57.6** |

Table 13: TPR results on the challenging MIMIR benchmark.[†]Neighbor results are from Zhang et al. (2025b), induces significant extra computational cost than others ($25\times$ in this case), for which reason we don't run on the 12B model.

| Method | Wikipedia | | | | | Github | | | | | Pile CC | | | | | PubMed Central | | | | |
|---|---|---|---|---|---|---|---|---|---|---|---|---|---|---|---|---|---|---|---|---|
| | 160M | 1.4B | 2.8B | 6.9B | 12B | 160M | 1.4B | 2.8B | 6.9B | 12B | 160M | 1.4B | 2.8B | 6.9B | 12B | 160M | 1.4B | 2.8B | 6.9B | 12B |
| Lowercase | 4.6 | 4.5 | 4.9 | 5.2 | 5.6 | 24.4 | 32.2 | 34.3 | 38.1 | 38.6 | 3.4 | 5.3 | 5.3 | 6.2 | 6.4 | 3.5 | 5 | 5.3 | 6 | 5.2 |
| Zlib | 4.2 | 5.7 | 5.9 | 6.3 | 6.8 | 25.1 | 32.8 | 36.2 | 40.1 | 40.8 | 4 | 5.1 | 5.4 | 6.2 | 6.6 | 3.8 | 3.6 | 3.5 | 4.3 | 4.4 |
| [†]Neighbor | 4.0 | 4.5 | 4.9 | 5.8 | / | 24.7 | 31.6 | 29.8 | 34.1 | / | 3.9 | 3.6 | 4.0 | 5.3 | / | 3.9 | 3.7 | 4.5 | 4.5 | / |
| Loss | 4.2 | 4.7 | 4.7 | 5.1 | 5 | 22.6 | 32.1 | 33.6 | 38.5 | 40.7 | 3.1 | 5 | 4.8 | 4.9 | 5.1 | 4 | 4.4 | 4.3 | 4.9 | 5 |
| w/ LPDR | 4.6 | 4.6 | 5.4 | 5.7 | 6 | 21 | 30.8 | 32.4 | 34.7 | 36.4 | 2.4 | 3.4 | 4 | 4.8 | 5.8 | 4.1 | 4.6 | 4.5 | 4.9 | 5.5 |
| w/ EPDR | 4.5 | 4.8 | 5.1 | 5.5 | 5.7 | 21.5 | 31.6 | 32.6 | 36.6 | 38.7 | 3.2 | 4.5 | 4.6 | 4.7 | 5.2 | 3.7 | 4.6 | 4.6 | 4.8 | 4.9 |
| w/ PPDR | 4.2 | 4.7 | 4.9 | 4.9 | 5.3 | 22.7 | 32.1 | 35 | 38.8 | 39.8 | 3 | 4.3 | 4.7 | 5.3 | 6.1 | 4.1 | 4.2 | 4.1 | 4.4 | 5.2 |
| Min-$k$% | 4.8 | 5.6 | 5 | 6.1 | 5.8 | 22.6 | 31.5 | 34 | 39 | 40.7 | 3.5 | 4.5 | 4.8 | 5 | 4.8 | 4.7 | 4.6 | 4.5 | 5.1 | 4.9 |
| w/ LPDR | 5.5 | 6.4 | 6.2 | 6.2 | 7.2 | 21.4 | 30.9 | 32.7 | 36 | 36.7 | 2.3 | 2.9 | 3.1 | 2.4 | 3.6 | 6 | 5.6 | 5.6 | 5.7 | 4.8 |
| w/ EPDR | 5 | 5.9 | 5.9 | 5.8 | 6.8 | 20.4 | 31 | 32.9 | 37.1 | 38.4 | 3.3 | 4.4 | 4.4 | 4.1 | 5.2 | 4 | 4.6 | 4.9 | 5.5 | 5.8 |
| w/ PPDR | 5.3 | 5.3 | 5.4 | 5.6 | 5.6 | 22.9 | 31.9 | 34.6 | 39.1 | 40.2 | 3.8 | 4.5 | 4.2 | 4.7 | 5.2 | 5.3 | 5.2 | 4.7 | 5.2 | 6.4 |
| Min-$k$%++ | 5.2 | 5.3 | 5.9 | 7 | 7.8 | 25.2 | 33 | 34.2 | 38.2 | 40.1 | 5 | 3.7 | 3.7 | 4.8 | 4.6 | 4.8 | 6.1 | 4.8 | 5.6 | 6.4 |
| w/ LPDR | 4.6 | 6 | 6.3 | 7.5 | 7.9 | 22.6 | 30.6 | 32.7 | 35 | 39.7 | 4.7 | 2.9 | 3.4 | 4 | 5 | 4.2 | 6.4 | 5.5 | 7.8 | 6.2 |
| w/ EPDR | 5.1 | 5.9 | 6.6 | 7.8 | 8.2 | 24 | 33 | 33.5 | 37.4 | 40.7 | 4.6 | 3.4 | 3.6 | 4.3 | 4.9 | 4.9 | 5.4 | 5.4 | 6.8 | 5.8 |
| w/ PPDR | 5.2 | 5 | 5.5 | 7.7 | 7.5 | 26.2 | 34.2 | 34.9 | 38.8 | 40 | 5.2 | 3.5 | 3.3 | 4.3 | 4.6 | 5.1 | 6.4 | 5.1 | 5.8 | 6.7 |

| Method | ArXiv | | | | | DM Mathematics | | | | | HackerNews | | | | | Average | | | | |
|---|---|---|---|---|---|---|---|---|---|---|---|---|---|---|---|---|---|---|---|---|
| | 160M | 1.4B | 2.8B | 6.9B | 12B | 160M | 1.4B | 2.8B | 6.9B | 12B | 160M | 1.4B | 2.8B | 6.9B | 12B | 160M | 1.4B | 2.8B | 6.9B | 12B |
| Lowercase | 5.1 | 4.7 | 5.4 | 5.6 | 5.2 | 5.6 | 6.2 | 5.5 | 6.8 | 5.8 | 5.2 | 5.2 | 6.3 | 6.6 | 6.4 | 7.8 | 9.7 | 10.1 | 11.3 | 11.1 |
| Zlib | 2.9 | 4.3 | 4.1 | 4.6 | 4.7 | 4.1 | 5 | 4.6 | 4.3 | 4.3 | 5 | 5.5 | 5.8 | 5.6 | 5.8 | 7.4 | 9.4 | 10.0 | 11.0 | 11.3 |
| [†]Neighbor | 4.7 | 4.8 | 4.4 | 4.1 | / | **5.6** | 4.4 | 4.5 | 4.5 | / | **6.5** | 5.2 | 5.3 | 5.7 | / | 7.6 | 8.3 | 8.2 | 9.1 | / |
| Loss | 4 | 4.8 | 4.6 | 5.4 | 5.6 | 3.8 | 4.3 | 4.1 | 4.1 | 4 | 5 | 4.8 | 5.5 | 5.9 | 6.8 | 7.0 | 9.2 | 9.4 | 10.5 | 10.9 |
| w/ LPDR | 3.4 | 3.8 | 2.8 | 4.1 | 3.8 | 3.6 | 3.7 | 3.7 | 3.5 | 3.6 | 4.8 | 5.6 | 5.5 | 6 | 5.2 | 6.5 | 8.5 | 8.8 | 9.6 | 10.2 |
| w/ EPDR | 3.5 | 3.9 | 4 | 4.3 | 4.6 | 4.8 | 4.5 | 4.7 | 4.6 | 4.5 | 5.8 | 6.1 | 6 | 6 | 6.2 | 6.9 | 9.0 | 9.3 | 10.1 | 10.6 |
| w/ PPDR | 4.4 | 4.2 | 4.8 | 5.5 | 5.3 | 3.9 | 4.2 | 3.8 | 4.1 | 3.8 | 5.5 | 5.6 | 5.4 | 5.9 | 6.1 | **7.1** | 9.0 | **9.6** | 10.5 | 10.9 |
| Min-$k$% | 4.4 | 4.3 | 4.5 | 5.4 | 5.3 | 3.9 | 4.1 | 4.6 | 4.3 | 4.6 | 4.2 | 4.6 | 5.7 | 6.3 | 6.1 | 7.3 | 9.1 | 9.6 | 10.8 | 11.0 |
| w/ LPDR | 4 | 4.1 | 3.7 | 4.6 | 4.6 | 4 | 4.1 | 4.6 | 3.9 | 3.6 | 4.7 | 6.3 | 4.1 | 6 | 5.7 | 7.2 | 9.0 | 9.3 | 9.8 | 10.1 |
| w/ EPDR | 3.5 | 3.5 | 4 | 4.2 | 5 | 4.3 | 4.2 | 4.5 | 4.5 | 4.7 | 4.8 | 5.5 | 5 | 5.5 | 5.4 | 6.8 | 8.9 | 9.4 | 10.2 | 11.0 |
| w/ PPDR | 4.6 | 3.8 | 4.4 | 5 | 6.1 | 3.8 | 3.4 | 3.9 | 4.1 | 3.8 | 4.5 | 4.7 | 5.5 | 6.8 | 5.7 | **7.6** | 9.0 | 9.5 | 10.6 | **11.2** |
| Min-$k$%++ | 5.4 | 4.7 | 6.2 | 6.8 | 7 | 4.4 | 4.8 | 5.4 | 4.5 | 5.4 | 4.4 | 3.5 | 4.6 | 5.7 | 5.7 | 8.3 | 9.6 | 10.0 | 11.2 | 11.9 |
| w/ LPDR | 4.6 | 5.2 | 6.8 | 7.6 | 8.6 | 4.4 | 4.4 | 4.4 | 4.5 | 4.7 | 5.3 | 3.9 | 4.8 | 6.8 | 7.2 | 7.5 | 9.3 | 9.9 | 11.1 | **12.0** |
| w/ EPDR | 5.1 | 5 | 7.1 | 7.3 | 6.4 | 4.6 | 4.8 | 4.8 | 4.5 | 5 | 4.3 | 4.7 | 5.4 | 6.2 | 6.4 | 8.1 | 9.6 | **10.2** | **11.4** | 11.8 |
| w/ PPDR | 5.5 | 4.6 | 6.4 | 7.3 | 6.7 | 4.3 | 4.8 | 5.2 | 4.6 | 5.1 | 4.3 | 3.7 | 5.1 | 6.6 | 5.8 | **8.6** | **9.8** | 10.1 | **11.4** | 11.8 |

# H   COMPARE WITH FITTED LOSS SLOPE

Inspired by Context-Aware MIA (CAMIA) (Chang et al., 2025), which designs multiple dynamic signals for detection, we explore a variant based on one of its key signals: the loss decreasing rate. Specifically, for each sample $x$, we compute a slope by performing a linear regression of its token-level losses $L_t(x_t)$ against their positions $t$. The slope is calculated as:

$$f_{\text{Slope}}(x) = \frac{\sum_{t=1}^{T}(t-\bar{t})(L_t(x_t)-\bar{L})}{\sum_{t=1}^{T}(t-\bar{t})^2} \tag{20}$$

with $\bar{t} = \frac{T+1}{2}$ and $\bar{L} = \frac{1}{T}\sum_{t=1}^{T}L_t(x_t)$.

We then use this dynamically computed slope as the decay parameter $\alpha$ in our reweighting scheme, referring to this method as "w/ fitted slope PDR". We compare this dynamic approach with our fixed-hyperparameter LPDR. Table 14 shows the results on the WikiMIA benchmark across Pythia-2.8B, 6.9B, and 12B models, while Table 15 presents results on the MIMIR (DM Mathematics, Github, Pile CC) datasets.

The analysis reveals a consistent trend: while the sample-fitted slope method is dynamic, it often provides only marginal or inconsistent improvements over the baselines. This is because the slopes learned from individual samples tend to be relatively gentle or "smooth," resulting in weights that do not provide a strong enough reweighting signal to significantly enhance the distinction between member and non-member samples. In contrast, our LPDR, which often employs a steeper, pre-defined decay, more effectively amplifies the memorization signals present in the initial tokens. As shown in the tables, our LPDR method consistently and more substantially outperforms both the original baselines and the dynamic "w/ fitted slope PDR" approach across most settings, highlighting the effectiveness of a robust, albeit simpler, positional prior.

Table 14: AUROC comparison on the WikiMIA benchmark. For each group of three methods,"w/ fitted LPDR" denotes applying LPDR with slope fitted on single test sample loss, while "LPDR" uses a fixed default hyperparameter. The highest score per column is in **bold**. Our method's rows are highlighted in gray.

| Len. | Method | Pythia-2.8B | | Pythia-6.9B | | Pythia-12B | | Average | |
|---|---|---|---|---|---|---|---|---|---|
| | | *Ori.* | *Para.* | *Ori.* | *Para.* | *Ori.* | *Para.* | *Ori.* | *Para.* |
| | Loss | 61.37 | 61.57 | 63.83 | 64.07 | 65.44 | 65.60 | 63.55 | 63.75 |
| | w/fitted slope PDR | 61.42 | 61.60 | 63.79 | 64.06 | 65.33 | 65.60 | 63.52 | 63.75 |
| | w/LPDR | **61.73** | **61.83** | **63.97** | **64.20** | 65.45 | **65.70** | **63.71** | **63.91** |
| | Ref | 61.34 | 61.17 | 63.57 | 63.52 | 65.12 | 64.86 | 63.34 | 63.19 |
| | w/fitted slope PDR | 61.22 | 61.11 | 63.44 | 63.31 | 64.95 | 64.69 | 63.20 | 63.04 |
| 32 | w/LPDR | 61.34 | **61.18** | **63.55** | **63.47** | **65.09** | **64.90** | **63.33** | **63.18** |
| | Min-$k$% | 61.68 | 60.89 | 66.28 | 65.09 | 68.07 | 67.20 | 65.35 | 64.39 |
| | w/fitted slope PDR | 61.56 | 60.81 | 66.19 | 64.98 | 67.88 | 67.00 | 65.21 | 64.26 |
| | w/LPDR | **61.91** | **61.12** | **66.33** | **65.15** | **68.02** | 67.20 | **65.42** | **64.49** |
| | Min-$k$%++ | 63.97 | 61.33 | 70.27 | 67.56 | 72.24 | 69.40 | 68.83 | 66.10 |
| | w/fitted slope PDR | 64.08 | 61.48 | 70.37 | 67.70 | **72.32** | **69.56** | 68.92 | **66.24** |
| | w/LPDR | **64.74** | **61.66** | **70.85** | 67.74 | 72.26 | 69.36 | **69.28** | 66.25 |
| | Loss | 58.44 | 56.49 | 60.74 | 59.28 | 61.86 | 60.02 | 60.35 | 58.60 |
| | w/fitted slope PDR | 58.59 | 56.78 | 60.89 | 59.60 | 61.97 | 60.23 | 60.48 | 58.87 |
| | w/LPDR | **60.09** | **59.35** | **62.58** | **62.41** | **63.47** | **62.97** | **62.05** | **61.57** |
| | Ref | 59.62 | 59.22 | 62.38 | 62.89 | 63.04 | 63.18 | 61.68 | 61.76 |
| | w/fitted slope PDR | 59.77 | 59.34 | 62.51 | 63.10 | 63.16 | 63.33 | 61.81 | 61.92 |
| 64 | w/LPDR | **60.52** | **60.82** | **63.34** | **64.00** | **64.05** | **64.51** | **62.64** | **63.11** |
| | Min-$k$% | 61.20 | 56.68 | 64.97 | 61.07 | 66.50 | 62.49 | 64.22 | 60.08 |
| | w/fitted slope PDR | 61.39 | 57.03 | 65.19 | 61.40 | 66.69 | 62.79 | 64.42 | 60.41 |
| | w/LPDR | **63.26** | **61.59** | **66.67** | **65.07** | **67.94** | **66.08** | **65.95** | **64.25** |
| | Min-$k$%++ | 64.79 | 57.71 | 71.64 | 64.23 | 72.62 | 65.09 | 69.68 | 62.34 |
| | w/fitted slope PDR | 64.91 | 57.90 | 71.82 | 64.53 | 72.75 | 65.34 | 69.83 | 62.59 |
| | w/LPDR | **65.94** | **61.95** | **72.08** | **68.30** | **72.76** | **68.22** | **70.26** | **66.16** |
| | Loss | 62.81 | 62.31 | 65.08 | 64.66 | 65.77 | 65.40 | 64.55 | 64.12 |
| | w/fitted slope PDR | 62.85 | 62.32 | 65.07 | 64.82 | 65.81 | 65.47 | 64.58 | 64.20 |
| | w/LPDR | **63.12** | **63.97** | **65.64** | **66.63** | **66.25** | **67.01** | **65.00** | **65.87** |
| | Ref | 59.57 | 59.54 | 63.27 | 62.92 | 63.93 | 63.91 | 62.26 | 62.12 |
| | w/fitted slope PDR | 59.84 | 59.80 | 63.47 | 63.29 | 64.02 | 64.06 | 62.44 | 62.38 |
| 128 | w/LPDR | **61.42** | **62.56** | **65.07** | **65.90** | **65.05** | **66.16** | **63.85** | **64.87** |
| | Min-$k$% | 66.86 | 64.74 | 69.47 | 67.03 | 70.68 | 68.54 | 69.00 | 66.77 |
| | w/fitted slope PDR | **66.90** | **64.87** | 69.46 | 67.21 | **70.71** | **68.71** | 69.02 | 66.93 |
| | w/LPDR | 64.27 | 63.46 | 67.83 | 68.88 | 69.95 | 69.08 | 67.35 | 67.14 |
| | Min-$k$%++ | 66.32 | 62.67 | 69.77 | 65.88 | 71.83 | 67.70 | 69.31 | 65.42 |
| | w/fitted slope PDR | 66.40 | 62.75 | 69.82 | 66.09 | 71.97 | 67.81 | 69.40 | 65.55 |
| | w/LPDR | **66.80** | **65.77** | **72.39** | **72.20** | **73.50** | **72.19** | **70.90** | **70.05** |

Table 15: AUROC results on sub MIMIR datasets (DM Mathematics, Github, Pile CC). "w/ fitted LPDR" denotes applying LPDR with slope fitted on single test sample loss, while "LPDR" uses a fixed default hyperparameter. Results are bolded if they improve upon their respective baseline (Loss, Min-k%, or Min-k++).

| Method | DM Mathematics | | | | | Github | | | | | Pile CC | | | | |
|---|---|---|---|---|---|---|---|---|---|---|---|---|---|---|---|
| | 160M | 1.4B | 2.8B | 6.9B | 12B | 160M | 1.4B | 2.8B | 6.9B | 12B | 160M | 1.4B | 2.8B | 6.9B | 12B |
| Loss | 48.84 | 48.53 | 48.36 | 48.47 | 48.47 | 65.75 | 69.82 | 71.29 | 73.01 | 73.99 | 49.56 | 50.01 | 50.09 | 50.69 | 51.07 |
| w/ fitted LPDR | 48.84 | 48.53 | 48.36 | 48.47 | 48.47 | 65.76 | 69.83 | 71.29 | 73.01 | 74.00 | 49.56 | 50.02 | 50.09 | 50.69 | 51.07 |
| LPDR | 48.72 | **48.60** | **48.40** | **48.51** | **48.57** | **66.04** | **70.04** | **71.50** | 72.99 | 73.94 | **49.91** | **50.43** | **50.49** | **51.05** | **51.39** |
| Min-k% | 49.27 | 49.29 | 49.10 | 49.16 | 49.21 | 65.65 | 69.96 | 71.44 | 73.26 | 74.24 | 50.10 | 50.47 | 50.46 | 51.20 | 51.48 |
| w/ fitted LPDR | 49.28 | 49.29 | 49.09 | 49.16 | 49.21 | 65.68 | 69.99 | 71.46 | 73.27 | 74.25 | 50.09 | 50.48 | 50.46 | 51.21 | 51.48 |
| LPDR | **49.52** | **49.70** | **49.25** | **49.47** | **49.54** | **65.93** | **70.20** | **71.66** | 73.16 | 74.09 | **50.24** | **51.00** | **51.07** | **51.65** | **51.94** |
| Min-k%++ | 50.12 | 50.17 | 50.21 | 50.52 | 50.42 | 64.66 | 69.63 | 70.88 | 72.80 | 74.18 | 49.67 | 50.04 | 49.78 | 51.19 | 51.75 |
| w/ fitted LPDR | 50.12 | 50.17 | 50.21 | 50.52 | 50.43 | 64.67 | 69.64 | 70.89 | 72.81 | 74.19 | 49.68 | 50.05 | 49.77 | 51.20 | 51.76 |
| LPDR | 50.11 | **50.32** | 50.16 | 50.37 | **50.50** | 64.59 | 69.46 | 70.57 | 72.29 | 73.69 | 49.69 | **50.63** | 49.94 | 51.84 | 51.87 |

## I   ENTROPY ANALYSIS WITH VARYING PREFIX LENGTH

To provide a more rigorous theoretical foundation for our work, we analyze the entropy of the *same* token conditioned on prefixes of varying lengths. According to the principles of information theory, conditioning on more information cannot increase entropy. This implies that for a given token $x_T$, its predictive entropy should monotonically decrease as the length of its conditioning prefix $x_{<T}$ increases:

$$H(x_T|x_{T-1}) \geq H(x_T|x_{T-2}, x_{T-1}) \geq \cdots \geq H(x_T|x_1, \ldots, x_{T-1}). \quad (21)$$

As visualized in Figure 8, we plot the entropy of $x_T$ against the context length $k$ of member samples and non-member samples. We observe that as the prefix length grows, the entropy for predicting the final token decreases for both member and non-member samples. Besides, member samples exhibit a rapid entropy drop-off in the early context window ($k < 10$), while Non-member samples maintain higher entropy for longer. This discriminative gap in the early positions provides the empirical justification for PDR's decay weighting scheme.

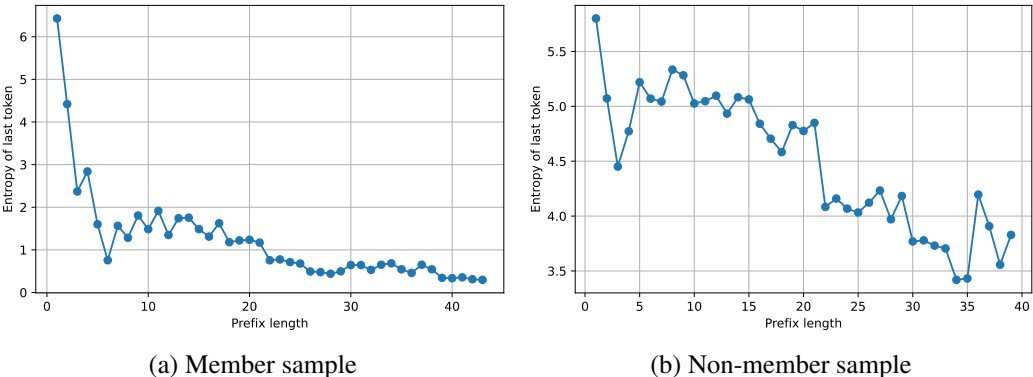

(a) Member sample                                (b) Non-member sample

Figure 8: Visualization of last token entropy changes by given different length prefix on WiliMIA-32 benchmark based on Pythia-6.9B for (a) member sample and (b) non-member sample.

## J   TRUNCATION ANALYSIS

We compare PDR to a simple **Truncation** baseline, which discards a suffix of the sequence before scoring. We varied the truncation percentage (the portion of the sequence retained) to find the optimal performance for each base method. As shown in Figure 9, the optimal truncation percentage is highly inconsistent across different methods, making it difficult to find a single best setting. In contrast, our LPDR method (with a fixed $\alpha = 1.0$, shown as dashed lines) robustly outperforms

even the best possible truncation result for all base methods. This demonstrates that PDR's "soft" reweighting is more effective and reliable than the "hard" cutoff of truncation.

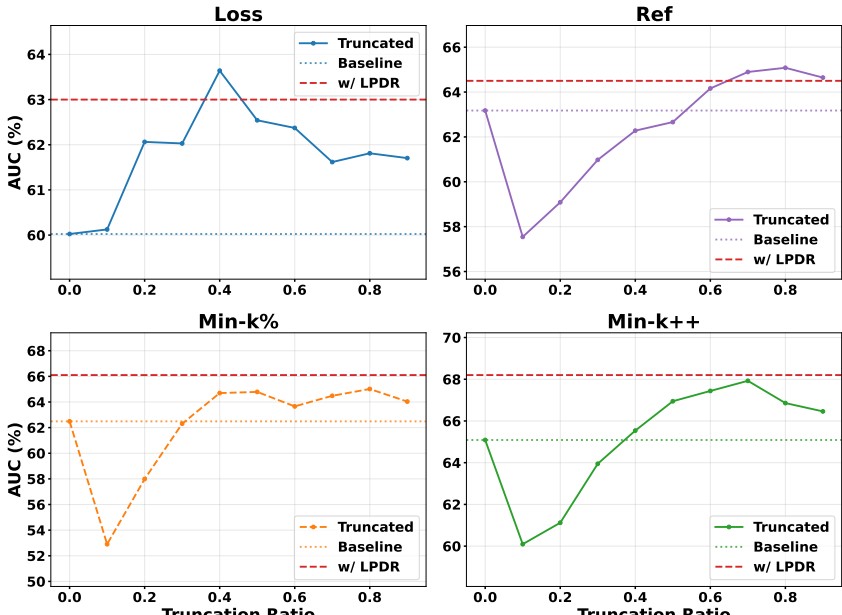

Figure 9: Performance comparison between the Truncation baseline and our LPDR method ($\alpha = 1.0$) on the WikiMIA dataset (length 64, Pythia-12B). The x-axis represents the percentage of the sequence retained for the truncation method. Solid lines show the performance of base methods with truncation, while dashed lines show the performance of the same base methods enhanced with LPDR.

## K  SAMPLE REWEIGHTED ANALYSIS

This section presents additional visualizations of its effect on individual samples. We specifically select pairs of member and non-member samples that are challenging for the baseline Min-$k$% method, meaning their original scores are very close and difficult to distinguish.

Figures 10, 11, and 12 illustrate how applying our LPDR, EPDR, and PPDR methods, respectively, alters the token-level scores for these ambiguous pairs. In each case, the reweighting process amplifies the scores of the member samples more significantly than the non-member samples by emphasizing the low-probability tokens that appear early in the sequence. This creates a more distinct separation between them, demonstrating how PDR enhances detection accuracy at the individual sample level.

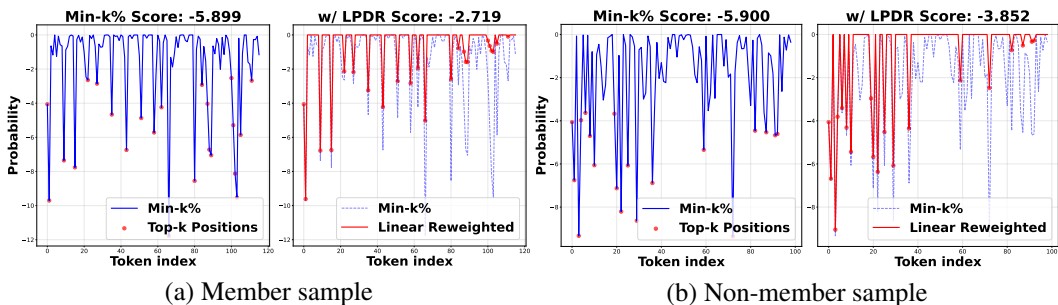

Figure 10: Visualization of token-level score changes for (a) member sample and (b) non-member sample after applying LPDR to the Min-$k$% method.

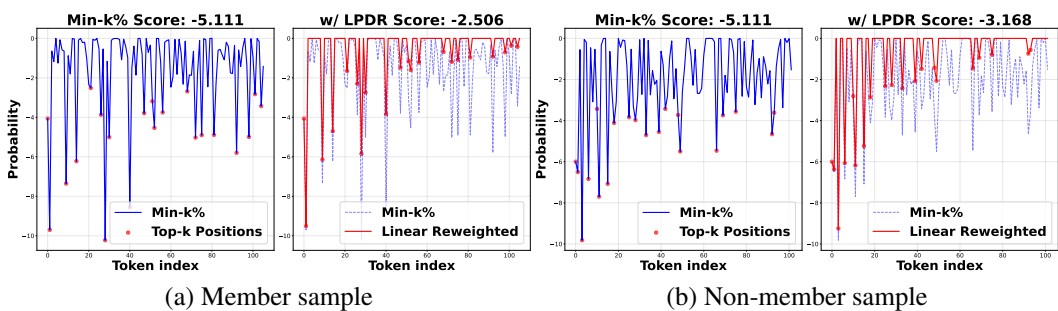

(a) Member sample                                    (b) Non-member sample

Figure 11: Visualization of token-level score changes for (a) member sample and (b) non-member sample after applying EPDR to the Min-$k$% method.

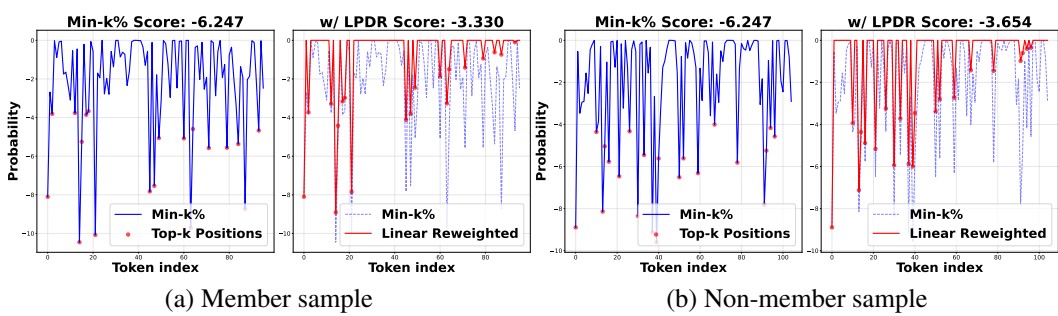

(a) Member sample                                    (b) Non-member sample

Figure 12: Visualization of token-level score changes for (a) member sample and (b) non-member sample after applying PPDR to the Min-$k$% method.

## L  ANALYSIS OF SELECTED TOKEN DISTRIBUTION AND CASE STUDY

We analyze token-level probability distributions to explain PDR's effectiveness. Non-member samples often feature high-surprise factual tokens (e.g., dates) early in the sequence, whereas member samples, being memorized, show low surprise on these tokens. Standard methods dilute this early signal by averaging across the sequence. PDR, by assigning higher weights to the prefix, acts as a "matched filter": it amplifies the informative early tokens while suppressing the noise from later function words.

**Error Study (Figure 15):** While PDR shows consistent improvements in most cases, examining failure cases provides valuable insights into its limitations. Figure 15 shows challenging examples: (a) a member sample that remains misclassified after PDR, and (b) a non-member incorrectly pushed towards a higher score. A key observation is that the highlighted Top-$k$ tokens (yellow background) are distributed uniformly across the sequence rather than concentrated at the start. This anomaly suggests **weak memorization**—the model encountered the text but formed no strong memory trace, possibly due to low training frequency, generic content (common function words lacking distinctive features), or **sentence fragmentation** where the dataset's fixed-length segmentation splits sentences mid-stream, causing the "new sentence start" in the latter half to carry unexpectedly high informativeness and scatter the Top-$k$ tokens. When such positional patterns are absent, PDR's monotonic decay assumption becomes less effective or even counterproductive. These cases highlight potential improvements: adaptive weighting that detects weak memorization or sentence boundaries, and sentence-aware segmentation to preserve natural information flow.

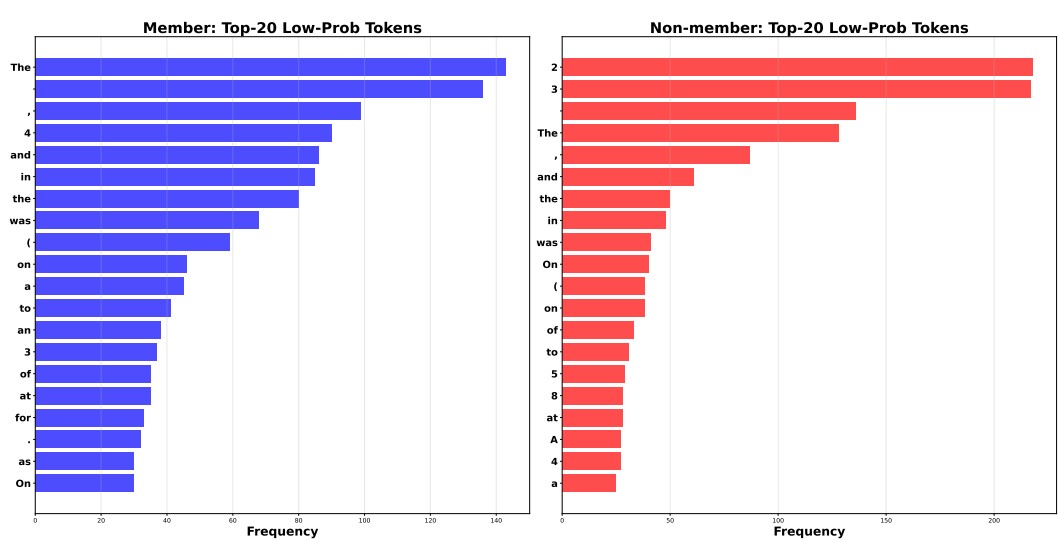

Figure 13: Top $k$ token frequency comparison between member and non-member samples on LLaMA-13B model with 64-token input length on WikiMIA dataset.

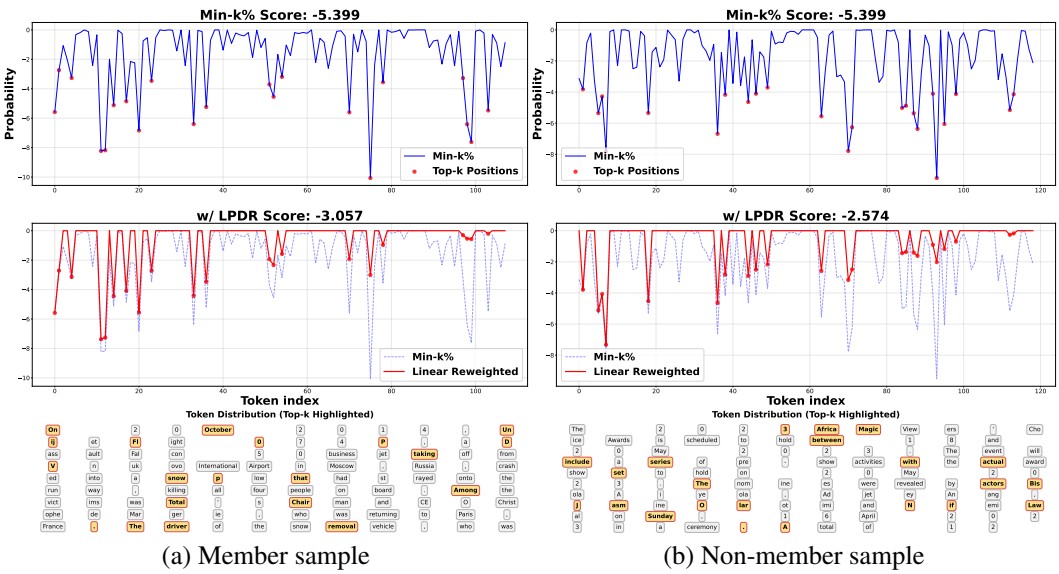

(a) Member sample

(b) Non-member sample

Figure 14: Visualization of token-level score changes and highlight top $k$ tokens for (a) member sample and (b) non-member sample after applying PPDR to the Min-$k$% method.

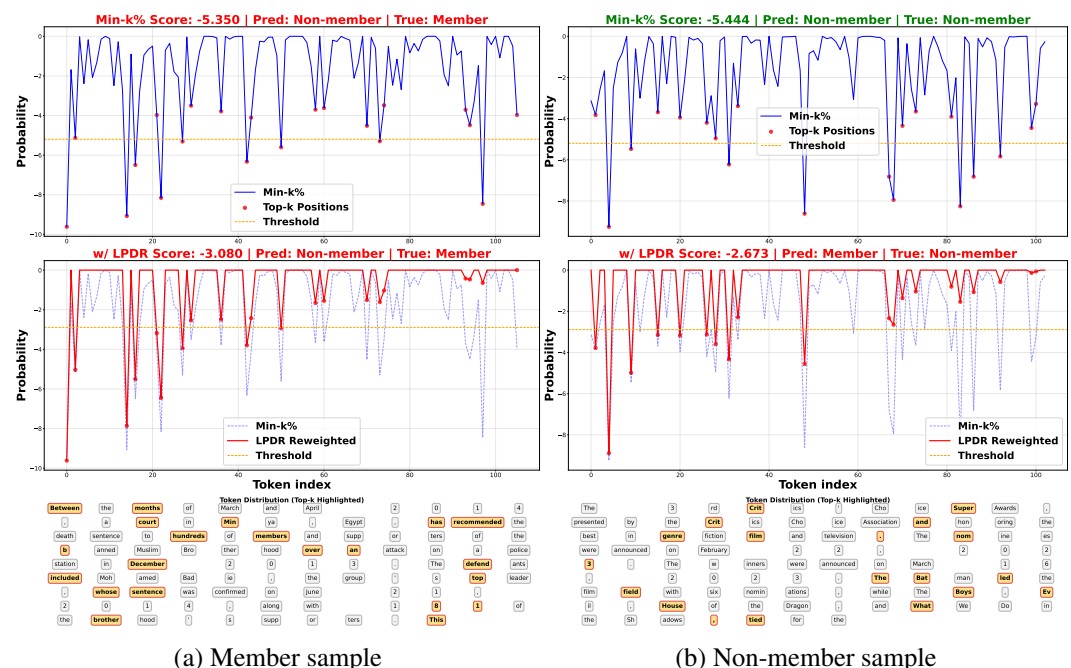

(a) Member sample        (b) Non-member sample

Figure 15: isualization error case about token-level score changes and highlight top $k$ tokens for (a) member sample and (b) non-member sample.

## M   SCORE DISTRIBUTION.

To visually demonstrate the effectiveness of our method, we analyze the score distributions of member and non-member samples before and after applying PDR. Figure. 16 illustrates this comparison for the Min-$k$%++ method on the LLaMA-13B model, using the WikiMIA dataset with a sequence length of 64. For a clearer visualization, the scores are normalized to a range of [0,1]. As the figure shows, the original Min-$k$%++ method already provides some separation between the two distributions. However, after applying our Linear PDR (LPDR), the distributions are pushed further apart. The member sample distribution shifts noticeably towards higher scores, while the non-member distribution remains relatively stable. This increased separation makes it easier to distinguish between member and non-member samples, directly contributing to the improved AUROC performance we observe in our experiments.

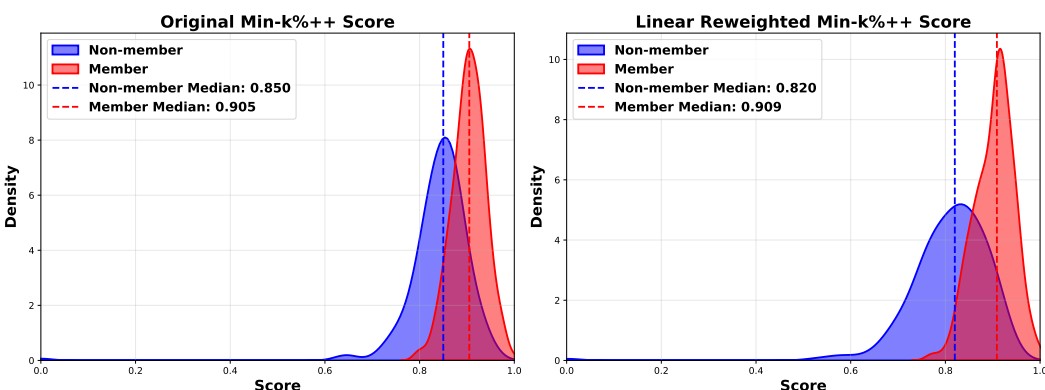

Figure 16: Member and non-member score distribution comparison between Min-$k$% and LPDR-Min-$k$% on LLaMA-13B model with 64-token input length on WikiMIA dataset. Our PDR method enhances the separation between member and non-member distributions.

# N    LLM USE

LLMs were used solely for polishing the writing, e.g., improving clarity and readability. All research ideas, methods, and results were entirely developed and conducted by the authors. The authors take full responsibility for the content of the manuscript.

