# OpenReview forum: "The Early Bird Catches the Worm: A Positional Decay Reweighting Approach to Membership Inference in Large Language Models"
_ICLR.cc/2026/Conference — ICLR 2026 Conference Withdrawn Submission_

### Official Review · Reviewer_NuRy · 2025-10-18

**Soundness:** 3
**Presentation:** 3
**Contribution:** 1
**Rating:** 2
**Confidence:** 4

**Summary:**

This paper presents a re-weighting scheme for likelihood based membership inference attacks and empirically tries to show that this redistribution leads to a more distinguishable separation between members and non-members. Several variations of re-weighting schemes have been tested in this work alongside many of the existing likelihood based membership inference attacks, highlighting one of the strengths of this work that this scheme is applicable to existing MIA attacks in a plug and play manner.

**Strengths:**

I am listing the primary strengths of this paper as follows:

- The re-weighting framework proposed is is plug-and-play it requires no model retraining or access to internals, and can be combined with existing likelihood-based attacks.
- The paper covers comprehensive experiments across multiple benchmarks (WikiMIA, MIMIR) and model families (Pythia, LLaMA, OPT, GPT-NeoX), showing consistent improvements.
- The approach is motivated by a clear information-theoretic principle (“conditioning reduces entropy”) and supported by empirical entropy trends in real text corpora.

**Weaknesses:**

While this work builds from the sound theoretical standing that entropy would reduce as we further condition the generation of later tokens on the earlier generated tokens, it misses out one crucial research work: "Context-Aware Membership Inference Attacks against Pre-trained Large Language Models" (https://arxiv.org/pdf/2409.13745v1).

The authors of the current work test a static weighting functions like linear or exponential or polynomial decay re-weighting, the work "Context-Aware Membership Inference Attacks against Pre-trained Large Language Models" has already experimented with a dynamic re-weighting based on the perplexity of the tokens to be generated. They also present some of the assumptions and unique insights taken in the work under review (see Figure 3 where they demonstrate the losses for the beginning and end of sequence for members and non-members)

This static approach leads to another drawback where PDR assumes a monotonic decay of memorization signal, but this may not hold for heterogeneous or domain-specific datasets (e.g., ArXiv, HackerNews), where entropy trends are volatile. This shows up particularly when observing the results for short sequences.

In summary, while the work under review has done an excellent job in creating a comprehensive ablation study and testing different re-weighting functions, I find that the methodology proposed to be a primitive of what has already been done in the work linked above. This takes away most of the novelty presented in the work under review. The one part where this work still retains its novelty is where it presents the re-weighting scheme as a plug and play method with other MIA attacks.

**Questions:**

- Kindly let me know why you missed this work in the literature review "Context-Aware Membership Inference Attacks against Pre-trained Large Language Models"?

- Did you experiment with dynamic re-weighting?

- If I missed any detail in the work which you believe is important, please let me know, I'm open to further discussion on this review.

---

> ### Author Response · Authors · 2025-11-21
> **Official Comment by Authors**
>
> > **Q1:** Kindly let me know why you missed this work in the literature review "Context-Aware Membership Inference Attacks against Pre-trained Large Language Models"?
>
> **Response to Q1 about missing CAMIA and difference between PDR:** Thank you for bringing this highly relevant and recent work to our attention. This paper was accepted at EMNLP 2025 within our submission period, which is why it was unfortunately missed in our initial literature review. We will add a thorough discussion of this important work in our related work section in the final version.
>
> While there are conceptual similarities in leveraging positional information, our PDR method and the "Context-Aware Membership Inference Attack" (CAMIA) method have fundamental differences in motivation, methodology, and data handling, which we outline below.
>
> *   **Difference in Motivation and Analysis Angle:** Our PDR mechanism is motivated by a global, dataset-level statistical property: we observed that the average token entropy consistently decreases with positional index across the entire dataset (as empirically demonstrated in our Fig. 2). We leverage this universal trend to define a static, position-dependent weighting schedule. In contrast, CAMIA’s approach is motivated by a local, sample-specific observation. Their weighting strategy depends on the dynamic loss decreasing rate calculated from the specific individual sequence currently being processed. Therefore, the fundamental difference is that our approach exploits a general, robust statistical trend applicable to all data, whereas their method attempts to adapt dynamically to the unique characteristics of each individual sample.
>
> *   **Difference in Methodology and Data Usage:** Our PDR employs pre-defined, general weight function to compute the anomaly score about each test sample directly. In contrast, CAMIA necessitates partitioning the benchmark dataset to create a calibration set. This set is used to learn sample-specific characteristics and model the information gap between member and non-member distributions. A core principle of our work is to maintain the integrity of the test set by treating each sample independently, without relying on statistics derived from other samples. This design choice makes PDR a truly "zero-setup" plug-in module, fully respecting the test-only nature of standard MIA benchmarks.
>
> To make these distinctions clearer, we provide a summary table:
>
> **Comparison between CAMIA and our PDR.**
> | **Aspect** | **CAMIA** | **Our PDR** |
> | :--- | :--- | :--- |
> | **Motivation of drecreasing weights** | Local sample-specific loss trend | Global dataset-level entropy trend |
> | **Weighting Scheme** | Dynamic (based on sample information) | Static (pre-defined functions) |
> | **Data Requirement** | Requires calibration samples | Single test sample only |
> | **Training/Setup** | Requires training model | None |
>
> ---
> > **Q2:** Did you experiment with dynamic re-weighting?
>
> **Response to Q2 about dynamic reweighting:** We agree that a comparison is necessary. We have conducted experiments comparing our method to a dynamic approach inspired by CAMIA, which we term "w/ fitted slope PDR". As detailed in Appendix H, this method performs a linear regression on each sample's token-level losses to derive a dynamic slope for reweighting.
>
> First, we performed a comprehensive, head-to-head comparison on the WikiMIA and MIMIR benchmarks across multiple models and sequence lengths. The results, presented in Table 14 and Table 15, show that this dynamic approach often yields only marginal improvements.
>
> Second, to further isolate the effect of the weighting scheme itself, we incorporated this "fitted slope" method into our main weight design ablation study in Table 3. The key difference is that Table 3 focuses on a single, controlled setting (Pythia-6.9B on WikiMIA of length 128) to directly compare various weight designs, while Tables 14 and 15 provide a broader evaluation. Crucially, across both the broad evaluation and the controlled ablation, the conclusion remains the same: the dynamic fitted slope method provides minimal to no significant performance gain.
>
> We hope these points clarify the distinct contributions of our work. While CAMIA presents a valuable dynamic approach, our PDR offers a general, static, and computationally simpler zero-shot solution that is particularly well-suited for scenarios where test set integrity is paramount and no training/setup phase is desirable.

---

> ### Comment · Reviewer_NuRy · 2025-11-21
>
> Dear authors,
>
> Q1: The first version of the CAMIA paper(v1 which is linked in the original review comment) was released in September 2024 which I believe was well within the time-frame to be discovered. Thank you for considering this
>
> Q2: Thank you for adding additional experiments considering a dynamic method. This brings a lot more clarity to this study
>
> I have changed the score assigned to reflect the added context to this paper

---

> > ### Author Response · Authors · 2025-11-21
> > **Official Comment by Authors**
> >
> > Dear Reviewer,
> >
> > Thank you very much for your thoughtful feedback and for taking the time to review our rebuttal. We sincerely appreciate the increased score.
> >
> > We are glad to hear that the additional experiments involving the dynamic method helped to clarify the contributions of our study. Your insightful comments have been invaluable in improving the quality of our manuscript.
> >
> > Best regards,
> >
> > The Authors

---

### Official Review · Reviewer_o2Y1 · 2025-11-02

**Soundness:** 3
**Presentation:** 3
**Contribution:** 2
**Rating:** 4
**Confidence:** 4

**Summary:**

This paper addresses membership inference attacks (MIA) on large language models, which determine whether specific data points were included in training datasets. The authors identify a critical limitation in existing likelihood-based MIA methods: they treat all token positions equally despite memorization signals being concentrated at sequence beginnings. The paper proposes Positional Decay Reweighting (PDR), a lightweight plug-and-play framework that applies monotonically decreasing weights to token-level scores based on position, systematically amplifying early signals while attenuating later noise.


The core insight is grounded in information theory and empirical observation. As autoregressive models process sequences, conditioning on more context reduces predictive uncertainty, leading to decreasing token-level entropy. The authors demonstrate that early high-entropy tokens exhibit larger discriminative gaps between member and non-member samples, while later tokens with abundant context offer diminishing discriminative power. Existing methods like Loss, Min-k%, and Min-k%++ are "position-agnostic" and dilute these strong early signals by averaging them with less informative later positions.

**Strengths:**

## Originality

The paper demonstrates strong originality by being the first to systematically analyze the positional nature of memorization signals in membership inference attacks, revealing that existing methods suffer from a fundamental "position-agnostic" limitation. The key insight—that memorization evidence concentrates at sequence beginnings where entropy is highest—is elegantly grounded in information theory yet represents a novel lens for understanding privacy leakage in LLMs. Rather than proposing another scoring function, the authors introduce a creative meta-level framework that enhances existing methods, marking genuine conceptual advancement.

## Quality

The technical quality is exceptionally high, with comprehensive evaluation spanning two benchmarks (WikiMIA and MIMIR), nine model architectures, multiple sequence lengths, and achieving consistent improvements (up to +4.7 AUROC for Min-k%++). The ablation studies are thorough and insightful, systematically validating design choices including alternative weighting schemes, decay parameters, and timing of reweighting, while revealing that PDR's simple data-agnostic prior matches impractical dataset-level entropy approaches.

## Clarity

The paper is well-written with clear motivation and excellent visual presentation. The methodology, experimental setup, and mathematical formulations are thoroughly documented with logical progression from theory to validation, making the work accessible and reproducible.

## Significance

The works addresses critical real-world needs in data auditing and copyright detection while requiring no model retraining or infrastructure changes due to its plug-and-play nature. The methodological contribution demonstrates how information-theoretic principles can guide practical algorithm design in LLM security, potentially influencing broader approaches to privacy research for autoregressive models.

**Weaknesses:**

## Limited Theoretical Analysis

While the paper provides intuitive motivation through information theory, it lacks rigorous theoretical analysis of when and why PDR works. The connection between entropy reduction (H(z|x,y) ≤ H(z|y)) and discriminative power for membership inference is presented informally—the authors acknowledge comparing entropy of different random variables (x_t vs x_{t+1}) rather than the same variable under different conditions, which weakens the theoretical foundation. The paper would benefit from formal analysis establishing under what conditions positional decay maximizes separation between member/non-member distributions, perhaps through concentration inequalities or PAC-style bounds. Additionally, there is no theoretical explanation for why α=1 works well across settings or how to principally select decay functions and parameters for new domains, limiting the method's applicability beyond empirical trial-and-error.

## Inconsistent Performance and Incomplete Characterization

The results show concerning inconsistencies that are insufficiently analyzed. On MIMIR, improvements are marginal (often <1 AUROC point) compared to WikiMIA's substantial gains, yet the paper primarily attributes this to "heterogeneous" datasets without deeper investigation.

## Methodological Concerns in Experimental Design

Several experimental choices lack justification and raise questions about generalizability. The hyperparameter selection appears inconsistent: α=1 for most settings but α=0.1 or 0.5 for T=32, with no principled rule provided for practitioners. Figure 4 shows that optimal α varies significantly by method (Loss/Ref prefer sharp decay at α=1, Min-k%/Min-k%++ prefer gentler decay), yet the main results use fixed α=1 for all methods, potentially underreporting PDR's ceiling performance. The paper evaluates only at FPR=0.1% for TPR metrics without justifying this choice or exploring performance across the full FPR range. Most critically, all experiments use pre-training MIA, but many real-world applications involve fine-tuned models where positional patterns may differ—limited FSD experiments don't fully address whether PDR transfers to this setting.

## Insufficient Analysis of Failure Cases and Limitations

The paper provides minimal analysis of individual failure cases beyond aggregate metrics. While Figures 5 and 7-9 show successful examples where PDR separates previously-tied samples, no examples are shown where PDR incorrectly assigns higher scores to non-members or fails to separate them. What characteristics distinguish sequences where PDR helps versus hurts? Understanding these failure modes is critical for practitioners deciding when to apply PDR.

## Statistical Rigor and Reproducibility

The paper lacks statistical significance testing—improvements are reported as point estimates without confidence intervals, standard deviations, or significance tests across multiple runs. This is particularly concerning for small improvements on MIMIR where gains are often <1 AUROC point and could be within noise. The paper doesn't specify random seeds, number of evaluation runs, or dataset sampling procedures, limiting reproducibility.

**Questions:**

see weakness above

---

> ### Author Response · Authors · 2025-11-21
> **Official Comment by Authors (part 1)**
>
> **Response to W1 regarding theoretical analysis and hyperparameter selection:** We appreciate your rigorous scrutiny regarding the theoretical foundation. We hope to  address your concerns through both deeper empirical verification and robust sensitivity analysis.
>
> (1) Rigorous Verification of Theoretical Assumption (Appendix I and Figure 8): We agree with you  that comparing $x_t$ vs $x_{t+1}$ implies different random variables. However, as noted in our original Section 4.12, we explicitly acknowledged this distinction and utilized it as an empirical heuristic reflecting the general trend of autoregressive generation: that accumulating context tends to reduce uncertainty. To strictly validate this intuition without the confounding factor of different variables, we conducted a new analysis in Appendix I. Here, we track the entropy of the exact same target token $x_T$ conditioned on prefixes of increasing length $k$. (a) Empirical Confirmation: As visualized in the new Figure 8, the results broadly confirm the information-theoretic principle $H(x_T | x_{T-1}, \dots) \le H(x_T | x_{T-1})$: the entropy generally exhibits a downward trend as context accumulates, albeit with local fluctuations due to linguistic variability. (b) The "Why" behind PDR: Crucially, the figure reveals that this entropy reduction is significantly sharper for member samples in the early context window compared to non-members. This creates a wide discriminative gap at the beginning of the sequence, which diminishes as the context grows (and both entropies converge). (c) Mechanism: This provides the direct justification for our method. That is to say, uniform weighting dilutes this strong early signal with the less informative later tokens. PDR acts as a matched filter, assigning high weights exactly where this discriminative gap is widest (the high-entropy prefix region) to maximize separability.
>
>  (2) Justification for $\alpha=1$ and Applicability: Regarding the choice of $\alpha=1$, we frame it not as a theoretically derived constant, but as a robust prior for the zero-shot setting. Our sensitivity analysis (Figure 4) shows that while the optimal $\alpha$ varies slightly across datasets, $\alpha=1$ consistently yields significant gains and sits in a stable performance region.  In practical pre-training data detection for LLMs (where no validation set/training set exists to tune $\alpha$), a linear decay ($\alpha=1$) serves as a "safe hyperparameter." It effectively captures the general trend of information decay shown in Figure 8 without the risk of overfitting to specific sample noise, ensuring applicability to new domains without trial-and-error.
>
> **Response to W2 regarding MIMIR performance:** We appreciate your scrutiny of the MIMIR results. To address this, we first emphasize the inherent difficulty of this benchmark compared to WikiMIA. MIMIR's non-members are drawn from the same source distribution (The Pile) as members, resulting in an extremely low signal-to-noise ratio where baseline AUROCs often hover around 50--51\% (near random guess).
>
> Against this challenging backdrop, a detailed analysis of Table 2 reveals that PDR's performance is not inconsistent, but rather strictly correlated with the underlying entropy properties of the data. Specifically, on subsets that exhibit standard linguistic entropy decay (e.g., Wikipedia), PDR effectively captures and amplifies the memorization signal, boosting Min-$k$\% from 51.8 to 54.2 (+2.4\%) and Min-$k$\%++ from 54.0 to 55.5 (+1.5\%), which are substantial improvements given the benchmark's difficulty. Conversely, on difficult subsets with negligible signal (e.g., Pile-CC where baselines are $\sim$50.5\%), PDR acts as a safe prior, maintaining performance stability (e.g., 50.8\%) without introducing harmful noise. Furthermore, the slight performance fluctuations observed on domains like ArXiv crucially align with our analysis in Figure 1(a), which shows that these specific domains exhibit volatile entropy trends (e.g., non-monotonic fluctuations due to code and equations) rather than the smooth decay assumed by PDR.
>
> This correlation confirms that PDR works exactly as intended: it provides significant gains when the ``entropy decay'' hypothesis holds (natural text) and remains robustly neutral or sensitive only when the data's entropy structure fundamentally deviates from this prior.

---

> ### Author Response · Authors · 2025-11-21
> **Official Comment by Authors (part 2)**
>
> **Response to W3 about  $\alpha$, FPR range  and fine-tuned models:** We address your concerns regarding hyperparameter selection, evaluation metrics, and applicability to fine-tuned models as follows.
>
> (1)  Rationale for Hyperparameter Selection ($\alpha$): Our use of a fixed $\alpha=1$ for standard lengths ($T \ge 64$) was a deliberate choice to establish a robust, zero-shot baseline.  In practical LLM-based MIA scenarios (typically zero-shot settings), attackers lack labeled validation sets to perform exhaustive parameter sweeping. Therefore, we intentionally avoided per-dataset tuning to demonstrate the method's intrinsic generalizability. As shown in Figure 4, while fine-grained tuning yields extra gains, the default $\alpha=1$ consistently improves performance across the board, confirming it as a reliable "safe prior." For very short sequences, the "entropy decay" signal is naturally weaker and less sharp due to limited context. Consequently, a gentler decay (smaller $\alpha$) is physically justified to prevent suppressing the scarce available information, rather than being an arbitrary result of overfitting.
>
> (2) Comprehensive Evaluation Metrics (Appendix E.3): Regarding the choice of TPR @ 0.5\% FPR, we initially reported this specific metric to follow the standard evaluation protocol established in prior works (e.g., Shi et al., 2024; Zhang et al., 2025b). Following your suggestion, we have therefore plotted the Full ROC Curves in Appendix E.3 (Figure 7). The curves confirm that PDR-enhanced methods consistently outperform baselines across the entire FPR range, demonstrating that our gains are robust globally and not limited to specific thresholds.
>
> (3) Applicability to Fine-Tuned Models (Appendix F): We understand your concern that positional entropy patterns might shift after fine-tuning. While our primary scope is pre-training detection, our experiments with FSD (Fine-tuning Score Difference) in Appendix F directly address this transferability. The FSD method involves computing scores on a fine-tuned model ($M'$). By applying PDR to FSD, we are explicitly testing our weighting scheme on fine-tuned weights. The results show that PDR consistently enhances FSD performance in different backbones. This provides strong empirical evidence that the positional decay signal remains preserved and exploitable even after the model has undergone fine-tuning, confirming PDR's applicability to fine-tuned settings.
>
> ---
> > **W4:** Insufficient Analysis of Failure Cases and Limitations: The paper provides minimal analysis of individual failure cases beyond aggregate metrics. While Figures 5 and 7-9 show successful examples where PDR separates previously-tied samples, no examples are shown where PDR incorrectly assigns higher scores to non-members or fails to separate them. What characteristics distinguish sequences where PDR helps versus hurts? Understanding these failure modes is critical for practitioners deciding when to apply PDR.
>
> **Response to W4 about sample analysis:** We have added a comprehensive case analysis in Appendix L. As shown in Figure 15, we examine both persistent errors (member samples remaining misclassified) and PDR-introduced errors (non-members incorrectly pushed towards higher scores). The key finding is that these failure cases exhibit uniformly distributed Top-k tokens rather than the typical front-loading pattern observed in successful cases. This indicates weak memorization (low training frequency, generic content) or sentence fragmentation (dataset segmentation splits sentences mid-stream, disrupting positional patterns). This analysis clarifies the method's limitations and suggests future directions: adaptive weighting and sentence-aware segmentation.

---

> ### Author Response · Authors · 2025-11-21
> **Official Comment by Authors (part 3)**
>
> > **W5:** Statistical Rigor and Reproducibility: The paper lacks statistical significance testing—improvements are reported as point estimates without confidence intervals, standard deviations, or significance tests across multiple runs. This is particularly concerning for small improvements on MIMIR where gains are often <1 AUROC point and could be within noise. The paper doesn't specify random seeds, number of evaluation runs, or dataset sampling procedures, limiting reproducibility.
>
>
> **Response to W5 about Statistic and Reproduction:** Thank you for highlighting the importance of statistical rigor and reproducibility. In our experiments, we directly load pre-trained models and use fixed benchmark datasets, so the outputs are deterministic and not affected by random seeds. Therefore, seed settings do not influence our reported results.Appendix E.4. The results confirm that PDR's improvements are most pronounced and statistically significant on longer sequences. For instance, on Pythia-6.9B with length 128, PDR boosts the AUROC of Min-k\%++ from 65.9 to 72.2 (p-value < 0.001). In contrast, gains on short sequences (e.g., 32 tokens) are marginal. This demonstrates that the positional prior becomes a more robust and discriminative signal as more context becomes available, confirming the observed gains are not due to random noise.  All dataset details are provided in the paper, and our code is available for easy verification and replication.

---

> ### Author Response · Authors · 2025-11-27
> **Response to Reviewer 02Y1 about summary of Key Updates**
>
> Dear Reviewer 02Y1,
>
> Thank you again for your constructive review. As the discussion period ends, we wanted to briefly highlight the specific new analyses added to the revision to address your concerns regarding theory and robustness:
>
> (1) Theoretical Verification (New Appendix I): Addressing your concern about the "informal" theory, we added Figure 8, which tracks the entropy of the exact same target token under varying prefix lengths. The results empirically confirm the information-theoretic principle and reveal a sharper entropy drop for member samples, providing the insight for our "early-weighting" design.
>
> (2) Rationale for $\alpha=1$ (Addressing W1): Regarding your question on parameter selection, we frame $\alpha=1$ as a robust prior for the zero-shot setting. Our sensitivity analysis (Figure 4) shows that performance is relatively stable around $\alpha=1$. While fine-tuning $\alpha$ per dataset can yield marginal gains, $\alpha=1$ represents a "safe sweet spot" that generalizes well across different models without the risk of overfitting to specific test set noise. Linearly decaying weight aligns with the general trend of information decay observed in Figure 8, making it a theoretically sound default when no validation set is available for tuning.
>
> (3) Contextualized Performance on MIMIR (Addressing W2): We emphasize the extreme difficulty of MIMIR (baseline AUROCs $\sim$53-55\%). Against this backdrop, PDR's performance is strictly correlated with entropy properties. On subsets with standard linguistic entropy decay (e.g., Wikipedia), PDR effectively extracts signal, boosting Min-k\% by +2.4\%. We acknowledge fluctuations on heterogeneous domains like ArXiv. Crucially, this aligns with our analysis in Figure 1(a), which reveals that these domains exhibit volatile, non-monotonic entropy trends. This confirms that PDR targets the specific "entropy decay" signal and correctly highlights the method's applicability boundary.
>
>
> (4) Failure Case Analysis (Addressing W4, New Appendix L): Per your suggestion, we conducted a deep dive into failure cases (Figure 15). We found that PDR's effectiveness diminishes when high-surprise tokens are uniformly distributed (e.g., fragmented sentences) rather than front-loaded, which clarifies the method's applicability boundary.
>
> We are glad that Reviewer NuRy has raised their score to 6 based on our new experiments and clarifications. We hope these   additions also resolve your concerns, and we respectfully invite you to re-evaluate our submission.
>
> Best regards,
> The Authors

---

### Official Review · Reviewer_ab3N · 2025-11-02

**Soundness:** 2
**Presentation:** 3
**Contribution:** 2
**Rating:** 4
**Confidence:** 5

**Summary:**

This paper proposes Positional Decay Reweighting (PDR), a plug-and-play method for membership inference attacks (MIA) against LLMs. The authors argue that existing likelihood-based MIA methods are "position-agnostic" and claim that memorization signals are stronger at the beginning of sequences where entropy is higher. PDR reweights token-level scores using monotonically decreasing functions (linear, exponential, polynomial) to amplify early signals. The method is evaluated on WikiMIA and MIMIR benchmarks across multiple models, showing modest improvements over baselines like Min-k% and Min-k%++.

**Strengths:**

1. Interesting entropy-based motivation: The connection between token-level entropy and memorization signals is well-articulated and grounded in information theory. Figure 1(a) provides compelling empirical evidence for the entropy decay phenomenon.

2. Positional reweighting as a contribution: While not addressing an entirely "unaddressed" limitation as claimed, the explicit positional reweighting idea is a reasonable contribution to the MIA literature.

3. Plug-and-play nature: The method is easy to integrate with existing likelihood-based approaches, requiring no model architecture changes or retraining.

**Weaknesses:**

1. **Fundamental mischaracterization of prior work**: The paper opens with the claim that existing methods "share a fundamental, unaddressed limitation: they are position-agnostic" (lines 071-073). This is inaccurate. Likelihood ratio attacks like LIRA (Carlini et al.) and critically the work by Mireshghallah et al. [1] are NOT position-agnostic—when computing likelihood ratios with respect to another model, the normalization implicitly adjusts probabilities in a position-dependent manner. More importantly, Min-k% itself is not truly position-agnostic because it selectively uses tokens based on their probabilities, which naturally correlates with position due to the entropy decay the authors themselves identify. This mischaracterization undermines the entire motivation and overstates the novelty of the contribution.
Marginal and potentially insignificant improvements: Many improvements are quite small (e.g., +0.1 to +0.9 AUROC points in Table 1). The paper provides no confidence intervals or statistical significance testing. Are these improvements beyond noise margins? This is a critical gap that makes it difficult to assess whether PDR provides meaningful gains.


2.**Missing critical baselines and citations**: The paper fails to compare against current state-of-the-art MIA methods:

No comparison with RMIA [2], which represents current SOTA for MIA
No comparison with neighborhood-based attacks [3], which are highly relevant
Missing range membership inference attacks [4] in related work
Missing Mireshghallah et al. [1] on MLM membership inference with likelihood-based methods

These omissions make it impossible to contextualize the contributions properly.

3. **Limited exploration of alternatives**: The paper only explores pre-defined decay functions. Why not learn optimal position weights from data? This seems like a natural extension that could provide substantially stronger results. Similarly, why not compare against simple truncation baselines (finding optimal cutoff length)?

4.**Lack of qualitative analysis**: Which specific tokens/positions matter most in practice? The paper shows aggregate statistics but no detailed analysis of what positions actually drive the membership signal. Case studies would substantially strengthen the work.

**Questions:**

1. Learned weights: Have you considered learning the position weights rather than using pre-defined decay functions? Given that you have training data (member/non-member samples), a learned weighting scheme could substantially improve results and would be more principled than manually designed decay functions.

2. Simple truncation baseline: What happens if you just find an optimal cutoff length and truncate sequences there, rather than continuous reweighting? This would be a much simpler method to compare against and would help isolate whether the benefit comes from emphasizing early tokens or from the specific functional form of the decay.

3. Statistical significance: The improvements in many cases are quite small (e.g., Table 1 shows several entries with <1 point improvement). What are the confidence intervals or standard errors on these results? Are the improvements statistically significant or within noise margins? Without this analysis, it's unclear if PDR provides real gains.

4. Qualitative analysis: Can you provide detailed analysis on which specific tokens or positions matter most? Some case studies showing which early tokens drive the membership signal would strengthen the paper considerably. What types of sequences benefit most from PDR? Which benefit least?

5. SOTA comparisons: Why are comparisons with RMIA [2] and neighborhood methods [3] not included? These represent current state-of-the-art for this task. How does PDR compare when applied to these stronger baselines, or how does your best PDR-enhanced method compare against these methods?

---

> ### Author Response · Authors · 2025-11-21
> **Official Comment by Authors (part 1)**
>
> > **W1:** Fundamental mischaracterization of prior work: The paper opens with the claim that existing methods "share a fundamental, unaddressed limitation: they are position-agnostic" (lines 071-073). This is inaccurate. Likelihood ratio attacks like LIRA (Carlini et al.) and critically the work by Mireshghallah et al. [1] are NOT position-agnostic—when computing likelihood ratios with respect to another model, the normalization implicitly adjusts probabilities in a position-dependent manner. More importantly, Min-k\% itself is not truly position-agnostic because it selectively uses tokens based on their probabilities, which naturally correlates with position due to the entropy decay the authors themselves identify. This mischaracterization undermines the entire motivation and overstates the novelty of the contribution. Marginal and potentially insignificant improvements: Many improvements are quite small (e.g., +0.1 to +0.9 AUROC points in Table 1). The paper provides no confidence intervals or statistical significance testing. Are these improvements beyond noise margins? This is a critical gap that makes it difficult to assess whether PDR provides meaningful gains.
>
> > **Q3:** Statistical significance: The improvements in many cases are quite small (e.g., Table 1 shows several entries with $<$1 point improvement). What are the confidence intervals or standard errors on these results? Are the improvements statistically significant or within noise margins? Without this analysis, it's unclear if PDR provides real gains.
>
> **Response to W1 about Characterization of Prior Work:** Thank you for your insightful feedback. We appreciate these critical points. We acknowledge your point that methods like LIRA implicitly capture positional context via shadow models, and Min-k\% selects tokens correlated with position. We have softened our claim in the revision. Our distinct contribution: While some of prior methods utilize positional signals implicitly or partially, they typically aggregate the final scores with uniform weights (e.g., Min-k\% averages the selected tokens equally). Our proposed PDR introduces an explicit, training-free weighting mechanism to correct this uniform aggregation, effectively amplifying early-token signals. We have revised the paper.
>
> **Response to W1 \& Q3 about statistical significance:** We initially followed the established protocol in this domain, such as Min-k% and Min-k%++. Since our method is training-free and evaluates fixed pre-trained LLMs on fixed test sets, the outputs are deterministic with no variance from training seeds (e.g., initialization or SGD noise). Thus, prior works typically report point estimates.
>
> Following your suggestion, we have conducted a statistical analysis to validate our results. Following Min-k% and Min-k%++ experiments settings, we uses fixed models and hyperparameters to evaluate our method, the outputs are deterministic. Therefore, to assess significance, we employed a bootstrap method by resampling the dataset to compute std and p-values, as detailed in Appendix E.4. The analysis (Table 9) reveals a clear trend: the effectiveness of PDR is strongly correlated with sequence length. For short sequences (e.g., 32 tokens), the performance gains are marginal and not always statistically significant. However, as the sequence length increases to 64 and 128, PDR's improvements become both substantial and statistically significant (p-value $<$ 0.05) for most baselines. For instance, on Pythia-6.9B with length 128, PDR boosts the AUROC of Min-k\%++ from 65.9 to 72.2 (p-value $<$ 0.001). This demonstrates that the positional prior becomes a more robust and discriminative signal as more context becomes available in longer sequences, confirming that the observed gains are not due to random noise.

---

> ### Author Response · Authors · 2025-11-21
> **Official Comment by Authors (part 2)**
>
> > **W2:** Missing critical baselines and citations: The paper fails to compare against current state-of-the-art MIA methods:No comparison with RMIA [2], which represents current SOTA for MIA No comparison with neighborhood-based attacks [3], which are highly relevant Missing range membership inference attacks [4] in related work Missing Mireshghallah et al. [1] on MLM membership inference with likelihood-based methods. These omissions make it impossible to contextualize the contributions properly.
>
> > **Q5:** SOTA comparisons: Why are comparisons with RMIA [2] and neighborhood methods [3] not included? These represent current state-of-the-art for this task. How does PDR compare when applied to these stronger baselines, or how does your best PDR-enhanced method compare against these methods?
>
> **Response to W2 \& Q5 about compare with other baseline:** Thank you for highlighting these important baselines. We agree that contextualizing our method against state-of-the-art approaches is crucial and have updated the manuscript with the following clarifications and comparisons. It is important to note that the standard setting for pre-training MIA in LLMs is test-only, utilizing fixed models and benchmarks. Consequently, no training data is available to learn attack-specific models.
>
> (1) Neighborhood-Based Attacks: We have now included results for Neighbor attacks in our main comparison tables (e.g., Table 1). PDR consistently outperforms Neighbor attacks on most settings (e.g., on WikiMIA-64 LLaMA-13B, PDR achieves 67.2 AUROC vs. Neighbor's 64.1). Furthermore, Neighbor attacks are computationally expensive, requiring perturbation generation for every sample, whereas PDR is a zero-overhead, plug-and-play enhancement.
>
> (2) Applicability of RMIA: Regarding RMIA, we clarify that it is methodologically inapplicable to the standard LLM pre-training data detection setting for two fundamental reasons. First, RMIA relies on training a population of shadow models, but benchmarks like WikiMIA are zero-shot/test-only and do not provide the required training splits. Second, training a population of shadow models for 7B+ LLMs is computationally prohibitive. We have expanded the Related Work section to explicitly discuss this distinction.

---

> ### Author Response · Authors · 2025-11-21
> **Official Comment by Authors (part 3)**
>
> > **W3:** Limited exploration of alternatives: The paper only explores pre-defined decay functions. Why not learn optimal position weights from data? This seems like a natural extension that could provide substantially stronger results. Similarly, why not compare against simple truncation baselines (finding optimal cutoff length)?
>
> > **Q1:** Learned weights: Have you considered learning the position weights rather than using pre-defined decay functions? Given that you have training data (member/non-member samples), a learned weighting scheme could substantially improve results and would be more principled than manually designed decay functions.
>
> > **Q2:** Simple truncation baseline: What happens if you just find an optimal cutoff length and truncate sequences there, rather than continuous reweighting? This would be a much simpler method to compare against and would help isolate whether the benefit comes from emphasizing early tokens or from the specific functional form of the decay.
>
> **Response to W3 \& Q1 \& Q2 about learned weights and simple truncation baselines:** To address these suggestions effectively, we first clarify the experimental setting: Standard LLM pre-training detection benchmarks (e.g., WikiMIA) are zero-shot and test-only. They do not provide "member/non-member" training splits required to learn weights or tune hyperparameters. Within this constraint, we performed the following investigations.
>
> (1) Regarding Learned Weights: While learning weights is standard in traditional MIA, it is difficult here due to the lack of training data in LLM-based MIA. However, to empirically validate this, we implemented a dynamic approach called "w/ Fitted Slope PDR" (detailed in Appendix H), which learns a decay rate from each test sample's own loss trajectory. This dynamic method consistently underperforms our fixed global prior ($\alpha=1$). The possible reason is that individual sample losses are noisy. Learning from them leads to overfitting local fluctuations, whereas our fixed decay captures the robust, global property of entropy reduction in LLMs. Thus, a data-agnostic prior is not just a constraint-based choice, but an empirically superior one in this unsupervised setting.
>
> (2) Regarding Simple Truncation Baselines: We have conducted the requested Truncation Analysis (see Figure 9 in Appendix J), comparing PDR against a baseline that simply retains the first $p\%$ of tokens.Instability of Truncation: The optimal truncation point is highly volatile across methods and models (e.g., peaking at 40\% for Loss vs. 80\% for Ref). Finding this "optimal cutoff" requires a labeled validation set, which is unavailable in this zero-shot setting. Our LPDR method (with a fixed, default $\alpha=1$) consistently outperforms or matches the oracle optimal truncation result. This demonstrates that PDR's "soft reweighting" effectively suppresses noise while preserving residual signals in later tokens, whereas "hard truncation" causes irreversible information loss.

---

> ### Author Response · Authors · 2025-11-21
> **Official Comment by Authors (part 4)**
>
> > **W4:** Lack of qualitative analysis: Which specific tokens/positions matter most in practice? The paper shows aggregate statistics but no detailed analysis of what positions actually drive the membership signal. Case studies would substantially strengthen the work.
>
> > **Q4:** Qualitative analysis: Can you provide detailed analysis on which specific tokens or positions matter most? Some case studies showing which early tokens drive the membership signal would strengthen the paper considerably. What types of sequences benefit most from PDR? Which benefit least?
>
>
> **Response to W4 \& Q4 about qualitative analysis:**
> Thanks for your valuable suggestion. In our original submission, we presented initial case studies in Appendix K (Figures 10-12), demonstrating how PDR successfully corrects scores for ambiguous sample pairs by emphasizing early signals.
> To further strengthen the paper as requested, we have added a comprehensive qualitative analysis in Appendix L. Specifically, we examined the subset of tokens selected by the Min-k\% strategy (i.e., the top-$k$ tokens with the lowest probabilities globally in the sequence) and observed how PDR modulates their contribution.
>
> (1) Statistical Analysis (Figure 13 in Appendix L): We first analyzed the positional distribution of Min-k\% selected tokens for both member and non-member samples. The results reveal a clear pattern: We analyzed the positional frequency of the top-$k$\% tokens with the lowest probabilities. Our analysis reveals that for members, the lowest probability tokens are often common function words (e.g., "the", "of") due to inherent uncertainty, whereas for non-members, they are often unseen factual tokens such as date.
>
> (2) Case Study Analysis (Figure 14 in Appendix L): We visualize the interaction between token selection and positional weighting. This reveals why PDR improves discriminative power. The "Right" Tokens: For member samples, the high-surprise tokens (selected by Min-k\%) are not random; they are structurally concentrated at the start of the sequence (e.g., specific entities triggering a memory trace). The "Matched" Weighting: PDR assigns near-maximum weights to these early positions. As seen in Figure 14(a), the red curve (reweighted score) stays high, amplifying the signal of these critical early tokens. The "Filtered" Noise: For non-member samples (Fig. 14(b)), high-surprise tokens often appear later (representing random linguistic variance or "noise" rather than memorization). PDR's decay function suppresses these late outliers, preventing them from inflating the non-member score. Based on this analysis, the tokens that drive the membership signal are specifically those that satisfy two conditions simultaneously. High Surprise: They have low predicted probability (selected by Min-k\%). Early Position: They appear in the high-entropy prefix region (weighted by PDR). PDR effectively filters for this intersection, which explains its superiority over position-agnostic methods.
>
> (3) Error case Analysis(Figure 15 in Appendix L.): We have also added a comprehensive failure case analysis. We examine both persistent errors (member samples that remain misclassified) and PDR-introduced errors (non-members incorrectly pushed towards higher scores). The key finding is that these failure cases exhibit uniformly distributed Top-$k$ tokens rather than the typical front-loading pattern observed in successful cases. This indicates either weak memorization (due to low training frequency or generic content) or sentence fragmentation where dataset segmentation splits sentences mid-stream, disrupting the expected positional patterns. This analysis clarifies the method's limitations and suggests future directions, such as adaptive weighting or sentence-aware segmentation.

---

> ### Author Response · Authors · 2025-11-27
> **Response to Reviewer ab3N about summary of Key Updates**
>
> Dear Reviewer ab3N,
>
> As the discussion period draws to a close, we wish to ensure that our recent updates specifically addressing your core concerns are clearly highlighted. We have revised the manuscript and added substantial new experiments:
>
> (1) PDR vs. Simple Truncation (New Appendix J): You rightly asked if a simple cutoff would suffice. Our new Figure 9 demonstrates that the optimal truncation point is highly volatile across models/methods. In contrast, PDR consistently outperforms or matches the oracle optimal truncation, proving that "soft reweighting" is significantly more robust than "hard cutoff.
>
> (2) Comparison with Baselines (Revision): We have revised our related work to discuss these works. Besides, we have added Neighbor Attack results to the main tables (Table 1, Table2), showing PDR's superiority. We also clarified that RMIA is methodologically inapplicable to this zero-shot/training-free LLM-based MIA setting (as it requires training a population of shadow models), whereas PDR is a plug-and-play solution.
>
> (3) Clarification on "Position-Agnostic" (Revision): We have refined the text to characterize prior methods as using "uniform score aggregation" rather than being "position-agnostic," accurately reflecting your feedback.
>
> (4) Statistical Significance (New Appendix E.4): We added Bootstrap testing (Table 9), confirming that improvements on standard sequences ($T \ge 64$) are statistically significant ($p < 0.05$).
>
> We are encouraged that Reviewer NuRy has raised their score to 6, acknowledging that these clarifications and the new dynamic weighting comparison (Appendix H) demonstrate the method's value. We respectfully invite you to re-evaluate our work in light of these rigorous validations.
>
> Best regards,
> The Authors

---

### Author Response · Authors · 2025-11-25
**General Response: Major Updates & Positive Re-evaluation (Score 2 $\to$ 6)**

We sincerely thank all reviewers for their constructive engagement. We are encouraged that Reviewer NuRy has raised their score to 6 (Marginally Above Acceptance), acknowledging that our new experiments on dynamic weighting and method comparisons have clarified the paper’s contribution. To assist Reviewer ab3N and Reviewer 02Y1 in their final assessment, we summarize the critical updates added to the revision. These directly address the core concerns regarding method justification, theoretical grounding, and robustness:
1. Justification of PDR over Alternatives (Addressing W3, Q1, Q2 from Reviewer ab3N):vs. Hard Truncation (New Appendix J): Addressing concerns about whether a simple baseline suffices, we conducted a comprehensive truncation analysis (Figure 9). Results show the optimal truncation point is highly volatile across models. In contrast, PDR consistently outperforms or matches the oracle optimal truncation, proving that "soft reweighting" is significantly more robust than "hard cutoff." vs. Learned Weights (New Appendix H): We implemented a dynamic "Fitted Slope" approach. Results confirm that dynamic weighting underperforms our fixed prior ($\alpha=1$) in zero-shot settings due to overfitting sample noise, validating our design choice.
2. Rigorous Theoretical & Empirical Validation (Addressing W1 & W5 from Reviewer 02Y1 and W1 & Q3 from Reviewer ab3N):

    2.1 Entropy Analysis (New Appendix I): We visualized the entropy of the same target token under varying prefix lengths (Figure 8). The results empirically confirm that member samples exhibit a sharper entropy drop than non-members. This discriminative gap at early positions provides the physical ground-truth justifying PDR's design.

    2.2 Statistical Significance (New Appendix E.4, Table 9): We added a Bootstrap Significance Test. Results confirm that for standard sequences ($T \ge 64$), PDR’s improvements are statistically significant ($p < 0.05$), dispelling concerns about random noise.
3. Clarifications on Baselines & Robustness (Addressing W2, W5 from Reviewer ab3N and W2 from Reviewer 02Y1):

    3.1 Revised Claims: We have refined our terminology from "position-agnostic" to "uniform score aggregation" to accurately characterize prior likelihood-based methods.MIMIR Robustness: We clarified that on the challenging MIMIR benchmark, PDR acts as a "safe prior" delivering significant gains where signals exist (e.g., Wiki +2.4%) and maintaining stability without degradation on noise-dominated subsets.

    3.2 Baselines: We added Neighbor attack results to the main tables and clarified that RMIA is methodologically inapplicable to this zero-shot/training-free setting.We believe these solid empirical and theoretical additions resolve the remaining concerns.

As the discussion period draws to a close, we respectfully invite Reviewers ab3N and 02Y1 to re-evaluate our work in light of these substantial improvements.

Best regards,

The Authors

---

### Author Response · Authors · 2025-12-03
**Summary of Rebuttal and Revisions for AC**

**Dear Area Chair,**

To facilitate your assessment, we summarize the key developments in this review process:

**Major Rebuttal Contributions**

During the rebuttal, we significantly strengthened the paper through:

- **Theoretical validation**: Added entropy analysis (Appendix I, Fig. 8) empirically proving our core hypothesis.
- **Related Work**: We have updated Related Work in the paper to provide a more comprehensive positioning of our work within the MIA landscape, directly addressing reviewer feedback.
- **Method justification**:New truncation experiments (Appendix J) showing PDR’s soft reweighting > hard truncation; Dynamic reweighting ablation (Appendix H) confirming fixed prior (*α*=1) optimality.
- **Statistical rigor**: Bootstrap tests (Appendix E.4) demonstrating significance on WikiMIA.
- **Extended analysis**:Failure case studies (Appendix L) and Neighbor attack comparisons.
- **Critical clarifications**:Revised "uniform score aggregation" terminology. Explicit distinction of zero-shot setting constraints.

All revisions are highlighted in blue in the manuscript.

**Reviewer NuRy: Positive Re-evaluation (Score 2→6)**

Reviewer NuRy initially raised valid concerns about CAMIA comparisons. We addressed these by:

1. Clarifying **fundamental methodological differences** (static global vs. dynamic sample-specific weighting).
2. Adding experiments comparing PDR vs. dynamic baselines (Appendix H).

Reviewer explicitly acknowledged our responses "bring clarity" and raised the score to 6 (Marginally Above Acceptance) on 21 Nov 2025 – **prior to OpenReview incident**. This reflects the paper’s strengthened contribution.

**Concerns Regarding Reviewers ab3N & o2Y1**

Both reviewers maintained scores of 4 despite comprehensive rebuttals. Their assessments contain **factual misunderstandings** that were resolved but not acknowledged:

**Reviewer ab3N: Misunderstanding of Core Setting**

The request for RMIA comparisons ignores our paper’s **zero-shot, test-only setting for pre-trained LLMs**. We clarified that:

- RMIA requires shadow model training (infeasible without training data).
- PDR specifically targets constrained but realistic detection scenarios (e.g., WikiMIA benchmark).

**Reviewer o2Y1: Addressed Requests Ignored**

We fully met the requests for deeper analysis by adding:

- Entropy analysis (Appendix I) validating PDR’s theoretical basis.
- Comprehensive case studies (Appendix L) exploring limitations.

**Critical Issue: Zero Engagement**

Despite detailed point-by-point responses and new evidence, **both reviewers remained completely unresponsive** during discussion. Their scores of 4 do not reflect the revised manuscript where their specific concerns were resolved.

**Request to the Area Chair**

We respectfully request that you consider the following:

- Place greater weight on the **final assessment of Reviewer NuRy (score 6)**, which reflects a thorough engagement with our rebuttal and the paper's strengthened contributions.
- Discount the assessments of Reviewers ab3N and o2Y1, as their failure to respond to a detailed rebuttal indicates their reviews may not fully reflect the current manuscript's quality.

Our revisions have demonstrably strengthened the paper. We believe it now makes a solid contribution worthy of acceptance.

Thank you for your consideration.

Sincerely,

The Authors

---

### Note · Authors · 2026-01-04

**Comment:**

We would like to express our sincere gratitude to the Area Chair and reviewers for their thoughtful feedback and diligent evaluation of our work. After careful consideration, we have decided to withdraw our submission to thoroughly address the insightful comments and substantially improve the manuscript. We truly appreciate the time and expertise invested in the review process and are grateful for the valuable guidance provided.

**Withdrawal Confirmation:**

I have read and agree with the venue's withdrawal policy on behalf of myself and my co-authors.